# SLC35D3 promotes white adipose tissue browning to ameliorate obesity by NOTCH signaling

Hongrui Wang[1,3], Liang Yu[1,3], Jin'e Wang[2,3], Yaqing Zhang[2], Mengchen Xu[1], Cheng Lv[1], Bing Cui[1], Mengmeng Yuan[1], Yu Zhang[1], Yupeng Yan[1], Rutai Hui[1] & Yibo Wang [1]✉

White adipose tissue browning can promote lipid burning to increase energy expenditure and improve adiposity. Here, we show that Slc35d3 expression is significantly lower in adipose tissues of obese mice. While adipocyte-specific *Slc35d3* knockin is protected against diet-induced obesity, adipocyte-specific *Slc35d3* knockout inhibits white adipose tissue browning and causes decreased energy expenditure and impaired insulin sensitivity in mice. Mechanistically, we confirm that SLC35D3 interacts with the NOTCH1 extracellular domain, which leads to the accumulation of NOTCH1 in the endoplasmic reticulum and thus inhibits the NOTCH1 signaling pathway. In addition, knockdown of Notch1 in mouse inguinal white adipose tissue mediated by orthotopic injection of AAV8-adiponectin-shNotch1 shows considerable improvement in obesity and glucolipid metabolism, which is more pronounced in adipocyte-specific *Slc35d3* knockout mice than in knockin mice. Overall, in this study, we reveal that SLC35D3 is involved in obesity via NOTCH1 signaling, and low adipose SLC35D3 expression in obesity might be a therapeutic target for obesity and associated metabolic disorders.

Obesity is one of the major global health challenges with an increasing global incidence, leading to higher rates of morbidity and mortality due to obesity-related complications, such as type 2 diabetes, hypertension, metabolic syndrome, accelerated cardiovascular diseases and increased cancer risks[1–4]. While reducing caloric intake is an effective way to combat obesity, an alternative strategy would be to modify metabolic efficiency and increase energy expenditure in critical metabolic tissues, such as adipose tissues[5–7].

Classical brown adipose tissue (BAT) dissipates extra energy to generate heat through uncoupled respiration mediated by uncoupling protein 1 (UCP1), thus increasing energy expenditure and counteracting obesity[8–10]. Interestingly, brown adipocytes and beige adipocytes coexist in human adipose depots, which are highly heterogeneous[11,12]. Beige adipocytes have multilocular lipid droplets

and high mitochondrial content and play an important role in energy homeostasis[13,14]. Due to their remarkable plasticity and association with human obesity, beige adipocytes are an attractive therapeutic target for counteracting the obesity pandemic.

NOTCH, a cell membrane protein, is activated by NOTCH receptor (NOTCH1-NOTCH4) binding of Dll or Jag family ligands, leading to γ-secretase-mediated proteolytic cleavage[10,15–17]. The released intracellular domain of NOTCH (NICD) subsequently enters the nucleus to modify gene expression and then regulates a variety of important biological events[18,19]. Previous studies have indicated that the NOTCH pathway inhibits the browning of white adipose tissue (WAT) and regulates body energy homeostasis[10,20].

SLC35D3 (solute carrier family 35, member D3) is predicted to be an orphan nucleotide sugar transporter or a fringe connection-like

---

[1]State Key Laboratory of Cardiovascular Disease, Fuwai Hospital, National Center for Cardiovascular Diseases, Chinese Academy of Medical Sciences and Peking Union Medical College, Beijing, China. [2]College of Basic Medical Sciences, China Three Gorges University, Yichang, Hubei, China. [3]These authors contributed equally: Hongrui Wang, Liang Yu, Jin'e Wang. ✉e-mail: yibowang@hotmail.com

protein with 10 transmembrane domains[21]. In associated studies, the *Slc35d3* gene was found to be related to the obesity phenotype[22,23]. It has also been reported that mutation of *Slc35d3* causes metabolic syndrome by impairing dopamine signaling in striatal D1 neurons in mice[21,24]. However, the role of SLC35D3 in adipose tissues has not been clearly elucidated.

In this study, we show that SLC35D3 regulates white adipose tissue browning and obesity via NOTCH1 signaling. While *Slc35d3* knockout in mice can lead to increased susceptibility to obesity and impaired glucose tolerance, *Slc35d3* knock-in can promote WAT browning and protect against obesity. We also confirm that SLC35D3 overexpression increases ER accumulation of NOTCH1 and thus inhibits NOTCH1 signaling, which is achieved by interacting with the NOTCH1 extracellular domain (ECD). Our study reveals a regulatory function of SLC35D3 in obesity, and low SLC35D3 expression in adipose tissues might be a potential therapeutic target for obesity and associated metabolic disorders.

## Results

### Slc35d3 expression was lower in the fat tissues of obese mice

First, Slc35d3 tissue distribution was analyzed in various tissues of wild-type (WT) mice, including inguinal white adipose tissue (IngWAT), epididymal WAT (EpiWAT), anterior-subcutaneous WAT (AsWAT), brown adipose tissue (BAT), kidney, liver, spleen, heart, and brain. Quantitative PCR (qPCR) and western blot analyses indicated that Slc35d3 was relatively highly expressed in IngWAT, BAT, and brain (Fig. 1a, b). To elucidate the correlation between SLC35D3 and obesity, we examined Slc35d3 expression in adipose tissues from obese mice, including *leptin* deletion-induced obese (ob/ob) mice and diet-induced obesity (DIO) mice (Fig. 1c, f). Slc35d3 expression levels in the IngWAT, EpiWAT, and BAT were significantly lower in obese mice than in WT mice (Fig. 1d, e, g, h). Taken together, these findings suggested that Slc35d3 expression was lower in the adipose tissues of obese mice.

### Adipose-specific *Slc35d3* knockout induced obesity in mice

To understand the specific role of adipocyte SLC35D3 in regulating obesity and adipose inflammation, we generated adipocyte-specific *Slc35d3* knockout mice using the adiponectin Cre-lox system. We bred mice carrying the two loxP-flanked exon 2 of *Slc35d3* with transgenic mice expressing Cre recombinase driven by the *adiponectin* promoter to generate *Slc35d3*-Flox; *Adiponectin*-Cre[+/−] mice (SAKO). Mice with the floxed *Slc35d3* alleles, which did not express Cre recombinase (*Slc35d3*-Flox; *Adiponectin*-Cre[−/−]), were used as controls and hereafter referred to as Flox mice (Fig. 2a). As expected, qPCR analysis revealed a substantial reduction in *Slc35d3* expression in the IngWAT, EpiWAT, AsWAT and BAT of SAKO mice (Fig. 2b), while there was no change in other tissues, indicating the specific knockout of SAKO mice. Immunoblot analysis further revealed that Slc35d3 protein levels were greatly reduced in the adipose tissues of SAKO mice (Fig. 2c).

We then fed both SAKO and Flox mice a normal chow diet (NCD) to investigate the functional significance of adipocyte-specific *Slc35d3* deletion and found that the body weight was higher in the NCD-fed SAKO mice (Fig. 2d), which was evident in both males and females, whereas their food intake and body lengths did not change significantly (Fig. 2e, f). Since male and female mice showed no apparent differences, only the experimental data from male mice are shown in our subsequent results. We subsequently isolated different adipose tissues from SAKO and Flox mice for weighing, and the results showed that the weights of different adipose tissues (IngWAT, EpiWAT, AsWAT and BAT) were significantly higher in SAKO mice (Fig. 2g). In addition, serum triglyceride (TG), total cholesterol (TC) and free fatty acid (FFA) levels were also higher in SAKO mice (Fig. 2h). Obesity is associated with many health problems, including glucose intolerance and insulin resistance[1,25]. To assess glucose homeostasis in the NCD-fed SAKO and

Flox mice, a glucose tolerance test (GTT) and insulin tolerance test (ITT) were performed, and the results showed that glucose tolerance and insulin sensitivity were significantly impaired in SAKO mice (Fig. 2i, j). Hyperinsulinemic–euglycemic clamps were then performed to further characterize insulin sensitivity in male SAKO mice and Flox mice. The glucose infusion rate required to maintain euglycemia was significantly lower in SAKO mice (Fig. 2k, l). Whole-body insulin-stimulated glucose turnover was significantly downregulated (Fig. 2m), while 2-deoxyglucose uptake in the IngWAT of SAKO mice was also markedly reduced (Fig. 2n). All these results demonstrated that genetic disruption of *Slc35d3* in adipose tissues impaired glucose homeostasis and insulin sensitivity.

Adipose tissue enlargement is associated with a prominent inflammatory response in the visceral compartment and infiltrating macrophages in EpiWAT are key mediators of this inflammatory process[1]. Higher expression levels of macrophage marker gene (*F4/80*) and chronic inflammation-related genes (*Il-1β*, *Il-6*, *Tnf-α*, and *Ccl2*) in EpiWAT were observed in SAKO mice (Supplementary Fig. 1a–d). Ectopic lipid accumulation results in lipotoxic metabolic stress, which promotes metabolic dysfunction in organs such as the liver[1,26]. To investigate whether adipocyte-specific *Slc35d3* deletion affected hepatic lipid accumulation, Flox, and SAKO mouse livers were isolated for testing. As the results showed, the liver weight in the SAKO group was markedly higher (Supplementary Fig. 1e), and H&E staining results revealed that the livers of obese SAKO mice were more steatotic than those of controls (Supplementary Fig. 1f). Furthermore, hepatic TG, TC, and FFA contents were also measured. As expected, markedly higher TG, TC, and FFA levels were found in the livers of SAKO mice (Supplementary Fig. 1g).

Taken together, these data indicated that adipose-specific *Slc35d3* knockout induced obesity and hepatic lipid accumulation and significantly increased obesity-related glucose metabolic disorders and chronic inflammation.

### More Beige-to-white transition of IngWAT was observed in SAKO mice

Obesity mainly occurs due to an imbalance in energy homeostasis, such as high energy intake and low energy expenditure[1]. We then examined the metabolic rate in SAKO mice using an indirect calorimetry approach according to the methods previously described[27]. The results showed no significant difference in activity between the SAKO and Flox mice (Fig. 3a). However, oxygen consumption ($VO_2$), carbon dioxide production ($VCO_2$), and energy expenditure (EE) were lower in SAKO mice than in controls (Fig. 3b–d). To examine whether *Slc35d3* knockout reduced the rate of mitochondrial respiration, we also determined the $O_2$ consumption rate (OCR) and found that inguinal adipocytes from SAKO mice exhibited a lower OCR than controls (Fig. 3e). To further examine whether Slc35d3 was necessary for cold tolerance, SAKO and Flox mice were exposed to a 4 °C cold chamber for 4 h followed by a 10-min recovery at room temperature and then their rectal temperature was monitored. The SAKO mice maintained a lower core body temperature than the controls during the 4 h cold stimulation and room temperature recovery phases (Fig. 3f). These data demonstrated that the metabolic rate of SAKO mice was markedly lower than that of controls.

As energy-consuming tissues, both IngWAT browning and BAT activation can increase thermogenesis to consume energy, in which beige cells play an important role in metabolism and thermogenesis[1,10]. Reduced energy expenditure may suggest the suppression of WAT browning and/or BAT activation in SAKO mice. RT-qPCR results showed that the mRNA expression levels of thermogenesis genes, including *Ucp1*, *Prdm16*, and *Pgc1-α*, were decreased in the IngWAT of SAKO mice, as well as beige cell markers (*Tbx1* and *Tmem26*) and mitochondrial genes (*Cpt1a*, *Cpt1b*, and *Cpt2*) (Fig. 3g–i). Similarly, immunoblot analysis also revealed significantly lower expression of

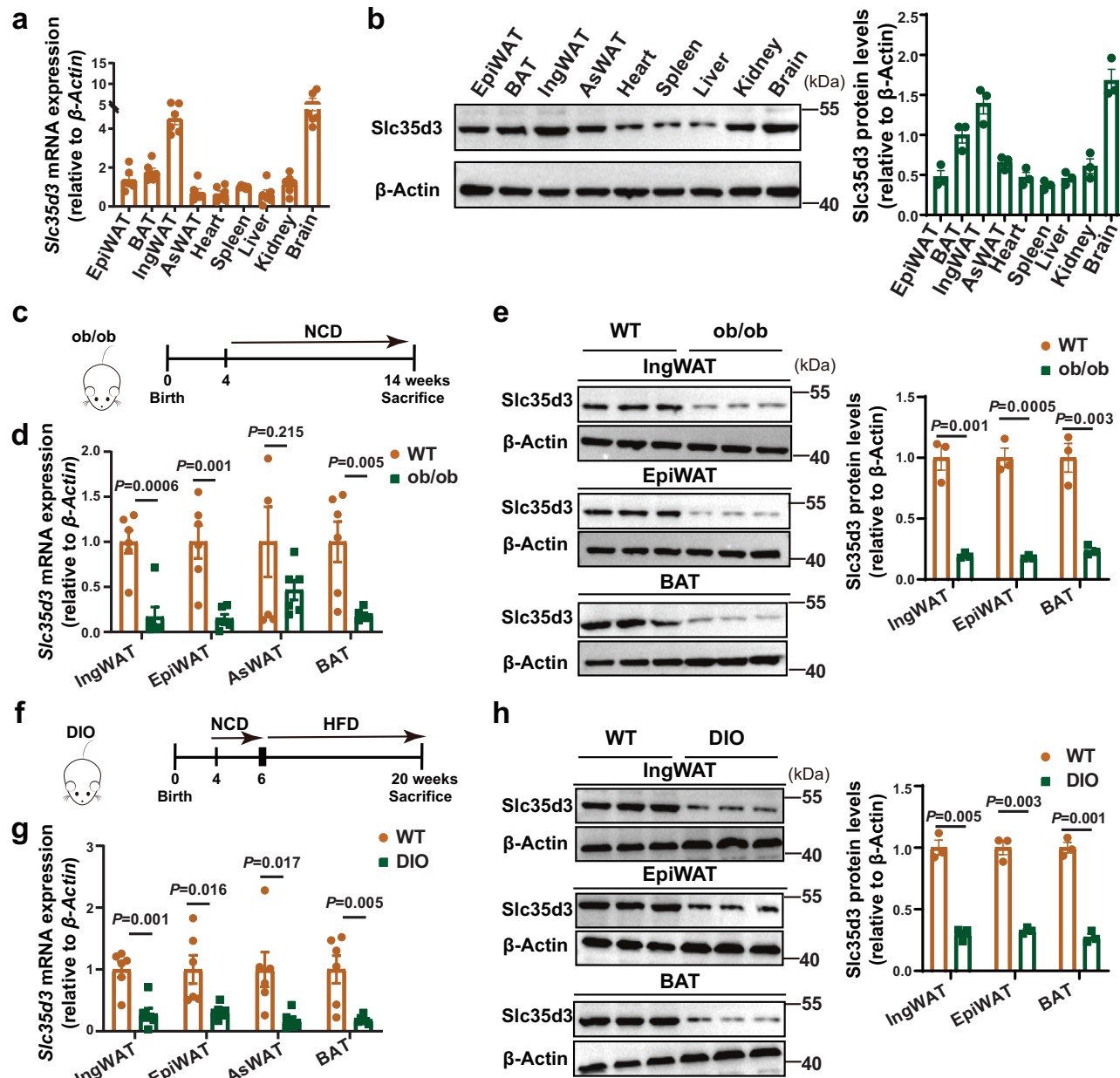

**Fig. 1 | Slc35d3 expression was lower in the fat tissues of obese mice. a** RT-qPCR analysis of *Slc35d3* levels in EpiWAT, BAT, IngWAT, AsWAT, heart, spleen, liver, kidney, and brain from WT (*n* = 6) mice. **b** Representative western blot of Slc35d3 and β-Actin protein in EpiWAT, BAT, IngWAT, AsWAT, heart, spleen, liver, kidney, and brain from WT (*n* = 3) mice. The summary of the quantifications of three independent experiments is shown in the right panel. **c** Schematic diagram of ob/ob mouse treatment. **d** RT-qPCR analysis of *Slc35d3* levels in IngWAT, EpiWAT, AsWAT and BAT from ob/ob (*n* = 6) and WT (*n* = 6) mice. **e** Representative western blot of Slc35d3 and β-Actin protein in IngWAT, EpiWAT, and BAT from ob/ob (*n* = 3) and WT mice (*n* = 3). Right panel, the summary of quantifications of three independent experiments. **f** DIO mice were fed an NCD from 4 to 6 weeks of age and a HFD from 6 to 20 weeks of age. **g** RT-qPCR analysis of *Slc35d3* levels in IngWAT, EpiWAT, AsWAT, and BAT from DIO (*n* = 6) and WT (*n* = 6) mice. **h** Representative western blot of Slc35d3 and β-Actin protein in IngWAT, EpiWAT, and BAT from DIO (*n* = 3) and WT (*n* = 3) mice. Right panel, summary of quantifications of three independent experiments. Values are the fold induction of gene expression levels normalized to the housekeeping gene *β-Actin* (**a**, **b**, **d**, **e**, **g**, **h**), and data are expressed as the mean ± SEM. Statistical significance was assessed by unpaired two-sided Student's *t* test (**d**, **e**, **g**, **h**). The exact *P* values are shown in the figure. EpiWAT epididymal white adipose tissue, BAT brown adipose tissue, IngWAT inguinal white adipose tissue, AsWAT anterior-subcutaneous white adipose tissue, NCD normal chow diet, HFD high-fat diet. Source data are provided as a Source Data file.

Ucp1 and Pgc1-α in SAKO mouse IngWAT (Fig. 3j). Less positive UCP1 staining in the IngWAT of SAKO mice further suggested decreased IngWAT browning (Fig. 3k). In addition, adipose tissue weight was higher and adipocytes were larger in SAKO mice (Figs. 2g and 3l). Given the correlation between lipolysis and metabolic homeostasis, we also examined the expression levels of lipolysis-associated genes (*Atgl*, also called *Pnpla2*, and *Hsl*, also called *Lipe*) by qPCR, which were decreased in the IngWAT of SAKO mice (Fig. 3m). We also observed similar phenotypes in the EpiWAT of SAKO mice

(Supplementary Fig. 2a–d). However, the color of BAT from SAKO mice was consistent with that from Flox mice, indicating a similar lipid content. H&E staining further confirmed that the multilocular lipid droplets of BAT from the SAKO mice were similar to those of the controls (Supplementary Fig. 2e). In addition, no differences in the mRNA or protein levels of Ucp1 were observed in classical BAT (Supplementary Fig. 2f–h).

In summary, these results suggested that adipose tissue-specific knockout of *Slc35d3* impaired browning of WAT.

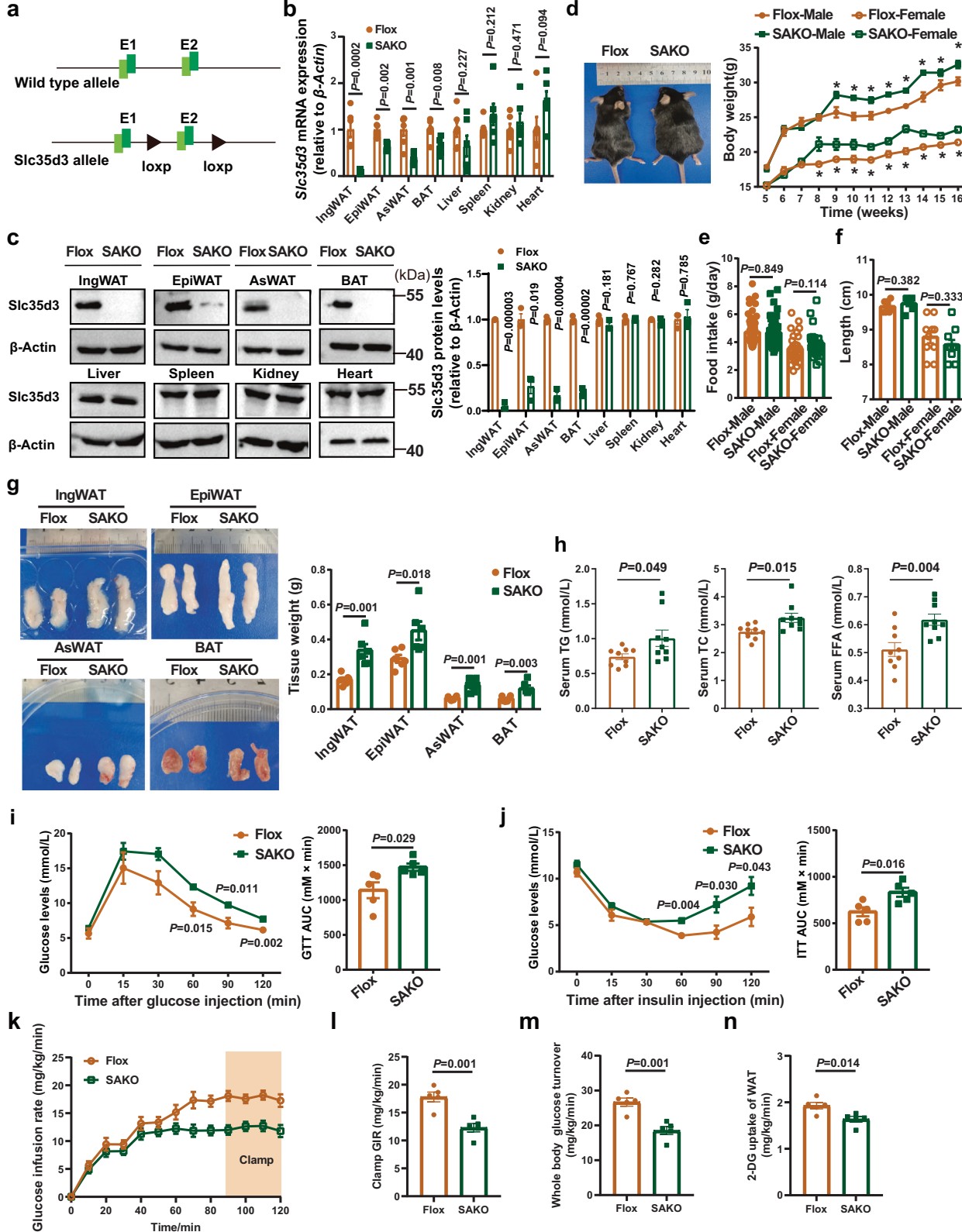

## Adipose-specific *Slc35d3* knock-in protected mice from HFD-induced obesity

To further assess the roles of adipocyte SLC35D3 in the regulation of obesity and adipose inflammation, in "reverse" experiments, we generated adipose-specific *Slc35d3*-knock-in (SAKI) mice by crossbreeding *Slc35d3*-WT mice carrying the LSL-Slc35d3 transgene locus with *adiponectin*-derived Cre mice. Mice with the *Slc35d3* alleles, which did not

express Cre recombinase (*Slc35d3*-WT; *Adiponectin*-Cre[−/−]), were used as controls and hereafter referred to as WT mice (Fig. 4a). Specific overexpression of *Slc35d3* in IngWAT, EpiWAT and BAT was validated by qPCR and immunoblot analysis (Fig. 4b, c). When fed an NCD (Supplementary Fig. 3a), compared to WT littermates, the body weight was lower in SAKI mice (Supplementary Fig. 3b, c), whereas no marked change was observed in their body lengths (Supplementary Fig. 3d).

**Fig. 2 | Adipose-specific *Slc35d3* knockout induced obesity in mice. a** Schematic diagram showing the generation of *Slc35d3* flox mice. **b** qPCR analysis of *Slc35d3* expression in different tissues from SAKO ($n = 6$) and Flox ($n = 6$) mice. **c** Western blot results of Slc35d3 expression in different tissues from SAKO ($n = 3$) and Flox ($n = 3$) mice. Right panel, summary of the quantification of three independent experiments. **d** Body weight curve of SAKO and Flox mice ($n = 11$ per group). $P = 0.011, 0.0003, 0.004, 0.00005, 0.0002, 0.0001, 0.036, 0.006$ for male SAKO mice and controls from 9 to 16 weeks; $P = 0.010, 0.018, 0.021, 0.0003, 0.007,$ $0.00008, 0.0004, 0.001, 0.004$ for female SAKO mice and controls from 8 to 16 weeks. **e** Food intake of SAKO (male, $n = 44$; female, $n = 30$) and Flox (male, $n = 44$; female, $n = 30$) mice on an NCD. **f** Body length of SAKO mice and Flox mice (male, $n = 7$ per group; female, $n = 10$ per group). **g** Representative images and quantification of fat pad weights ($n = 6$ per group). **h** Serum lipid profiles (TG, TC, and FFA) in SAKO ($n = 9$) mice and controls ($n = 9$). **i**, **j** Blood glucose concentrations

during IP-GTT (**i**) and IP-ITT (**j**) in SAKO mice and their controls ($n = 5$ per group). The area under the curve (AUC) values are shown. **k**–**n** Glucose infusion rates (**k**, **l**), whole-body insulin-stimulated glucose turnover (**m**) and 2-deoxyglucose (2-DG) uptake in IngWAT (**n**) during hyperinsulinemic–euglycemic clamps of NCD-fed SAKO ($n = 5$) and Flox ($n = 5$) mice. Values are the fold induction of gene expression normalized to the housekeeping gene β-Actin (**b**, **c**). Data are expressed as the mean ± SEM. Statistical significance was assessed by unpaired two-sided Student's *t* test (**b**, **c**, **e**–**h**, **l**–**n**) or two-way ANOVA followed by Bonferroni's multiple comparisons test (**d**, **i**, **j**). The exact *P* values are shown in the figure or corresponding legends; *$P < 0.05$. EpiWAT epididymal white adipose tissue, BAT brown adipose tissue, IngWAT inguinal white adipose tissue, AsWAT anterior-subcutaneous white adipose tissue, SAKO *Slc35d3* adipocyte-specific knockout, TG triglyceride, TC total cholesterol, FFA free fatty acid. Source data are provided as a Source Data file.

Consistently, when fed a high-fat diet (HFD) (Fig. 4d), the SAKI mice also gained less weight and were leaner than WT mice (Fig. 4e), while body lengths did not change significantly (Fig. 4f), indicating that there was no growth defect in SAKI mice. In addition, fat pad weight and serum TG and TC levels were lower in SAKI mice fed either an NCD or HFD (Fig. 4g, h and Supplementary Fig. 3e, f). Compared to controls, SAKI mice showed a significant improvement in glucose tolerance when fed an NCD (Supplementary Fig. 3g), whereas insulin sensitivity was similar to that of their controls (Supplementary Fig. 3h). In contrast, both glucose tolerance and insulin sensitivity were significantly improved in SAKI mice fed a HFD (Fig. 4i, j). We further performed hyperinsulinemic–euglycemic clamps using male SAKI mice and control mice. The results showed that the glucose infusion rate required to maintain euglycemia in SAKI mice was significantly higher than that in control mice (Fig. 4k, l). In addition, whole-body insulin-stimulated glucose turnover and 2-deoxyglucose uptake in the IngWAT of SAKI mice were also significantly increased (Fig. 4m, n). Summarizing the above results, *Slc35d3* knock-in in adipose tissues improved metabolism in mice.

Adipose tissue macrophages accumulate in large quantities in obese WAT, leading to the development of chronic tissue inflammation[28]. The expression levels of *F4/80*, a marker of macrophages, and chronic inflammation-related genes were markedly decreased in the EpiWAT of SAKI mice fed either an NCD or HFD (Supplementary Fig. 4a–d, i–l). The liver weight of SAKI mice was also lower than that of WT mice (Supplementary Fig. 4e, m). We then investigated whether Slc35d3 reduced lipid accumulation in the liver. The H&E staining results showed fewer lipid-containing vacuoles in the livers of SAKI mice (Supplementary Fig. 4f, n). Furthermore, hepatic TG and TC levels and FFA contents were lower in SAKI mice (Supplementary Fig. 4g, o).

Taken together, these data indicated that *Slc35d3* knock-in protected mice from obesity and significantly improved obesity-related glucose intolerance and chronic inflammation, which was more marked in HFD-fed mice.

### Browning of WAT was increased in SAKI mice

To further investigate the role of adipocyte *Slc35d3* in energy homeostasis, we first examined the food intake and energy expenditure of SAKI mice fed either a HFD or NCD. No significant differences were found in food intake and activity between the SAKI and WT mice fed either diet (Fig. 5a, b and Supplementary Fig. 5a, b). Then, we examined the catabolic rates of SAKI and WT mice using an indirect calorimetry approach. Both HFD- and NCD-fed SAKI mice showed higher rates of $VO_2$, $VCO_2$ and EE than their controls (Fig. 5c–e and Supplementary Fig. 5c–e). In addition, a higher OCR was exhibited in IngWAT adipocytes of SAKI mice (Supplementary Fig. 5f). The cold tolerance test showed that SAKI mice maintained their core body temperature better than controls (Fig. 5g and Supplementary Fig. 5g). To further investigate the regulatory role of

Slc35d3 in the browning of IngWAT, the expression of browning markers, beige cell markers and mitochondrial genes in the IngWAT of SAKI mice was detected. Higher thermogenesis gene expression was found in SAKI mice (Fig. 5h–j and Supplementary Fig. 5h–j). Both immunoblotting and immunohistochemical staining showed higher Ucp1 levels in the IngWAT of SAKI mice (Fig. 5k, l and Supplementary Fig. 5k, l). These results demonstrated that body energy metabolism in SAKI mice was improved.

In obesity, adipose tissue enlarges to store excess energy, which is mainly driven by adipocyte hypertrophy[28]. Next, we determined the effect of *Slc35d3* knock-in on adipocyte architecture. Lower IngWAT weight and smaller adipocytes were found in SAKI mice (Figs. 4g and 5m and Supplementary Figs. 3e and 5m). Higher expression of lipolysis-associated genes was found in SAKI mice (Fig. 5n and Supplementary Fig. 5n), which suggested that the inhibition of adipocyte hypertrophy in SAKI mice might be due to increased lipolysis. Similar adipocyte sizes of EpiWAT were observed in both HFD- and NCD-fed SAKI mice (Supplementary Fig. 6a, b).

Taken together, these results showed that SAKI mice consumed more energy and were protected against obesity due to increased browning of IngWAT.

### Slc35d3 inhibited the activation of Notch1 signaling in mice

We next intended to investigate the potential mechanism by which Slc35d3 regulates IngWAT browning. We isolated IngWAT tissues from SAKI mice for transcriptome sequencing analysis. Both Kyoto Encyclopedia of Genes and Genomes (KEGG) pathway enrichment analysis and Gene set Enrichment Analysis (GSEA) revealed that Notch signaling was markedly inhibited in IngWAT from SAKI mice (Fig. 6a, b). The qPCR results also suggested that the expression levels of Notch1 signaling-related genes were significantly downregulated in the IngWAT of SAKI mice but markedly upregulated in the IngWAT of SAKO mice (Fig. 6c, e), which was further confirmed by western blotting assays (Fig. 6d, f).

The above results implied that Slc35d3 might negatively regulate Notch1 signaling. Given that Slc35d3 was expressed at relatively low levels in the liver (Fig. 1a, b), we examined whether the alterations in adipose tissue Slc35d3 could affect the levels of Notch1 signaling in the liver. The results showed that Notch1 signaling was not significantly altered in the livers of either SAKO or SAKI mice (Supplementary Figs. 1h and 4h, p).

Inhibition of Notch1 signaling may be caused by decreased receptor levels resulting from decreased de novo expression, increased degradation, and impaired transport leading to decreased insertion into the plasma membrane[29]. We speculated that the downregulation of Notch1 signaling might be due to the inhibition of Notch1 transcription or translation levels by Slc35d3 in mice. We then determined the NOTCH1 expression levels at different time points in HEK293T cells overexpressing SLC35D3. Intriguingly, no difference in *NOTCH1* mRNA levels was found after 36 h of SLC35D3 overexpression,

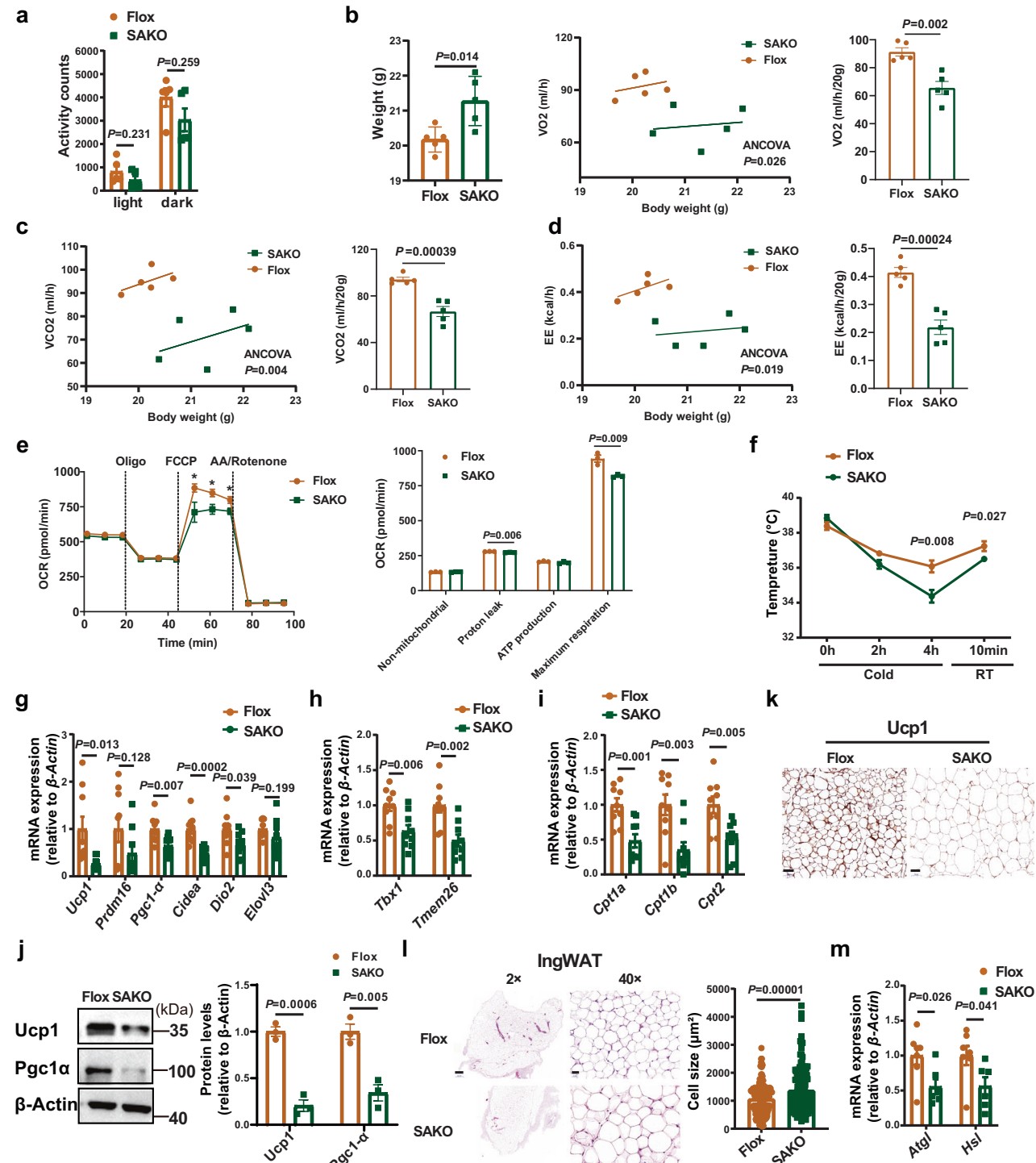

**Fig. 3 | Beige to white transition of IngWAT in SAKO mice. a** Activity of SAKO ($n = 5$) and Flox ($n = 5$) mice. **b–d** Oxygen consumption (**b**), carbon dioxide production (**c**), energy expenditure (**d**) of SAKO ($n = 5$) mice and controls ($n = 5$) were assessed over 24 h by ANCOVA (analysis of covariance). **e** The various components of oxygen consumption rates of mature adipocytes from SAKO ($n = 3$) mice and their controls ($n = 3$). The asterisks in the figure represent the exact $P$ values in the order: $P = 0.038$, $P = 0.023$, $P = 0.033$. **f** Rectal temperatures of SAKO ($n = 6$) mice and their Flox littermates ($n = 5$), which were recorded during cold stimulation for 4 h and returned to RT for 10 min. **g–i** qPCR detection results of thermogenesis gene (**g**), beige adipocyte marker (**h**) and mitochondria-related gene (**i**) expression in IngWAT from SAKO ($n = 9$) mice and controls ($n = 9$). **j** Western blot analysis of thermogenesis protein expression in IngWAT from SAKO ($n = 3$) mice and controls ($n = 3$). Right panel, summary of the quantification of three independent experiments. **k** Immunostaining for Ucp1 in the IngWAT of SAKO and Flox mice.

Representative images from three independent experiments are shown. Scale bar = 50 μm. **l** Hematoxylin and eosin (HE) staining and adipocyte cell size measurements of IngWAT. Representative HE images from three independent experiments are shown. Right panel, the quantitative results of the size of each adipocyte counted (Flox: $n = 162$ cells; SAKO: $n = 144$ cells). Scale bar, 500 μm (left); 20 μm (right). **m** qPCR analysis of the expression of lipolysis genes in IngWAT from SAKO mice ($n = 6$) and their Flox littermates ($n = 7$). Values are the fold induction of gene expression normalized to the housekeeping gene β-Actin (**g–j, m**). Data are expressed as the mean ± SEM. Statistical significance was assessed by unpaired two-sided Student's $t$ test (**a, e, f, g–j, m**) or two-way ANOVA (**b–d**). The exact $P$ values are shown in the figure or corresponding legends; *$P < 0.05$. SAKO *Slc35d3* adipocyte-specific knockout, IngWAT inguinal white adipose tissue, RT room temperature. Source data are provided as a Source Data file.

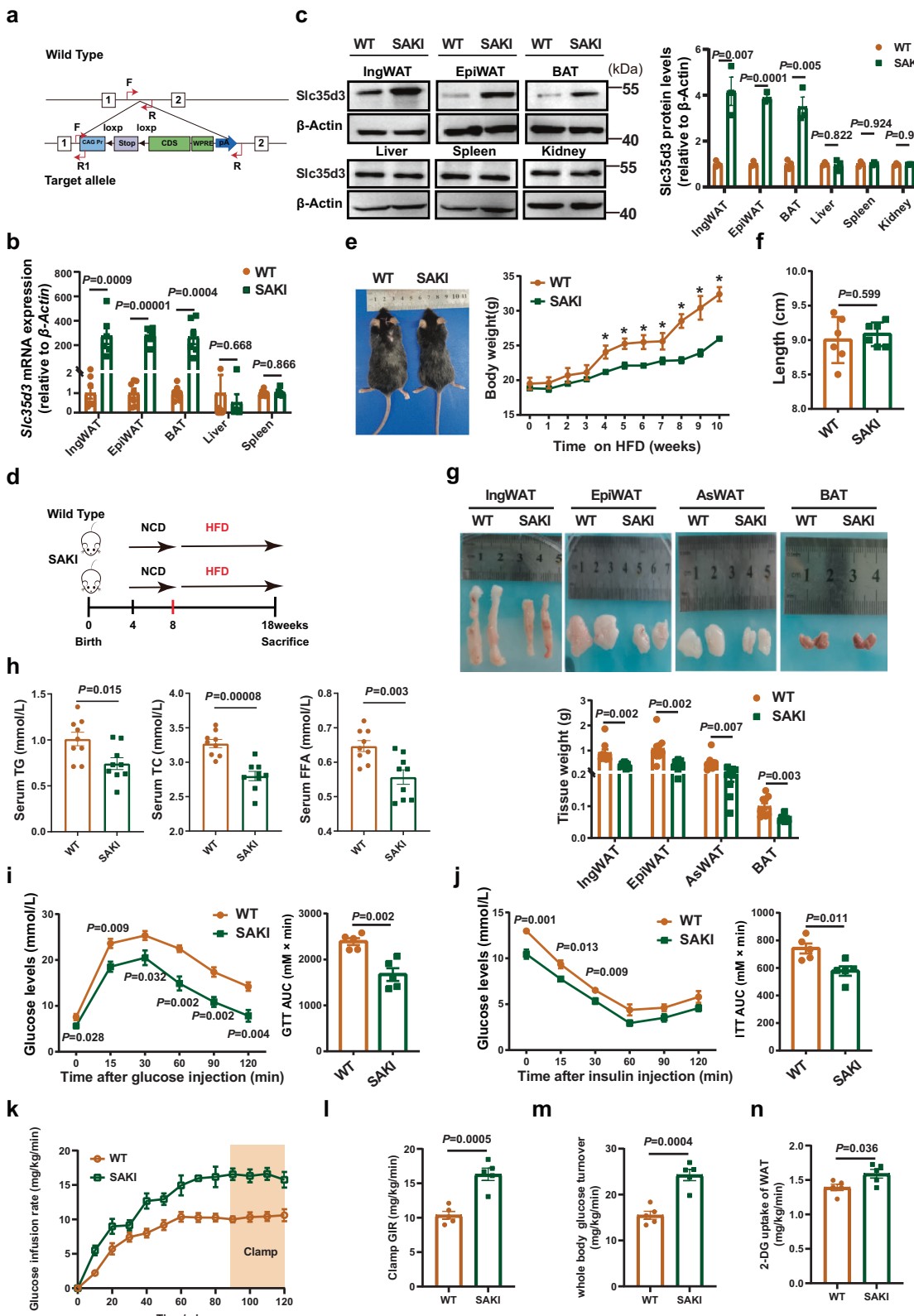

but a significant decrease was observed after 48 h (Fig. 6g). The results of the protein level assays were also interesting, showing that the protein levels of NOTCH1 did not change after either 36 h or 48 h but decreased significantly after 60 h (Fig. 6h). These results showed that reduced NOTCH1 signaling was not caused by the inhibition of SLC35D3 on NOTCH1 transcription or translation levels. Next, we determined whether SLC35D3 altered the half-life of the NOTCH1

protein in HEK293T cells treated with cycloheximide (CHX). The results showed that SLC35D3 did not change the protein degradation of NOTCH1 (Fig. 6i).

Together, these results suggested that the inhibition of NOTCH1 signaling by SLC35D3 was not caused by inhibition of NOTCH1 transcription or translation levels or increased protein degradation.

**Fig. 4 | Adipose-specific *Slc35d3* knock-in protected mice from HFD-induced obesity. a** Construction strategy of SAKI mice. **b** qPCR analysis of *Slc35d3* expression in different tissues from SAKI ($n = 6$) and WT ($n = 7$) mice. **c** Western blot analysis of Slc35d3 expression in different tissues from SAKI ($n = 3$) and WT ($n = 3$) mice. Right panel, summary of the quantification of three independent experiments. **d** SAKI and WT mice were fed an NCD from 4 to 8 weeks of age and then fed a HFD from 8 to 18 weeks. **e** Body weight curve of SAKI ($n = 11$) and WT ($n = 11$) mice fed a HFD ($P = 0.009, 0.002, 0.004, 0.041, 0.00004, 0.002, 0.000008$ for HFD-fed SAKI mice and controls from 4 to 10 weeks). **f** Body lengths of SAKI ($n = 6$) mice and WT mice ($n = 6$). **g** Representative images of fat pads from SAKI ($n = 9$) mice and controls ($n = 9$), as well as the tissue weights. **h** Serum lipid profiles (TG, TC and FFA) in SAKI ($n = 9$) mice and WT mice ($n = 9$). **i, j** Blood glucose concentrations during IP-GTT (**i**) and IP-ITT (**j**) in SAKI mice and their WT littermates; $n = 5$ per group. The area under the curve (AUC) values for GTT assays and ITT assays are shown.

**k**–**n** Glucose infusion rates (**k**, **l**), whole-body insulin-stimulated glucose turnover (**m**) and 2-deoxyglucose uptake in IngWAT (**n**) during hyperinsulinemic–euglycemic clamps of HFD-fed SAKI ($n = 5$) and WT ($n = 5$) mice. Values are the fold induction of gene expression normalized to the housekeeping gene β-Actin (**b**, **c**). Data are expressed as the mean ± SEM. Statistical significance was assessed by unpaired two-sided Student's $t$ test (**b**, **c**, **f**–**h**, **l**–**n**) or two-way ANOVA followed by Bonferroni's multiple comparisons test (**e**, **i**, **j**). The exact $P$ values are shown in the figure or corresponding legends; *$P < 0.05$. EpiWAT epididymal white adipose tissue, BAT brown adipose tissue, IngWAT inguinal white adipose tissue, AsWAT anterior-subcutaneous white adipose tissue, TG triglyceride, TC total cholesterol, FFA free fatty acid, NCD normal chow diet, HFD high-fat diet, SAKI *Slc35d3* adipocyte-specific knock-in. Source data are provided as a Source Data file.

## SLC35D3 increased the ER accumulation of NOTCH1 via interaction with the NOTCH1 extracellular domain

The above results further aroused our curiosity about the exact pathway through which SLC35D3 inhibited NOTHC1 signaling. Next, we examined the localization of SLC35D3 and NOTCH1 by immunofluorescence in HEK293T cells overexpressing SLC35D3. The results were very intriguing, and we found that SLC35D3 and NOTCH1 colocalized intracellularly (Supplementary Fig. 7a), suggesting a possible interaction between SLC35D3 and NOTCH1. Subsequently, we detected the interaction between SLC35D3 and NOTCH1 in HEK293T cells via a coimmunoprecipitation (CoIP) assay. As shown in Fig. 7a, NOTCH1 was coimmunoprecipitated with SLC35D3 in HEK293T cells (Fig. 7a). NOTCH1 proteins consist of a large extracellular domain (ECD), a transmembrane region, and a large NICD[30,31]. Furthermore, we determined the specific domain to which SLC35D3 binds. First, the binding between SLC35D3 and NICD was determined in HEK293T cells transfected with both the SLC35D3 plasmid and NICD plasmid via CoIP assays. As shown in Supplementary Fig. 7b, no coprecipitation of NICD with SLC35D3 was observed (Supplementary Fig. 7b). We then determined the binding between SLC35D3 and ECD, and myc-tagged ECD was coimmunoprecipitated with SLC35D3 in HEK293T cells (Fig. 7b). The direct physical interaction between SLC35D3 and the NOTCH1 ECD was further confirmed by an in vitro pull-down assay (Fig. 7c). Interestingly, SLC35D3 lacking the last 30 amino acids (SLC35D3-30aa) remained immunoprecipitated with the NOTCH1 ECD (Supplementary Fig. 7c). We subsequently transfected HEK293T cells with both the SLC35D3 plasmid and ECD plasmid, as well as the SLC35D3 plasmid and NICD plasmid. Multiple fluorescence staining results showed that SLC35D3 and ECD colocalized (Supplementary Fig. 7d); in contrast, NICD did not colocalize with SLC35D3 (Supplementary Fig. 7e). Taken together, these results revealed that SLC35D3 interacted with NOTCH1 through the ECD, the extracellular domain of NOTCH1.

We next sought to investigate how the interaction of SLC35D3 with NOTCH1 affected NOTCH1 signaling. In signal-receiving cells, NOTCH1 receptors are first generated in the ER and then transported to the Golgi apparatus where NOTCH1 receptors are cleaved into heterodimers (S1 cleavage) and transported to the cell membrane. Transport of NOTCH1 in the endoplasmic reticulum (ER) has been shown to play a critical role in the regulation of NOTCH1 levels at the cell membrane[29,31,32]. Considering that SLC35D3 also localizes to the ER intracellularly[21], we speculated that SLC35D3 regulated the trafficking of NOTCH1 from the ER to the plasma membrane. Then, we isolated membrane and cytoplasmic proteins from 293T cells overexpressing SLC35D3, and western blotting results showed that SLC35D3 overexpression significantly decreased NOTCH1 levels at the plasma membrane but increased NOTCH1 levels in the cytosol (Supplementary Fig. 7f, g). The flow cytometry assay results further confirmed that SLC35D3 overexpression significantly reduced NOTCH1 levels at the plasma membrane (Supplementary Fig. 7h, i). These results suggested that SLC35D3 might regulate NOTCH1 trafficking. To further

investigate the modulatory effect of SLC35D3 on NOTCH1, HEK293T cells were transfected with both SLC35D3 plasmid and GFP-tagged NOTCH1 plasmid, and cells transfected with both empty plasmid and GFP-tagged NOTCH1 plasmid were used as controls. Forty-eight hours after transfection, we labeled the ER in living cells using ER Tracker Red Dyes and detected NOTCH1 localization in living cells by a High Content Imaging System (Opera Phenix, PerkinElmer). Intriguingly, NOTCH1 did not show extensive colocalization with the ER in control cells; instead, NOTCH1 was more distributed at the periphery of the ER, surrounding the cell. However, in cells overexpressing SLC35D3, there was extensive colocalization of NOTCH1 with the ER, indicating that overexpression of SLC35D3 promoted NOTCH1 colocalization with the ER (Supplementary Fig. 7j).

The above results suggested that SLC35D3 might inhibit NOTCH1 signaling by promoting ER accumulation of NOTCH1. To confirm whether SLC35D3 actually regulated NOTCH1 trafficking, we performed multiple fluorescent staining in fixed cells using an Opal multiplex staining protocol with an Opal™ 7 color kit. HEK293T cells were transfected with SLC35D3 plasmid and NOTCH1 plasmid, and we then examined the subcellular localization of SLC35D3 and NOTCH1 in cells 48 h post transfection. The results showed that less NOTCH1 and ER colocalization was observed in control cells, while the overexpression of SLC35D3 promoted NOTCH1 and ER colocalization (Fig. 7d, e). We further transfected HEK293T cells with siRNA targeting SLC35D3 to mediate the knockdown of SLC35D3 and then examined the intracellular localization of NOTCH1. Immunofluorescence results showed that SLC35D3 knockdown reduced NOTCH1 retention in the ER, as shown by decreased NOTCH1 colocalization with the ER in SLC35D3-knockdown cells (Supplementary Fig. 8a, b). We have demonstrated that SLC35D3 interacts with the ECD of NOTCH1, but not with the NICD, in the above results. Interestingly, the fluorescence staining results showed that SLC35D3 and ECD colocalized and both colocalized with ER (Supplementary Fig. 7d). This further confirmed the interaction between SLC35D3 and the ECD, which was mainly in the ER.

To further confirm that SLC35D3 promoted NOTCH1 accumulation in the ER and decreased NOTCH1 levels at the cell membrane, we performed immunofluorescence staining for IngWAT from SAKO mice and SAKI mice. Tissue sections were not permeabilized to better label proteins at the cell membrane surface and reduce the effect of protein staining in the cytoplasm. Fluorescence results showed that NOTCH1 levels at the cell membrane were significantly increased in adipose tissues from SAKO mice compared to control mice (Fig. 7f, g). In addition, the adipocyte sizes in SAKO mice were visibly larger than those in control mice, which was consistent with our previous results (Figs. 3l and 7f). However, NOTCH1 levels at the cell membrane were significantly lower in adipose tissues from SAKI mice than in those from control mice (Supplementary Fig. 8c, d). We also isolated cell membrane proteins and endoplasmic reticulum proteins using a Plasma Membrane Protein Isolation and Cell Fractionation Kit and an

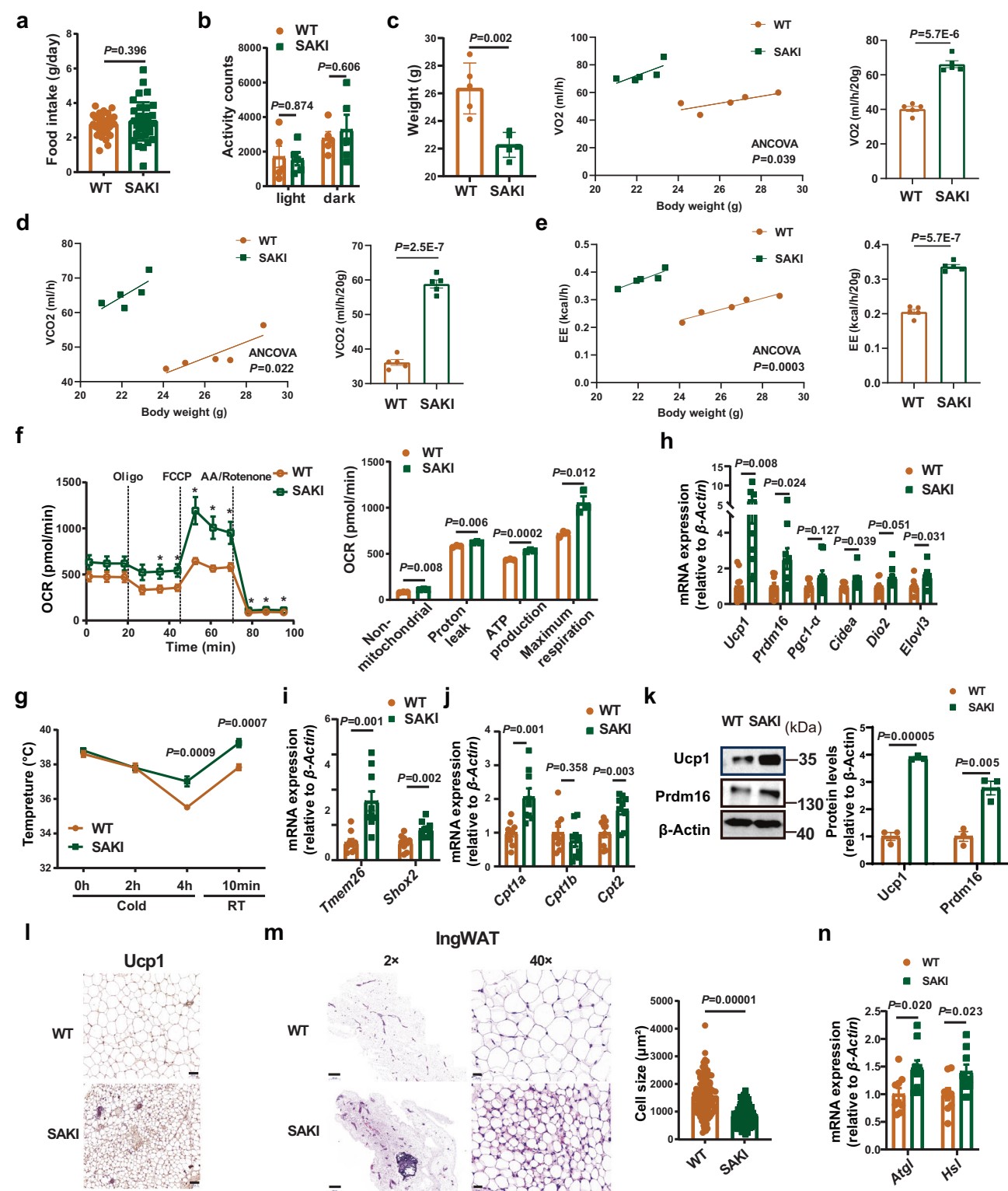

Endoplasmic Reticulum Isolation Kit and detected NOTCH1 levels by western blotting assays. HEK293T cells were transfected with SLC35D3 plasmid and siRNA targeting SLC35D3 and harvested at different time points to detect NOTCH1 protein levels. The results were intriguing, showing that overexpression of SLC35D3 did not alter total NOTCH1 protein levels at 48 h (Supplementary Fig. 9a, b). However, SLC35D3 overexpression significantly promoted the retention of NOTCH1 in the ER, accompanied by a decrease in NOTCH1 protein levels on the cell membrane at 40 h and 48 h (Supplementary Fig. 9a–d). In contrast, the knockdown of SLC35D3 promoted trafficking of the NOTCH1 protein

from the ER to the cell membrane, as shown by decreased NOTCH1 levels in the ER (Supplementary Fig. 9e–h). Interestingly, although NOTCH1 levels on the cell membrane did not change much at 40 h, the knockdown of SLC35D3 significantly increased NOTCH1 levels on the cell membrane over time, and total NOTCH1 levels also increased at 72 h (Supplementary Fig. 9i, j). This finding was also consistent with previous results showing that total NOTCH1 levels were altered after prolonged exposure (Fig. 6h). Subsequently, we also isolated the cellular fractions of IngWAT from SAKI mice and SAKO mice for detection, and the results showed that NOTCH1 levels on the cell membrane

**Fig. 5 | Browning phenotype of IngWAT in SAKI mice. a** Food intake of HFD-fed SAKI ($n = 35$) mice and controls ($n = 35$). **b** Activity of SAKI ($n = 5$) mice and controls ($n = 5$). **c–e** Oxygen consumption (**c**), carbon dioxide production (**d**) and energy expenditure (**e**) of SAKI ($n = 5$) mice and controls ($n = 5$) were assessed by ANCOVA (analysis of covariance) over 24 h. **f** The various components of oxygen consumption rates of mature adipocytes from SAKI ($n = 3$) and WT ($n = 3$) mice. The asterisks in the figure represent the exact $P$ values in the order: $P = 0.039$, $P = 0.024$, $P = 0.002$, $P = 0.002$, $P = 0.009$, $P = 0.002$, $P = 0.0007$, $P = 0.0007$. **g** Rectal temperatures of SAKI ($n = 6$) mice and WT mice ($n = 6$). **h–j** The expression levels of thermogenesis genes (**h**), beige adipocyte markers (**i**), and mitochondria-related genes (**j**) in the IngWAT of SAKI ($n = 9$) mice and controls ($n = 9$). **k** Western blot analysis of thermogenesis protein expression in IngWAT from SAKI ($n = 3$) mice and controls ($n = 3$). Right panel, summary of the quantification of three independent experiments. **l** Immunostaining for Ucp1 in the IngWAT of SAKI mice and controls.

Representative images from three independent experiments are shown. Scale bar = 50 μm. **m** HE staining and adipocyte cell size measurements of IngWAT from SAKI mice and WT mice. Representative HE images from three independent experiments are shown. Right panel, the quantitative results of the size of each adipocyte counted (WT: $n = 108$ cells; SAKI: $n = 106$ cells). Scale bar, 500 μm (left); 20 μm (right). **n** The expression levels of lipolysis genes in IngWAT from SAKI ($n = 9$) mice and WT mice ($n = 9$). Values are the fold induction of gene expression normalized to the housekeeping gene β-Actin (**h–k, n**). Data are expressed as the mean ± SEM. Statistical significance was assessed by unpaired two-sided Student's $t$ test (**a, b, f–k, m, n**) or two-way ANCOVA (**c–e**). The exact $P$ values are shown in the figure or corresponding legends; *$P < 0.05$. SAKI *Slc35d3* adipocyte-specific knock-in, IngWAT inguinal white adipose tissue, RT room temperature. Source data are provided as a Source Data file.

were significantly higher in adipose tissues of SAKO mice than in control mice, accompanied by decreased NOTCH1 levels in the ER (Fig. 7h, i). The opposite results were observed in SAKI mice, as shown by decreased NOTCH1 levels on the cell membrane (Fig. 7h, j). Summarizing the above results, SLC35D3 regulated NOTCH1 trafficking from the ER to the cell membrane.

To further confirm that SLC35D3 affected NOTCH1 levels on the cell membrane, we employed cell surface biotinylation assays to detect NOTCH1 protein on the cell membrane. We first incubated 293T cells and mouse primary adipocytes with PBS and biotin (Sulfo-NHS-SS-biotin), respectively, and detected NOTCH1 proteins by western blotting. The results showed that NOTCH1 proteins on the cell membrane could indeed be labeled by biotin (Supplementary Fig. 9k). Next, we performed cell surface biotinylation assays with 293T cells at 48 h and 72 h after transfection and the results showed that SLC35D3 overexpression significantly decreased NOTCH1 levels on the cell membrane, whereas SLC35D3 knockdown caused an increase in NOTCH1 levels on the cell membrane (Supplementary Fig. 9l–n). Primary adipocytes were tested similarly. Notch1 levels on the cell membranes in primary adipocytes from SAKO mice were significantly higher than those in primary adipocytes from control mice, and the opposite results were observed in adipocytes from SAKI mice (Fig. 7k–m).

Summarizing all the results above, SLC35D3 increased ER accumulation of NOTCH1 and thereby inhibited NOTCH1 signaling, which was achieved by interacting with the NOTCH1 extracellular domain.

### Notch1 knockdown in IngWAT showed a superior rescue effect in SAKO mice compared with SAKI mice

The above results revealed that SLC35D3 regulated obesity and fat metabolism through the NOTCH1 pathway. Next, we sought to investigate the effect of intervening in Notch1 on mouse phenotypes in both SAKO and SAKI mice. We first knocked down Notch1 in the IngWAT of SAKO mice, which was achieved by orthotopic injection of AAV8-adiponectin-shNotch1 (Fig. 8a). The qPCR results showed that Notch1 levels were significantly downregulated in the IngWAT of SAKO mice, but were not changed in other tissues (Fig. 8b). Intriguingly, knockdown of Notch1 resulted in a significant decrease in mouse body weight, accompanied by improved glucose tolerance and insulin sensitivity (Fig. 8c–e). Serum TG and TC levels also decreased significantly (Fig. 8f, g). We subsequently examined the expression levels of relevant genes involved in fat metabolism in IngWAT and found that knockdown of Notch1 caused a significant upregulation of lipolytic genes (*Atgl, Hsl*) and thermogenic genes (*Ucp1, Prdm16, Pgc1-α*) (Fig. 8h, i). The HE staining results of IngWAT further showed that knockdown of Notch1 resulted in a marked reduction in adipocyte size (Fig. 8j).

We have revealed that SAKI mice showed promising resistance to HFD-induced obesity progression. To further investigate the effect of Notch1 knockdown in HFD-fed SAKI mice, we also knocked down Notch1 in the IngWAT of SAKI mice (Fig. 8k). Consistently, qPCR results

showed specific knockdown of Notch1 in IngWAT (Fig. 8l). Interestingly, in HFD-fed SAKI mice, while there were no significant differences, a trend toward improvement in body weight was observed, as well as in glucose tolerance, insulin sensitivity and lipid levels (Fig. 8m–q). The expression levels of lipolytic genes (*Atgl, Hsl*) and thermogenic genes (*Ucp1, Prdm16, Pgc1-α*) also showed a tendency to be upregulated, accompanied by a tendency to decrease adipocyte sizes (Fig. 8r–t).

Taken together, knockdown of Notch1 in the IngWAT of SAKO mice showed a significant rescue effect, as shown by resistance to obesity and improved metabolism. In contrast, Notch1 knockdown showed only a slight improvement in adiposity and metabolism in SAKI mice. In SAKI mice, Notch1 signaling was suppressed by overexpressed Slc35d3, which was correlated with the protective efficacy against obesity. This might explain why knockdown of Notch1 significantly improved metabolism in SAKO mice, but the effect was not evident in SAKI mice.

## Discussion

Here, we demonstrated the role of SLC35D3 in regulating obesity and related metabolic disturbances by studying adipocyte-specific *Slc35d3* knockout and knock-in mice. Slc35d3 was expressed at lower levels in the fat tissues of obese mice, and Slc35d3 knockout induced more beige-to-white transition of IngWAT and showed more susceptibility to obesity, impaired glucose tolerance and exacerbated chronic inflammation, while Slc35d3 knock-in promoted browning of WAT and was protected against obesity. Furthermore, we found that SLC35D3 increased ER accumulation of NOTCH1 and thus inhibited NOTCH1 signaling, which was achieved by interacting with the NOTCH1 extracellular domain (ECD) and promoting the retention of NOTCH1 in the ER. Adiocyte-specific *Slc35d3* knockout mice showed a more pronounced rescue effect of Notch1 knockdown in IngWAT. All of these findings suggest an interesting function of SLC35D3 in obesity, and low SLC35D3 expression in adipose tissues in obesity might be a potential therapeutic target (Fig. 9).

To date, only a few studies on SLC35D3 have been reported. *Ceacam5, Klk6, Slc35d3, Postn,* and *Muc2* mRNA analysis improves the detection of tumor cells in the lymph nodes of colon cancer patients[33]. SLC35D3 causes platelet dysfunction by participating in the biogenesis of platelet-dense granules and increases autophagic activity in midbrain dopaminergic neurons by enhancing the formation of the BECN1-ATG14-PIK3C3 complex, suggesting that SLC35D3 is a potential regulator of tissue-specific autophagy[24,34,35]. In addition, *SLC35D3* is a candidate gene for obesity-related metabolic syndrome that is involved in metabolic control of the central nervous system by regulating dopamine signaling[21]. Furthermore, silencing of SLC35D3 resulted in increased adipogenic processes in porcine intramuscular adipocytes, suggesting that SLC35D3 is associated with adipocyte differentiation[36]. However, there was no additional information regarding the association of adipose SLC35D3 with obesity. In this

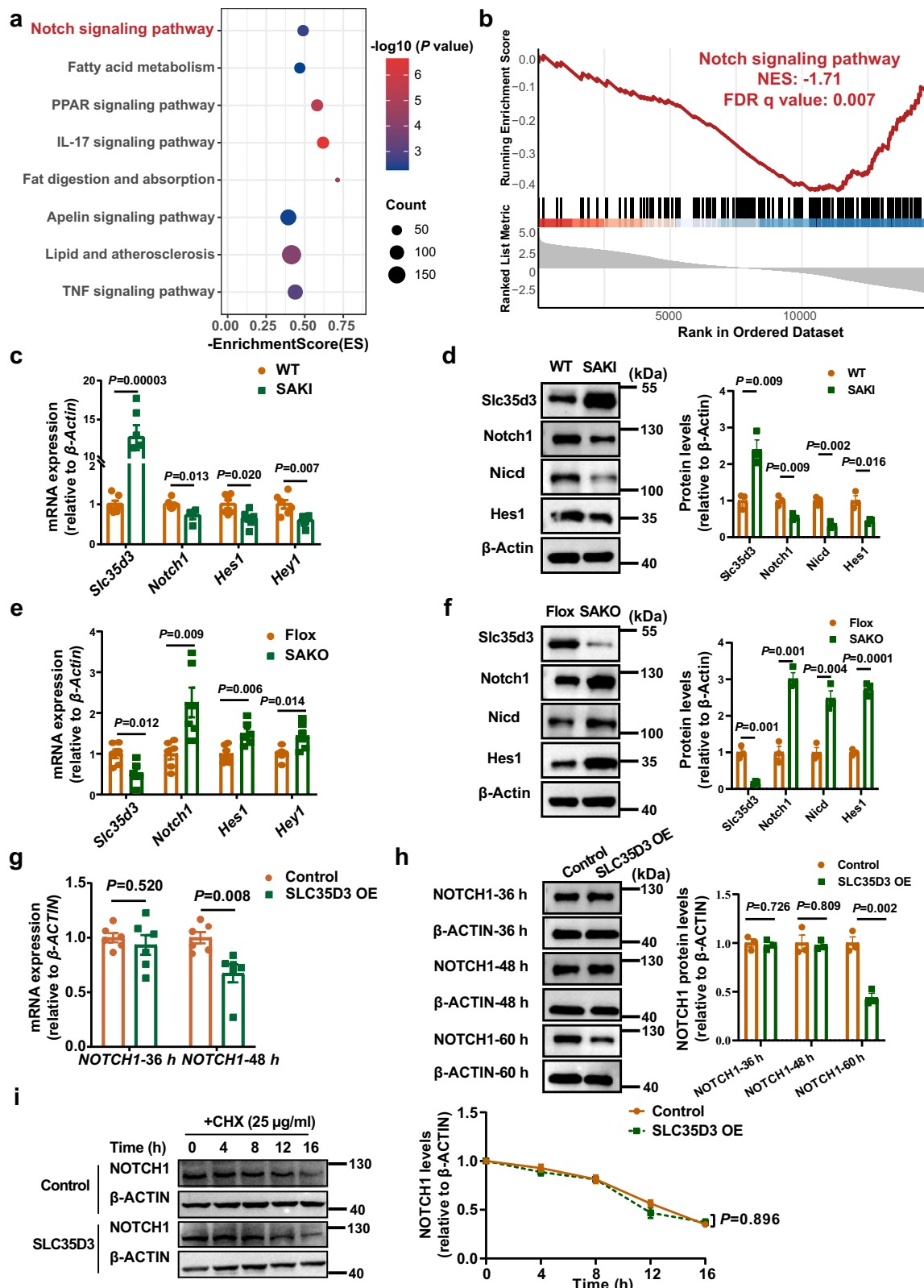

study, we found that Slc35d3 expression was lower in the adipose tissue of obese mice and revealed the molecular mechanism by which Slc35d3 was involved in obesity and related fat metabolism.

Obesity is a metabolic condition usually accompanied by metabolic-related diseases, which are characterized by excessive fat accumulation and are related to WAT dysfunction[37]. Based on the great plasticity of adipose tissue in thermogenesis, activation of BAT and

induction of WAT browning play a critical role in obesity[38,39]. Here, we explored the effect of SLC35D3 in adipose tissue on obesity. *Slc35d3* knockout significantly inhibited adipose tissue browning while *Slc35d3* knock-in significantly increased adipose tissue browning. The white-to-brown fat transition involves a series of cellular processes, including increased Ucp1 expression, mitochondrial production, and lipolysis[10]. These features were all observed in adipose tissues of *Slc35d3* knock-in

**Fig. 6 | SLC35D3 inhibited the activation of NOTCH1 signaling. a** KEGG (Kyoto Encyclopedia of Genes and Genomes) pathway enrichment analysis results of IngWAT from WT and SAKI mice ($n = 3$ per group). $P$ values are determined using the clusterProfiler R package's built-in hypergeometric test, and adjusted $P$ values are calculated using Benjamini–Hochberg correction. **b** GSEA (gene set enrichment analysis) results showed that the Notch signaling pathway was suppressed in the IngWAT of SAKI mice ($n = 3$ per group). The $P$ values in GSEA are calculated empirically with permutation test by permuting the gene labels at random according to the GSEA algorithm, and FDR $q$ values (corrected P values) are calculated by Benjamini–Hochberg FDR correction. The NES and FDR q values are shown in the images. NES, normalized enrichment score; FDR, false positive rate. **c**, **e** qPCR analysis of *Slc35d3* and *Notch1*-related genes in IngWAT from SAKI mice and WT mice (**c**), as well as in IngWAT from SAKO mice and Flox mice (**e**); $n = 6$ per group. **d**, **f** Western blot showing the expression of Notch1-related proteins in

IngWAT from SAKI mice and their controls (**d**), as well as in IngWAT from SAKO mice and their controls (**f**); $n = 3$ per group. **g** qPCR detection of *NOTCH1* expression levels in HEK293T cells overexpressing *SLC35D3*; $n = 6$ per group. **h** Western blot analysis of NOTCH1 protein in HEK293T cells overexpressing *SLC35D3*. Right panel, summary of the quantification of three independent experiments. **i** Western blot analysis of NOTCH1 protein in HEK293T cells treated with CHX (cycloheximide). Right panel, summary of the quantification of three independent experiments. Values are the fold induction of gene expression normalized to the housekeeping gene β-Actin (**c**–**i**). Data are expressed as the mean ± SEM. Statistical significance was assessed by unpaired two-sided Student's $t$ test (**c**–**i**). The exact $P$ values are shown in the figure. SAKI *Slc35d3* adipocyte-specific knock-in, IngWAT inguinal white adipose tissue. Source data are provided as a Source Data file.

mice, which showed considerable resistance to obesity and associated metabolic disorders.

In our study, the Ucp1 expression level was decreased in the WAT but not in the BAT of *Slc35d3* knockout mice. The differential responses of BAT and WAT to *Slc35d3* knockout were perhaps due to the different embryonic origins of these tissues, as well as cell type-specific gene regulatory programs[12]. A higher Ucp1 level is associated with enhanced uncoupled respiration and thermogenesis, which may lead to a higher core body temperature[40,41]. Our results showed that higher Ucp1 levels and enhanced uncoupled respiration were observed in inguinal adipocytes from *Slc35d3* knock-in mice, while Ucp1 levels and uncoupled respiration were inhibited in inguinal adipocytes from *Slc35d3* knockout mice. Meanwhile, core body temperature was higher in *Slc35d3* knock-in mice but lower in *Slc35d3* knockout mice.

Consistently, we also observed a marked improvement in hepatic steatosis in *Slc35d3* knock-in mice. *Slc35d3* knock-in mice displayed a reversal of hepatic triglyceride accumulation, suggesting that other factors contributed to organ-to-organ cross-talk and improvement in liver metabolism. These most likely represented a change in some adipose-secreted hormones that affected hepatic insulin sensitivity[42]. Meanwhile, the expression of inflammatory genes was decreased in *Slc35d3* knock-in mice, which might contribute to ameliorating hepatic steatosis. These results highlighted the physiological significance of adipose-specific *Slc35d3* function and its systemic effects on liver function and whole-body glucose metabolism.

The expression and secretion of proinflammatory cytokines are known to be inhibited by Notch inhibition[43]. Notch signaling is one of the pathways widely acting on the adult maintenance of a variety of tissues and cell types[16,44]. Previous studies showed that inhibition of Notch signaling was associated with ameliorated obesity[10,45–47]. We proposed that the decreased expression level of Notch1 was the primary molecular mechanism underlying the physiological function of Slc35d3 in regulating obesity in mice. SLC35D3 is an orphan nucleotide sugar transporter or a fringe connection-like protein with 10 transmembrane domains. NOTCH1 includes an ECD, a transmembrane domain, and an NICD. In our study, we confirmed that a direct interaction existed between SLC35D3 and the ECD rather than the NICD. Receptor-ligand binding initiates two sequential proteolytic cleavages, resulting in the release of the NICD[48]. Subsequently, NICD binds to the Rbpjk transcription complex and activates Notch target genes, including Hes and Hey family genes. Our results showed that the levels of Nicd were decreased in the IngWAT of *Slc35d3* knock-in mice but increased in the IngWAT of *Slc35d3* knockout mice. As expected, the expression levels of the Notch1 target genes, Hes1 and Hey1, were also decreased when Slc35d3 was overexpressed.

NOTCH signaling is closely associated with adipocyte browning, energy metabolism and obesity progression. Overexpression of Notch in adipocytes was demonstrated to promote severe insulin resistance in mice[49]. It has also been confirmed that

activation of Notch1 inhibits adipocyte browning and adipsin secretion[50], whereas inactivation of Notch1 can lead to higher energy expenditure, improved insulin sensitivity and greater resistance to obesity in mice[10]. In addition, transit out from the ER has been shown to be a critical step in regulating Notch1 receptor levels at the cell membrane[29]. Similarly, in our study, our data showed that SLC35D3 interacted with the NOTCH1 ECD and then retained NOTCH1 in the ER, thereby reducing the levels of NOTCH1 on the plasma membrane, which meant that SLC35D3 inhibited NOTCH1 signaling by inhibiting trafficking of NOTCH1. These data also suggested that the improved metabolic consequences observed in *Slc35d3* knock-in mice could result from inhibited Notch1 signaling. We also investigated the effect of Notch1 knockdown in IngWAT on the phenotypes of adipose tissue-specific *Slc35d3* knockout and knock-in mice. As expected, knockdown of Notch1 resulted in resistance to obesity as well as improved metabolism, which was more pronounced in *Slc35d3* knockout mice.

In conclusion, we identify that SLC35D3 is involved in obesity by interacting with the NOTCH1 extracellular domain and promoting the accumulation of NOTCH1 in the ER, which can lead to the inhibition of NOTCH1 signaling. Adipocyte-specific *Slc35d3* knock-in promotes browning of white adipose tissues and ameliorates obesity, while *Slc35d3* knockout can cause the opposite phenotypes. In addition, knockdown of Notch1 shows considerable improvement in obesity and glucose and lipid metabolism, which is more pronounced in *Slc35d3* knockout mice. Therefore, low SLC35D3 expression in adipose tissues might be a potential therapeutic target of obesity, and NOTCH1 inhibition could lead to greater gains.

## Methods

The research methods applied in this study followed the guidelines of the World Medical Association's Declaration of Helsinki and subsequent revisions, and the study was reviewed and approved by the ethics committee of Fuwai Hospital.

### Animals

All procedures involving mice were reviewed and approved by the ethics committee of Fuwai Hospital. All animal use and welfare adhered to the National Institutes of Health's "Guide for the Care and Use of Laboratory Animals" following the protocols reviewed and approved by the State Key Laboratory of Cardiovascular Disease, National Center for Cardiovascular Diseases, Fuwai Hospital (Beijing, China; number of permit: 0000869). All animals were housed in standard cages in a temperature- and humidity-controlled environment on a 12-h light/dark cycle (temperature: $23 \pm 1\,°C$; relative humidity: 50–60%) with free access to water. Both male and female mice aged 2–4 months were used for experiments unless otherwise indicated. The numbers of mice are indicated in the figure legends for each experiment. Mice were euthanized by $CO_2$ inhalation at the

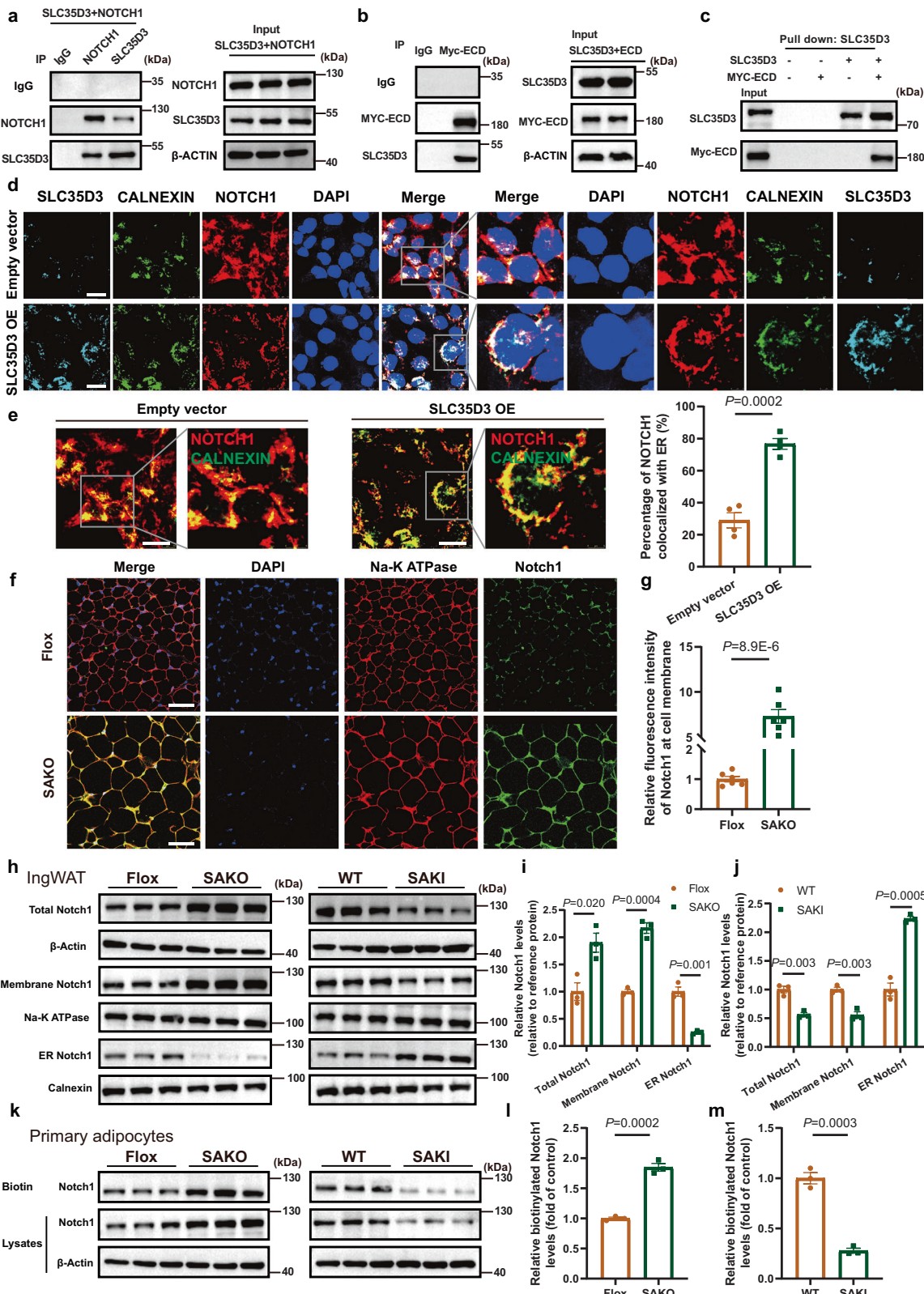

appropriate time during the study and tissue samples were removed for further experiments.

Slc35d3-Flox mice and Slc35d3-WT mice carrying additional LSL-Slc35d3 transgenes in the C57BL/6 background were generated at Beijing Biocytogen Incorporated Company. Adiponectin-Cre mice and ob/ob mice were purchased from Cyagen Biosciences (Suzhou, China). The targeting strategies of adipose-specific Slc35d3 knockout

and knock-in mice are shown in Figs. 2a and 4a, respectively. For HFD-induced obesity, 8-week-old mice were divided randomly and fed either an irradiated normal chow diet (NCD, 10 kcal% fat, D09100304, Research Diets, Inc) or irradiated high-fat diet (HFD, 60 kcal% fat, D12492, Research Diets, Inc) until the end of the experiment. Body weight was monitored every week. For adeno-associated virus (AAV) injections, 4-week-old male mice were

**Fig. 7 | SLC35D3 increased the ER accumulation of NOTCH1 by interacting with the NOTCH1 extracellular domain. a, b** CoIP assay results showed the interaction between SLC35D3 and NOTCH1 (**a**), as well as the interaction between SLC35D3 and ECD (**b**). Representative images from three independent experiments are shown. **c** Direct interaction between SLC35D3 and ECD was further confirmed by pull-down assay. Representative images from three independent experiments are shown. **d, e** SLC35D3 overexpression promoted the colocalization of NOTCH1 and ER. Representative confocal images of cells are shown (**d**), as well as the quantification results of four independent parallel experiments (**e**). The inset in the right panel shows a magnified view of the indicated area. Scale bar, 20 μm. CALNEXIN, an ER marker protein. **f, g** SAKO mice showed higher levels of Notch1 at the cell membrane in IngWAT than control mice. Representative confocal images are shown (**f**), as well as the quantification results of six independent parallel experiments (**g**). Scale bar, 50 μm. Na-K ATPase, a plasma membrane marker protein. **h–j** Membrane proteins and endoplasmic reticulum proteins of IngWAT were isolated from SAKO and SAKI mice (*n* = 3 per group). The western blot results are shown (**h**), as well as the quantitative analysis results (**i, j**). Total NOTCH1 expression was normalized to β-Actin expression, membrane NOTCH1 expression was normalized to Na-K ATPase, and ER NOTCH1 expression was normalized to CALNEXIN. **k–m** Primary adipocytes were isolated from the IngWAT of SAKO mice and SAKI mice (*n* = 3 per group). Cell surface biotinylation assay results are shown (**k**), as well as the quantitative analysis results (**l, m**). Data are expressed as the mean ± SEM. Statistical significance was assessed by unpaired two-sided Student's *t* test (**e, g, i, j, l, m**). The exact *P* values are shown in the figure. SAKO *Slc35d3* adipocyte-specific knockout, SAKI *Slc35d3* adipocyte-specific knock-in, OE overexpression, IngWAT inguinal white adipose tissue, ER endoplasmic reticulum. Source data are provided as a Source Data file.

injected orthotopically with $2.0 \times 10^{11}$ vg of AAV8-adiponectin-shNotch1 virus or control AAV8-adiponectin-shCtrl virus at multiple sites in the inguinal white adipose tissue. Four weeks after injection, the mice were prepared for analysis.

### Food intake and indirect calorimetry study
Food intake was calculated by measuring food consumption and the number of days. Oxygen consumption ($VO_2$), carbon dioxide production ($VCO_2$) and energy expenditure (EE) were measured using an indirect calorimetry system (Oxylet, PanLab, Spain) installed under a constant environmental temperature (22 °C) and a 12-h light/12-h dark cycle. Mice in each chamber had free access to food and water.

### Rectal temperature measurement
A digital thermometer (KW689) was used in combination with a copper thermocouple probe (FT3400 Type). The probe was inserted 1.6 cm into the anal ducts of male and female adult mice (2–4 months old).

### Hyperinsulinemic–euglycemic clamp
The hyperinsulinemic–euglycemic clamp was performed as previously described[51,52]. Briefly, five-month-old male mice were surgically operated to establish an indwelling catheter in the jugular vein seven days prior to the clamp test. The mice were subsequently allowed to recover for 6 days until the experiment started. On the day of the experiment, mice were fasted overnight (-15 h) and a 2-h hyperinsulinemic–euglycemic clamp was then performed in conscious mice with a primed-continuous human insulin infusion (100 mU/kg/min within 2 min, followed by 5 mU/kg/min), while the 20% glucose infusion was administered at a variable rate to maintain steady-state blood glucose. In our study, steady-state blood glucose was defined as a blood glucose level of $130.0 \pm 10$ mg/dL. To assess whole-body glucose turnover, glucose was continuously infused throughout the experiment. To determine insulin-stimulated glucose uptake in individual organs, 2-deoxyglucose was injected intravenously at 75 min from the start of the experiment. Tissues were dissected immediately after the clamp test was completed and were then stored in liquid nitrogen for further biochemical analysis.

### Blood glucose measurements
For glucose tolerance tests, mice were given an intraperitoneal injection of 100 mg/ml D-glucose (2 g/kg body weight) after overnight fasting for 16 h, and tail vein blood glucose concentrations were then monitored. For insulin tolerance tests, mice were fasted for 4 h from 9:00 am before intraperitoneal administration of human insulin (0.75 U/kg body weight; MCE, HY-P0035), and tail vein blood glucose concentrations were then monitored. Tail vein blood was taken at 0, 15, 30, 60, 90, and 120 min after injection, and blood glucose levels were measured using an Accu-Chek Active Blood Glucose Meter and corresponding test strips (Roche).

### Isolation, culture, and differentiation of primary adipocytes
Stromal vascular fraction (SVF) cells were isolated as previously described with modifications[10,53]. Briefly, adipose tissue was dissected, minced, and digested with 2 mg/ml Type I collagenase (Worthington, LS004196) for 1 h at 37 °C in a water bath under slight agitation. The digestion was stopped with centrifugation buffer (PBS containing 10% FBS), and then the tissue was filtered through 100-μm filters and centrifuged at 300×g for 5 min. Filters were washed with 10 ml of centrifugation buffer and the sample was centrifuged as described above. The SVF pellet from the initial centrifugation was resuspended in 2 ml of erythrocyte lysis buffer and incubated for 2 min at room temperature. Following the addition of 2 mL of centrifuge buffer, the samples were filtered through a 40-μm cell strainer. Filters were rinsed with 8 mL of centrifugation buffer, and samples were centrifuged at 300×g for 5 min. Then the SVF cells were resuspended and cultured in growth medium containing DMEM-F12 (Gibco, 10565018) with 20% FBS (Sigma, TMS-013-B), 10 ng/ml FGF (PeproTech, 450-33) and 1% penicillin/streptomycin (Gibco, 15140122) at 37 °C with 5% $CO_2$ for 2 days. Preadipocyte differentiation was induced with induction medium containing DMEM-F12, 10% FBS, 0.5 μg/mL insulin (MCE, HY-P0035), 5 μM dexamethasone (Sigma, D1756), 1 μM rosiglitazone (Sigma, R2408) and 0.5 mM 3-isobutyl-1-methylxanthine (Sigma, I7018) for 2 days followed by 4 days in differentiation medium containing DMEM-F12, 10% FBS, and 0.5 μg/mL insulin until adipocyte maturation.

### Cell culture
HEK293T cells (ATCC, CRL-3216) were obtained from ATCC and were cultured in DMEM (Gibco) with 10% fetal bovine serum (FBS) and 1% penicillin/streptomycin. Cells were used between passages 8 and 10. The cells were incubated at 37 °C (with 5% $CO_2$) until confluent and then transferred to 6-, 12-, or 24-well plates. Knockdown of SLC35D3 was mediated by siRNA transfection using RNAiMAX (Invitrogen, Rockford, IL) transfection reagent (the sequences of siRNA: sense 5′-CCA UGU ACG UGG UCU UCA ATT-3′; antisense 5′-UUG AAG ACC ACG UAC AUG GTT-3′). Overexpression of SLC35D3 was achieved by His-tagged SLC35D3 plasmid transfection using Lipofectamine 3000 (Invitrogen, Rockford, IL) transfection reagent, and overexpression of different domains of NOTCH1, ECD, and NICD, was achieved by myc-tagged ECD plasmid and myc-tagged NICD plasmid transfection. Transfection was performed when the degree of cell fusion reached 70–80%. After incubation for the appropriate time, the cells were harvested for further experiments.

### Live-cell imaging
HEK293T cells were cultured in black 96-well plates (PerkinElmer) and transfected with SLC35D3 plasmid or empty vector plasmid, along with NOTCH1-GFP plasmid expressing NOTCH1-GFP fusion proteins. Forty-eight hours after transfection, cells were treated with ER Tracker Red Dyes (Invitrogen, E34250) to label ER in living cells according to the manufacturer's instructions, and Hoechst 33342 (Thermo Scientific,

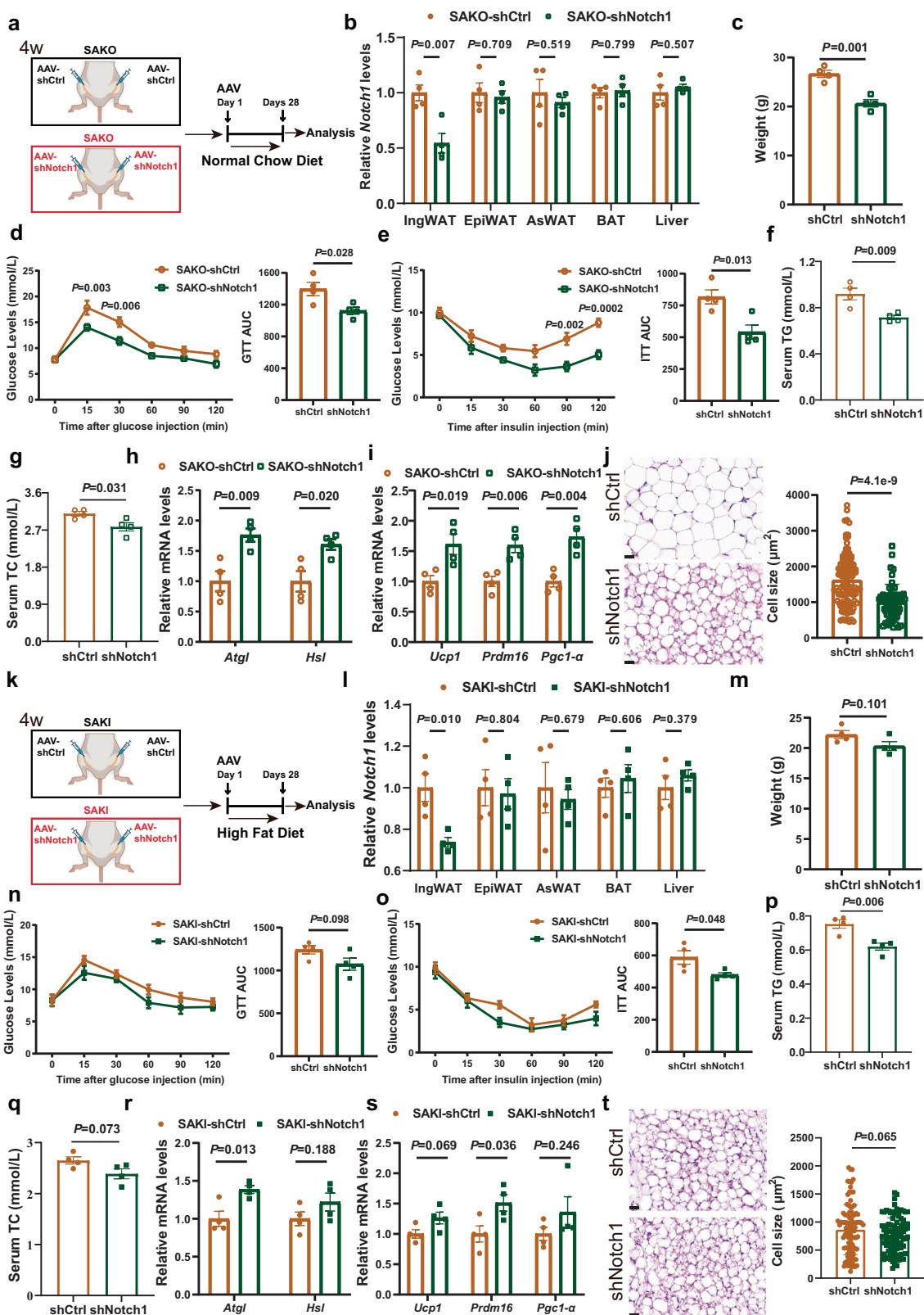

62249) was then used to label cell nuclei. Following labeling, a high-content imaging system (Opera Phenix, PerkinElmer) was used to perform live-cell imaging.

**Cell surface biotinylation assay**

Cells were seeded into 15-cm dishes and harvested for cell surface biotinylation assays at ~80–90% confluency. Cells were washed three times with prechilled PBS and then a Pierce™ Cell Surface Protein Biotinylation and Isolation Kit (Thermo Scientific, A44390) was used. Biotinylation experiments were performed in strict accordance with the manufacturer's instructions, including Sulfo-NHS-SS-Biotin incubation, isolation, and elution of labeled proteins. The separated protein was used for subsequent western blotting assays.

**Fig. 8 | Notch1 knockdown in IngWAT showed a superior rescue effect in SAKO mice compared with SAKI mice. a, k** Illustration of *Notch1* knockdown in the IngWAT of SAKO (**a**) and SAKI (**k**) mice. **b, l** *Notch1* was specifically knocked down in the IngWAT of SAKO (**b**) and SAKI (**l**) mice; *n* = 4 per group. **c, m** Body weights of NCD-fed SAKO mice (**c**) and HFD-fed SAKI mice (**m**); *n* = 4 per group. **d, e, n, o** GTT (**d**) and ITT (**e**) results of NCD-fed SAKO mice (*n* = 4 per group), as well as GTT (**n**) and ITT (**o**) results of HFD-fed SAKI mice (*n* = 4 per group). The area under the curve (AUC) values are shown in the right panel. **f, g, p, q** Serum TG and TC levels in the NCD-fed SAKO mice (**f, g**) and HFD-fed SAKI mice (**p, q**); *n* = 4 per group. **h, r** Lipolytic gene expression levels of IngWAT in the NCD-fed SAKO mice (**h**) and HFD-fed SAKI mice (**r**); *n* = 4 per group. **i, s** Thermogenic gene expression levels of IngWAT in the NCD-fed SAKO mice (**i**) and HFD-fed SAKI mice (**s**); *n* = 4 per group. **j, t** HE staining results and adipocyte size quantification of IngWAT in the NCD-fed

SAKO mice (**j**) and HFD-fed SAKI mice (**t**). Representative images of HE staining results from three independent experiments are shown. Right panel, the quantitative results of the size of each adipocyte counted (**j**: shCtrl, *n* = 104 cells; shNotch1, *n* = 107 cells. **t**: shCtrl, *n* = 102 cells; shNotch1, *n* = 101 cells). Scale bar, 20 μm. Values are the fold induction of gene expression normalized to the housekeeping gene β-Actin (**b, h, i, l, r, s**). Data are expressed as the mean ± SEM. Statistical significance was assessed by unpaired two-sided Student's t test (**b, c, f–k, l, m, p–t**) or two-way ANOVA followed by Bonferroni's multiple comparisons test (**d, e, n, o**). The exact *P* values are shown in the figure. SAKO *Slc35d3* adipocyte-specific knockout, SAKI *Slc35d3* adipocyte-specific knock-in, IngWAT inguinal white adipose tissue, TG triglyceride, TC total cholesterol, NCD normal chow diet, HFD high-fat diet. Source data are provided as a Source Data file.

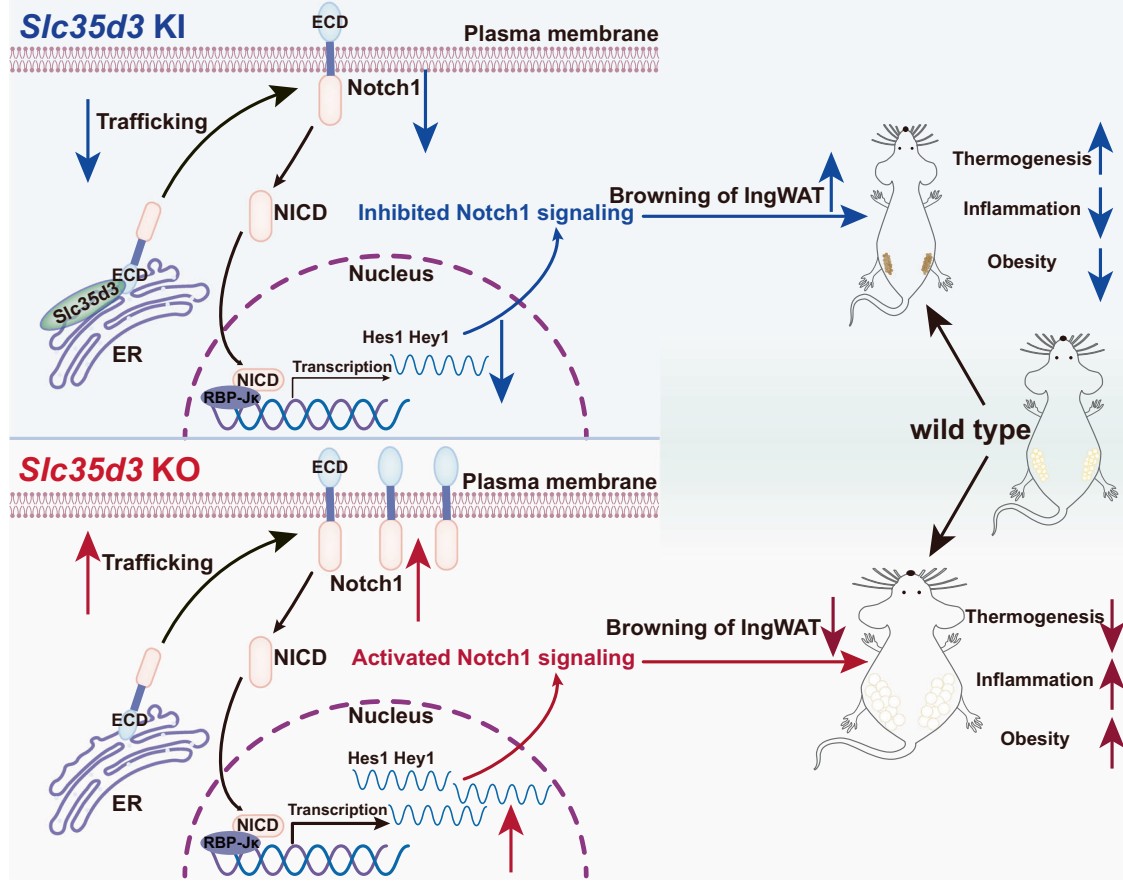

**Fig. 9 | SLC35D3 promotes white adipose tissue browning to ameliorate obesity via NOTCH signaling.** SLC35D3 regulates obesity by interacting with the extracellular domain (ECD) of NOTCH1, which promotes NOTCH1 accumulation in the

endoplasmic reticulum (ER) and inhibits NOTCH1 transport to the cell membrane, ultimately inhibiting NOTCH1 signaling. IngWAT, inguinal white adipose tissue.

## Immunofluorescence staining

For immunofluorescence staining of mouse adipose tissue sections, tissue sections were dewaxed to water following routine procedures. To better label the protein on the cell membrane and reduce the effect of protein staining in the cytoplasm, tissue sections were not permeabilized. Following blocking with goat serum for 1 h, the cells were incubated with Notch1 and Na-K ATPase primary antibodies overnight at 4 °C. Alexa Fluor 488-conjugated and Alexa Fluor 594-conjugated secondary antibodies were then used. DAPI was used to label cell nuclei. Confocal imaging was performed using a Leica SP8 confocal microscope. For all other immunofluorescence staining, the Opal™ 7 color kit (Akoya Biosciences, NEL811001KT) was used to perform multiple fluorescence staining of cells according to the recommended procedures. Both ECD and NICD were stained with antibodies to the fusion-expressed myc proteins, and the other proteins were

fluorescently stained with their own antibodies. The antibodies used are shown in Supplementary Table 1.

## Plasma membrane protein and endoplasmic reticulum protein extraction

For plasma membrane protein and ER protein extraction from cells, cells were harvested at various time points after transfection and washed with ice-cold PBS. For cellular fractionation from mouse adipose tissues, 12-week-old adult male mice were euthanized by $CO_2$ inhalation, and adipose tissues were then isolated. Then, isolation and extraction of plasma membrane proteins were performed by a spin column method using the Plasma Membrane Protein Isolation and Cell Fractionation Kit (Invent Biotechnologies, Inc, SM-005) according to the manufacturer's instructions, while the Endoplasmic Reticulum Isolation Kit (Sigma-Aldrich, ER0100) was

used for isolation and extraction of ER proteins following the recommended procedures.

## Adipocyte OCR measurement

Primary SVF cells from IngWAT were isolated and cultured for 3 days before plating in XF cell culture microplates (Seahorse Bioscience). SVF cells (20,000 cells) were seeded in each well, and each sample had five replicates. After 6 days of differentiation, cultured adipocytes were washed twice in prewarmed XF medium and then transferred to 37 °C without $CO_2$ for 50 min. The OCR was measured by the XF extracellular flux analyzer (Seahorse Biosciences). The chemicals (final concentration: 1.5 μM oligomycin, 2 μM FCCP, and 0.5 μM rotenone & antimycin A) were preloaded into cartridges and injected into XF wells in succession. The OCR was calculated as a function of time (picomoles per minute).

## Quantitative RT-qPCR and RNA-Seq

Total RNA was extracted from tissues using a Trizol reagent. RNA was treated with RNase-free DNase I to remove genomic DNA. The purity and concentration of the total RNA were determined by a spectrophotometer (Nanodrop 2000c, Thermo Fisher) at 260 and 280 nm. The ratios of absorption (260 nm/280 nm) of all samples were between 1.8 and 2.0. To perform the qPCR expression analysis, 1 μg of total RNA was converted to complementary DNA (cDNA). The mRNA expression levels were analyzed by a Prism 7500 sequence-detection system (ABI, Rockford, IL) using Hieff® qPCR SYBR Green Master Mix. The housekeeping gene (*β-Actin*) was used as a reference. Data were expressed as threshold cycle value (Ct) values and used to determine ΔCt values. Fold changes in expression levels were calculated according to the following transformation: fold increase = $2^{-\Delta\Delta Ct}$. For RNA-Seq, total RNA was isolated from three inguinal white adipose tissues of SAKI mice and controls. The library construction and sequencing were performed at Shanghai Biotechnology Corporation (Shanghai, China). The primers used for mRNA expression analysis are shown in Supplementary Table 2.

## Protein extraction and western blot analysis

Protein was isolated from cells or tissues using IP buffer containing proteasome inhibitors. Protein concentrations were determined using Pierce BCA Protein Assay Reagent (Pierce Biotechnology). Total proteins (20 μg) were separated by SDS–PAGE, transferred to a polyvinylidene fluoride (PVDF) membrane (Millipore Corporation), blocked in 5% fat-free milk for 2 h at room temperature and then incubated with primary antibodies overnight at 4 °C. All antibodies used in the study are shown in Supplementary Table 1. The results shown in the figures are representative of results from at least three independent experiments.

## HE staining

Adipose tissues and livers were fixed in 4% PFA (DF0135, Leagene, Beijing, China) for 48 h at room temperature. Then the tissues were embedded in paraffin, cut into 3-μm-thick slices, deparaffinized, and rehydrated using xylene, ethanol, and water by standard methods. For histological analysis of adipose and liver features, 3-μm sections were stained with hematoxylin/eosin to detect histopathological changes and viewed by a Leica DM6000B equipped with a digital camera (Leica, NY). Images shown in figures are representative results of at least three biological replicates. Adipocyte sizes were analyzed by ImageJ software.

## Coimmunoprecipitation (CoIP) assay

HEK293T cells were grown on 10 cm dishes for 48 h after transfection and washed two times with ice-cold PBS. Then, the cells were lysed in IP lysis buffer and incubated on ice for 5 min. The lysates were incubated with appropriate antibodies at 4 °C overnight, followed by protein A/G

Pierce bead incubation at room temperature for 1 h with mixing. After the beads were washed three times with 500 μL washing buffer and washed one time with 500 μL ultrapure water, the target antigens were eluted with 100 μL elution buffer. Proteins were separated by 10% SDS–PAGE and immunoblotted with the corresponding antibodies. The antibodies used are shown in Supplementary Table 1.

## Pull-down assay

For the pull-down assays, the bait protein, purified GST-SLC35D3, was immobilized on glutathione affinity resin and incubated with Myc-ECD protein lysate at 4 °C for at least 1 h with gentle rocking motion on a rotating platform. The prey protein, Myc-ECD, was captured by the bait protein, and then the bait-prey protein was eluted. Myc-ECD and SLC35D3 levels were subsequently analyzed by western blotting. The antibodies used are shown in Supplementary Table 1.

## Flow cytometry

The study was performed as previously described[48]. HEK293T cells were plated in 10-cm plates and transfected with 9 μg of SLC35D3 plasmid per plate using Lipofectamine 3000 transfection reagent. Forty-eight hours post transfection, the cells were fixed using 4% PFA for 10 min at room temperature, centrifuged at 115×$g$ for 10 min, and washed twice. Next, the cells were incubated for 15 min at 4 °C in FcR blocking medium. Then, the cells were incubated with PE-conjugated Notch1 antibody (CST, 15004) at room temperature for 1 h with rotation, and washed twice. Then 250 μL of ice-cold ligand binding buffer was added to each cell pellet, mixed and passed through the filter cap of a 5 mL polystyrene round-bottom tube. This process removed clumped cells immediately prior to flow cytometry. The MFI of the R-phycoerythrin (PE)-labeled antibody emitting fluorescence at 660 nm was determined with a FACS-Calibur (BD Biosciences). Data were acquired by a flow cytometer and analyzed by FlowJo CE (7.5.110.7).

## Statistical analyses

Statistical analyses for data were performed in Graph Pad PRISM 9.0 and SPSS.26.0, and data plotting was performed using Graph Pad PRISM 9.0. The quantitative analysis of immunofluorescence and western blotting results was performed using ImageJ software, and each dot in the quantitative graphs represents a parallel independent biological sample test. The images were created using Adobe Illustrator 2020 software. All replicate experiments (including cell- and mouse-based experiments) were biological replicates, which were repeated at least three times. Data are presented as means ± SEMs and individual data points are plotted. The normality of the data was tested using the Shapiro–Wilk normality test. Data between different groups were compared using unpaired, two-sided Student's $t$ test or two-way ANOVA followed by Bonferroni's multiple comparisons test. A $P$ value < 0.05 was considered statistically significant.

## Reporting summary

Further information on research design is available in the Nature Portfolio Reporting Summary linked to this article.

# Data availability

The data supporting the findings of this study are all available in the manuscript and its supplementary information. The RNA sequencing data for Fig. 6a, b are available at GSE247085. Other data for results and figures in the manuscript are all provided in the Source Data file. Source data are provided with this paper.

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

## Acknowledgements
This study was supported by the Chinese Academy of Medical Sciences Innovation Fund for Medical Sciences with grant 2021-I2M-1-016 and the National Natural Science Foundation of China with grants 81970430 and 81770424. The authors also thank State Key Laboratory of Cardiovascular Disease, Fuwai Hospital, for financial support (Opening and Operation Fund and Basic Research Fund). The corresponding author (Y.W.) received all the above grants.

## Author contributions
Y.W. designed the study and provided financial support for the study. H.W., L.Y., J.W., Yaqing Z., M.X., B.C., and M.Y. carried out the experiments and collected the experimental data. H.W., L.Y., Yu Z., and Y.Y. conducted the animal experiments. H.W., L.Y., J.W., and C.L. analyzed and summarized the experimental data. L.Y. wrote the manuscript. H.W., L.Y., and J.W. revised the manuscript. R.H. and Y.W. supervised the study. Y.W. has full access to all the data in the study.

## Competing interests
The authors declare no competing interests.
