## [Peer Review File · Nature Communications]

SLC35D3 promotes white adipose tissue browning to ameliorate obesity by NOTCH signalingREVIEWERS' COMMENTS:

Reviewer #1 (Remarks to the Author):

In their manuscript, Hongrui Wang and colleagues propose a novel role for SLC35D3 in regulating whole body energy expenditure by promoting browning of white adipocytes in mice. Starting from the observation that SLC35D3 expression is lower in obese mice as opposed to lean ones, the authors show that knockout of SLC35D3 reduces fat depot mass and increases energy expenditure in healthy, as well as obese animals. The authors further claim that SLC35D3 favors white to beige transition, reduces obesity-associated inflammation, and acts as an inhibitor of NOTCH1 signaling. While the finding that SLC35D3 might play a beneficial role in regulation adipocyte function and controlling energy metabolism is highly interesting, the manuscript is unfortunately not convincing. Many statements (too many!) are not sufficiently underpinned by experiments performed, phenotypes are often marginal and the manuscript contains many unclarities and mistakes.

Major comments:

1 Although it is apparent, that SLC35D3 mRNA and protein level decrease in different depots and most dominantly in ingWAT (where it also is most expressed), the western blots really lack quality. β -Actin bands vary strongly between samples. In panel C the 2nd ob/ob IngWAT samples doesn't show a band for SLC35D3, however also β -Actin is strongly reduced compared to the other samples. Similarly, in panel F all IngWAT samples have thinner β -Actin bands compared to WT. For the EpiWAT samples the exposure of β -Actin seems odd – the picture is of very poor quality. Also, in Fig. 1C the BAT blot shows only 5 samples instead of 6 like the others, which makes it unclear to which cohort the samples belong (WT or ob/ob).

2 The tissue specificity of the SLC35D3 KO is unclear. On mRNA level, the authors do document a reduction in white depots, however, especially in Ing- and asWAT, p values are very high. In contrast, differences in SLC35D3 mRNA level in BAT and liver look striking, however, significances have not been indicated. On protein level on the other side, the authors only document a convincing reduction in SLC35D3 in IngWAT. The authors should add statistics to all comparisons in panel B and either revise their statements on knockout

level or provide more convincing blots.

3 The role of the liver in the phenotypes observed by the authors is very controversial. As indicated in point 2, it seems as if the authors observe a decrease in SLC35D3 mRNA expression (F2B). At the same time, the authors state themselves, that similar to the fat depos, liver mass is increased in the KO animals. As also the liver plays a main role in glucose metabolism and whole-body energy homeostasis and, in addition, might be directly affected by the KO, it is unclear if the observed phenotypes are merely mediated by adipose tissue or if the liver could contribute, too. It would certainly be relevant for the paper to properly assess a potential KO of SLC35D3 in liver by qPCR and western blot.

4 The performed experiments shown in Figure 2 and 3 are insufficient to conclude that there is an increase or decrease in inflammation. Both, the changes in macrophage markers, as well as in inflammatory cytokines are rather small. To solidify their statements, the authors should include histological analysis of different WAT to assess potential differences in immune cell infiltration and perform ELISAs to measure cytokine level.

5 Similar to the gene expression analysis shown in F2, also the genes expression analysis shown in F3, 4 and 5 are barely convincing. For most genes, differences are marginal and not significant. The authors should either provide more convincing panels or tune down their statements. It is not valid to say that a certain phenotype occurs, here less browning, if only half of the tested markers show differences.

6 The potential interaction between SLC35D3 and NOTCH is not sufficiently shown by the authors. To indeed prove that there is a real interaction, additional bidirectional pulldown experiments, as well as pulldowns with mutants of the respective interactors would be required. Additionally, many of the already shown experiments require optimization and repetition. It is for example impossible to draw conclusions about the localization of a protein, if the control (e.g. Lamin b1) is not equal between different conditions.

Minor comments:

1 The manuscript contains many mistakes and typos (few of which are listed down below)

and should be revised thoroughly.

Line 48: “are coexistent” or “coexisted”

Line 50: “play” and not “plays”

Line 51: “their” and not “its”

Line 103: typo in “from”

2 What as the age of the mice depicted in panel E. It seems a little confusing, that the datapoints in the Graph shown in E don't match those shown in D. Of course, this might be due to different mice cohorts used, but this should be clarified.

3 Generally, many p values seem very low. For example, the reduction in serum TC level (1l). While it is certainly more challenging to reach high significance values when working with animal models, the authors should tune down some statements (e.g saying that they would observe a trend).

4 In Figure 3A it is not clear why the animals seem more active in the light phase, although they should be night active. Usually, oxygen consumption should rise in the active phases.

5 For all Continuous Graphs (e.g F2D, J and K or F3A, B, C and E) the authors should indicate how many animals were used in the figure legend.

6 Various western blots throughout the whole paper are of extremely poor quality and should be exchanged.

Reviewer #2 (Remarks to the Author):

SLC35D3 is expressed in striatal neurons, and it was recently shown that SLC35D3 is a UDP-glucose transporter present on vesicular membranes. In addition, mice with a recessive mutation in the SLC35D3 gene have reduced motor activity, lower energy expenditure, and develop obesity. In this manuscript, the authors address the hypothesis of whether C expressed in fat cells contributes to the obesity phenotype. To investigate the role of adipocyte-specific SLC35D3, the authors generated both adipocyte-specific SLC35D3-deficient mice and SLC35D3 knockin mice with high adipocyte-specific SLC35D3 expression. The SLC35D3 knockout mice develop obesity and are characterized by glucose intolerance, insulin resistance, decreased thermogenesis, hepatic lipid accumulation, and other metabolic disturbances associated with obesity. On the other hand, SLC35D3 knockin mice are protected from diet-induced obesity and are characterized by an overall healthy

metabolic phenotype. Mechanistically, the authors claim that SLC35D3 inhibits Notch1 signaling. This is consistent with a study showing that adipose-specific inactivation of Notch1 leads to browning of WAT and increased expression of UCP1.

The authors studied an interesting topic, and the phenotypes of the mice are convincing. However, the paper is not very well written, many details are missing and the mechanistic data are not convincing. Please find below my specific comments.

1. Many signaling pathways have been described in the literature that influence thermogenic pathways. A convincing rationale to study Notch1 is not provided.
2. It is hard to imagine that the presented model is correct. SLC35D3 is a transmembrane protein that is located in vesicular membranes. Based on the presented data, the authors proposed a model (Figure 7N) that SLC35D3 is a (secreted?) protein that inhibits the interaction of Notch1 with its ligands. The authors need to show how and in which cell compartment Notch1 can interact with SLC35D3.
3. Figure 1C. For ingWAT and epiWAT the authors showed 3 samples per group. For BAT 5 samples are shown, why?
4. Figure 1H. Except for the asWAT, all adipose tissue depots show high SLC35D3 protein expression. Why there is no protein in asWAT?
5. Figure 1 legend. Schematic diagram does not show the generation of ob/ob or DIO mice. This is one example of many others where the text does not describe the illustrations correctly or in detail. The entire text needs to be carefully checked and re-written, I do not see it as my task as a reviewer to edit the entire text.
6. Figure 2B. The reductions of SLC35D3 protein in epiWAT, asWAT and BAT are not very convincing. In addition, because protein expression (especially including thermogenic-acting proteins such as UCP1) in adipose tissues often exhibit high variability, it is not appropriate to show protein expression with an n of 1. This is true for almost all Western blots shown, also lacking adequate quantification.
7. Figures showing indirect calorimetry. Compared to their controls, SLC35D3 knockout and knockin mice have a different body weight. The indirect calorimetry data are presented per kg body weight, this is not correct and needs to be changed (see for instance PMID: 34489606).
8. Compared to the normal chow diet, high fat diet feeding should increase ectopic lipid accumulation in the liver. This is not visible in the HE stainings.

9. Presented plasma triglyceride levels are ca 0.05 mMol (4 mg/dl). This is impossible, these values are at least 10 times lower than normal.

10. Figure 4K. The glucose values after insulin injection run almost parallel, i.e. different insulin resistance between the genotypes cannot be assumed.

11. Figure 5L, the authors describe that lipolysis-associated genes are higher in knockin mice. Actually, they show the opposite.

12. Figure 6B. Knockin mice have 100-fold induction on the gene expression levels. However, on the protein level I cannot see a difference between wild type and knockin mice.

13. Figures 6H/J. The IMF pictures are not very convincing. For instance, the ECD of Notch1 is located in one dot close to nucleus, which co-localize with SLC35D3. How does fit to the model? The claims described by the authors in the text are not justified by the presented data.

14. The authors used quantitative flow cytometry data of primary adipocytes and HEK cells to show the inhibitory effect of SLC35D3 on Notch1 – ligand interaction. Details on the flow cytometry approach are lacking. Furthermore, Flow cytometry leads to the destruction of sensitive mature adipocytes. Thus, these data must be justified by alternative experimental approaches.

15. Figure legend 6N. Gene expression are normalized to the housekeeper beta-actin? In the figure, the authors show nucleo-cytoplasmatic ratios of proteins.

16. Line 418. If SLC35D3, as proposed by the authors, is an inhibitor of Notch1 signaling, the authors have to do the experiments with the pharmacological inhibitor in SLC35D3 knockin mice (and not in the knockout). In the SLC35D3 knockin mice, the effect by DBZ treatment should be blunted.

Reviewer #3 (Remarks to the Author):

The authors put forth intriguing work on the possible novel functions of the protein SLC35D3 in regulation of obesity in mice, possibly through regulation of Notch signaling. While this is a noteworthy result, there are several significantly worrisome issues throughout the report.

1) the authors would benefit from significant editorial oversight; multiple sentences

throughout the work are incomplete, confusing, or run-on. Furthermore, the constant shift between figures within the paper along with the supplementary data has led to text and factual inconsistencies (ie in describing expression of lipolysis-related genes and figure 5L, they state "higher expression..." when the figure shows the opposite; similarly, the description of 5K says "smaller" when the data shows larger cell size as plotted.) Connected to this and worrisome when considering data analysis is that MCP1 and CCL2 are suggested/inferred to be different proteins, even to the point of different primers used for expansion, when these are actually the same gene product/protein.

2) the work is original, but suffers from a several inconsistencies. While the protein itself is definitely expressed in adipose tissue, it is also expressed in liver. Liver Notch signaling alterations override the adipose effect when Notch is inhibited; further exploration of the effects observed in the liver during g-sec inhibition would therefore be a significant benefit. Additional evidence is therefore needed.

Furthermore, in analyzing metabolic outcomes in the KI and KO mice, food intake and movement should be accounted for. In addition, timing of the variety of tests (GTT/ITT, etc) should be clearly documented; what if the effects on change in GTT and ITT are simply due to the alterations in weight, or is there a direct effect of altering Notch signaling on insulin signaling (ie connections to the Foxo-Notch data)? It would be vital to see if there are changes to the physiology of these mice BEFORE the weight gain.

In the Notch data (figure 6M specifically), further quantification of expression is needed. Only a single data point is shown, and for some reason the WT and flox mice (which should have similar Notch levels) appear to be different. Furthermore, while the protein of interest does appear to bind the extracellular portion of Notch1, could all the data be explained by the down-regulation of Notch1 expression itself?

Finally, throughout the paper, "relative" is used for mRNA, protein quantification, etc. This is confusing, as it is unclear what the number is relative to...all this data would best be explained as fold change from a control

Reviewer #1 (Remarks to the Author):

In their manuscript, Hongrui Wang and colleagues propose a novel role for SLC35D3 in regulating whole body energy expenditure by promoting browning of white adipocytes in mice. Starting from the observation that SLC35D3 expression is lower in obese mice as opposed to lean ones, the authors show that knockout of SLC35D3 reduces fat depot mass and increases energy expenditure in healthy, as well as obese animals. The authors further claim that SLC35D3 favors white to beige transition, reduces obesity-associated inflammation, and acts as an inhibitor of NOTCH1 signaling. While the finding that SLC35D3 might play a beneficial role in regulation adipocyte function and controlling energy metabolism is highly interesting, the manuscript is unfortunately not convincing. Many statements (too many!) are not sufficiently underpinned by experiments performed, phenotypes are often marginal and the manuscript contains many unclarities and mistakes.

Response: Response: Thanks for the helpful comments. According to the comments, we have supplemented more data for supporting the statements, including 3vs.3 more samples in groups with marginal significance and additional experiments. We have also provided better representative immunoblots images and quantification of three independent experiments for all Western blots in the revised manuscript. We have doublechecked the whole manuscript and one native speaker have revised the manuscript. Detailed point-to-point responses were shown as below.

Major comments:

1 Although it is apparent, that SLC35D3 mRNA and protein level decrease in different depots and most dominantly in ingWAT (where it also is most expressed), the western blots really lack quality. β -Actin bands vary strongly between samples. In panel C the 2nd ob/ob IngWAT samples doesn't show a band for SLC35D3, however also β -Actin is strongly reduced compared to the other samples. Similarly, in panel F all IngWAT samples have thinner β -Actin bands compared to WT. For the EpiWAT samples the exposure of β -Actin seems odd – the picture is of very poor quality. Also, in Fig. 1C the BAT blot shows only 5 samples instead of 6 like the others, which makes it unclear to which cohort the samples belong (WT or ob/ob).

Response: Thank the reviewer for the helpful comment. We have repeated the experiment and selected better representative immunoblots images of Figure 1 and quantified three independent experiments. The revised Figure 1 was shown as below:

2 The tissue specificity of the SLC35D3 KO is unclear. On mRNA level, the authors do document a reduction in white deposes, however, especially in Ing- and asWAT, p values are very high. In contrast, differences in SLC35D3 mRNA level in BAT and liver look striking, however, significances have not been indicated. On protein level on the other side, the authors only document a convincing reduction in SLC35D3 in IngWAT. The authors should add statistics to all comparisons in panel B and either revise their statements on knockout level or provide more convincing blots.

Response: Thank the reviewer for the helpful comment. We have added statistics to all comparisons in Figure 2B. In addition, we have selected more representative immunoblots images to confirm the adipose tissue specificity of the SLC35D3 KO in Figure 2C. Our Figure 2 showed that the expression of Slc35d3 was significantly reduced in IngWAT, AsWAT, EpiWAT and BAT, while there was no change in other tissues, including liver, spleen, kidney and heart, indicating the adipose tissue specificity of the Slc35d3 KO. The revised Figures 2B-2C were shown as below:

3 The role of the liver in the phenotypes observed by the authors is very controversial. As indicated in point 2, it seems as if the authors observe a decrease in SLC35D3 mRNA expression (F2B). At the same time, the authors state themselves, that similar to the fat depositions, liver mass is increased in the KO animals. As also the liver plays a main role in glucose metabolism and whole-body energy homeostasis and, in addition, might be directly affected by the KO, it is unclear if the observed phenotypes are merely mediated by adipose tissue or if the liver could contribute, too. It would certainly be relevant for the paper to properly assess a potential KO of SLC35D3 in liver by qPCR and western blot.

Response: Thank the reviewer for the helpful comment. According to the reviewers' comments, we have added statistics to all comparisons in Figure 2B. There was no significant difference in hepatic expression of *Slc35d3* between the SAKO mice and their Flox littermates. It was the same with its protein levels (Figure 2C). The revised Figures 2B-2C were shown as below:

In addition, no changes were observed in Notch1 signaling in the SAKO livers when compared with the controls, indicating that ectopic lipid accumulation resulted in lipotoxic metabolic stress, which promoted metabolic dysfunction in liver. The information was shown as below:

4 The performed experiments shown in Figure 2 and 3 are insufficient to conclude that there is an increase or decrease in inflammation. Both, the changes in macrophage markers, as well as in inflammatory cytokines are rather small. To solidify their statements, the authors should include histological analysis of different WAT to assess potential differences in immune cell infiltration and perform ELISAs to measure cytokine level.

Response: Thank the reviewer for the helpful comment. We have supplemented histological analysis of EpiWAT to assess potential differences in immune cell infiltration in Figures S1A-S1B. We have also determined the expression in adipose tissue by western blot assay. Results showed that *Slc35d3* knockout upregulated the expression of chronic proinflammatory in EpiWAT (Figures S1C-S1D). We put these results in the supplemental file. The revised Figure S1 was shown as below:

Additionally, we have measured the expression of chronic proinflammatory in IngWAT from SAKI when fed an HFD via qPCR. We have also detected the expression levels of chronic proinflammatory factors (IL1 β and IL6) by ELISA assay. Results showed that *Slc35d3* overexpression downregulated the expression of chronic proinflammatory as below:

5 Similar to the gene expression analysis shown in F2, also the genes expression analysis shown in F3, 4 and 5 are barely convincing. For most genes, differences are marginal and not significant. The authors should either provide more convincing panels or tune down their statements. It is not valid to say that a certain phenotype occurs, here less browning, if only half of the tested markers show differences.

Response: For the results with small difference, we have repeated the experiments and increased the number of samples. We have provided more convincing panels in Figures 2-5. The revised Figures were shown as below:

Figure 2. Adipose-specific *Slc35d3* knockout induced obesity in mice.

Figure 3. Beige-to-white transition of IngWAT in SAKO mice.

Figure 4. Adipose-specific *Slc35d3* knockin protected mice from HFD-induced obesity.

Figure 5. Browning phenotype of WAT in SAKI mice.

6 The potential interaction between SLC35D3 and NOTCH is not sufficiently shown by the authors. To indeed prove that there is a real interaction, additional bidirectional pulldown experiments, as well as pulldowns with mutants of the respective interactors would be required. Additionally, many of the already shown experiments require optimization and repetition. It is for example impossible to draw conclusions about the localization of a protein, if the control (e.g. Lamin b1) is not equal between different conditions.

Response: We thank the reviewer for raising this point. According to the *in silico* analysis

(SPIDER), the last 30 amino acids of SLC35D3 protein might be critical to the interaction with the ECD, we have conducted CO-IP assay between truncation mutation of SLC35D3 and NOTCH1 ECD. But the results showed that SLC35D3 lacking the last 30 amino acids was also coimmunoprecipitated with NOTCH1 ECD. Figure S7 was shown as below:

Furthermore, we have conducted a pull-down experiment between SLC35D3 and NOTCH to detect the direct physical interaction. The results showed a direct interaction(Figure 6N). The added Figure 6N was shown as below:

In addition, we have repeated and optimized the experiment about the localization of a protein, the revised Figures 6T-6U were shown as below:

Minor comments:

1 The manuscript contains many mistakes and typos (few of which are listed down below) and should be revised thoroughly.

Line 48: “are coexistent” or “coexisted”

Line 50: “play” and not “plays”

Line 51: “their” and not “its”

Line 103: typo in “from”

Response: Thank the reviewer for these suggestions. All of these have been corrected. We have revised the manuscript thoroughly to obviate mistakes and errors.

2 What as the age of the mice depicted in panel E. It seems a little confusing, that the datapoints in the Graph shown in E don’t match those shown in D. Of course, this might be due to different mice cohorts used, but this should be clarified.

Response: We thank the reviewer for raising this point. We have double-checked and improved the figure legends of Figures 2D-2E. The information was shown as below: (D) Body weight curve of SAKO (n=11) and Flox (n=11) mice both in male and female. (E) Body weight of SAKO (n=6) and Flox (n=6) male mice on NCD in another mice cohort.

3 Generally, many p values seem very low. For example, the reduction in serum TC level (1I). While it is certainly more challenging to reach high significance values when working with animal models, the authors should tune down some statements (e.g saying that they would observe a trend).

Response: We thank the reviewer for raising this point. In order to further verify our results, we added 3V3 samples in each group to detect the serum TG, TC and FFA levels. At the same time, we have deleted some statements (e.g we have deleted the description of results without significant difference). The information was shown as below:

First, serum TG, TC and FFA levels were higher in the SAKO mice (Figure 2I).

Second, serum TG and TC levels were lower in the SAKI mice compared to their controls with either diet (Figures 4I (HFD) and S3F (NCD)).

4 In Figure 3A it is not clear why the animals seem more active in the light phase, although they should be night active. Usually, oxygen consumption should rise in the active phases.

Response: Thanks for the helpful suggestion. The reason why animals are more active in the light stage may be that someone else did experiments at that time, which led to the long time of lighting in their room, resulting in the deviation of results. To verify our results, we have reconducted the metabolic rate of the SAKO mice. The revised results were shown as below:

5 For all Continuous Graphs (e.g F2D, J and K or F3A, B, C and E) the authors should indicate how many animals were used in the figure legend.

Response: Thanks for the helpful suggestion. We have supplemented the number of animals in the figure legends (e.g F2D, J and K or F3A, B, C and E). The revised Figure legends were shown as below:

Figure 2. Adipose-specific Slc35d3 knockout induced obesity in mice. (D) Body weight curve of SAKO (n=11) and Flox (n=11) mice both in male and female. (J) Blood glucose concentrations during IP-GTT in SAKO mice and their controls. GTT area under the curve (AUC) values for SAKO (n=5) and Flox (n=5) NCD mice. (K) Blood glucose concentrations during IP-ITT in SAKO mice and their controls. ITT AUC values for SAKO (n=5) and Flox (n=5) NCD mice.

Figure 3. Beige-to-white transition of IngWAT in SAKO mice. (A) Activity of SAKO (n=5) and Flox (n=5) mice on NCD. (B) Oxygen consumption of SAKO (n=5) mice and their Flox littermates (n=5) fed a NCD. (C) Carbon dioxide production of SAKO (n=5) mice and their Flox littermates (n=5) fed a NCD. (E) The various components of oxygen consumption rates of mature adipocytes from SAKO (n=3) mice and their controls (n=3).

6 Various western blots throughout the whole paper are of extremely poor quality and should be exchanged.

Response: Sorry for these. We have provided better representative WB imaging in the manuscript. The revised Figures were shown as below:

Figure 1. *Slc35d3* expression was lower in fat tissues of obese mice.

Figure 2. Adipose-specific *Slc35d3* knockout induced obesity in mice.

Figure 4. Adipose-specific Slc35d3 knockin protected mice from HFD-induced obesity.

Figure 6. SLC35D3-Notch1 interaction inhibited the binding of Notch 1 receptor to its ligands.

Reviewer #2 (Remarks to the Author):

SLC35D3 is expressed in striatal neurons, and it was recently shown that SLC35D3 is a UDP-glucose transporter present on vesicular membranes. In addition, mice with a recessive mutation in the SLC35D3 gene have reduced motor activity, lower energy expenditure, and develop obesity. In this manuscript, the authors address the hypothesis of whether C expressed in fat cells contributes to the obesity phenotype. To investigate the role of adipocyte-specific SLC35D3, the authors generated both adipocyte-specific SLC35D3-deficient mice and SLC35D3 knockin mice with high adipocyte-specific SLC35D3 expression. The SLC35D3 knockout mice develop obesity and are characterized by glucose intolerance, insulin resistance, decreased thermogenesis, hepatic lipid accumulation, and other metabolic disturbances associated with obesity. On the other hand, SLC35D3 knockin mice are protected from diet-induced obesity and are characterized by an overall healthy metabolic phenotype. Mechanistically, the authors claim that SLC35D3 inhibits Notch1 signaling. This is consistent with a study showing that adipose-specific inactivation of Notch1 leads to browning of WAT and increased expression of UCP1.

The authors studied an interesting topic, and the phenotypes of the mice are convincing. However, the paper is not very well written, many details are missing and the mechanistic data are not convincing. Please find below my specific comments.

Response: Thanks for your helpful suggestion. According to the comments, we have supplemented more data for supporting the statements, including 3vs.3 more samples in groups with marginal significance and additional experiments. We have also provided better representative immunoblots images and quantification of three independent experiments for all Western blots in the revised manuscript. We have doublechecked the whole manuscript and one native speaker have revised the manuscript. Detailed point-to-point responses were shown as below.

1. Many signaling pathways have been described in the literature that influence thermogenic pathways. A convincing rationale to study Notch1 is not provided.

Response: Thanks for the helpful suggestion. To determine the mechanism underlying *Slc35d3*'s function in manipulating the browning of WAT, we analyzed the RNA-seq data obtained from IngWAT with SAKI mice. A Kyoto Encyclopedia of Genes and Genomes (KEGG) pathway-enrichment analysis and GSEA based on KEGG pathways revealed that the genes enriched by *Slc35d3* overexpression were significantly related to Notch signaling pathway (Figure 6A). RNA-seq data showed that genes associated with Notch signaling was down-regulated in *Slc35d3*-overexpression IngWAT from SAKI mice (Figures 6B). Given the central role of Notch1 in manipulating the browning of WAT.¹ Therefore, Notch1 signaling was further explored in our study. The Figures 6A-6B were shown as below:

A

B

(A) KEGG pathway-enrichment analysis and GSEA based on KEGG pathways) in the comparison between WT and SAKI IngWAT.

(B) GSEA of Notch signaling pathway ([NES] -1.92, [ES] -0.38; FDR q 1.51E-06) gene sets.

2. It is hard to imagine that the presented model is correct. SLC35D3 is a transmembrane protein that is located in vesicular membranes. Based on the presented data, the authors proposed a model (Figure 7N) that SLC35D3 is a (secreted?) protein that inhibits the interaction of Notch1 with its ligands. The authors need to show how and in which cell compartment Notch1 can interact with SLC35D3.

Response: Thanks for the helpful suggestion. SLC35D3 is predicted as an orphan nucleotide sugar transporter or a fringe connection-like protein with 10 transmembrane domains.² We have revised the model as below:

3. Figure 1C. For ingWAT and epiWAT the authors showed 3 samples per group. For BAT 5 samples are shown, why?

Response: We have unified the sample numbers and provided better representative WB imaging as below:

4. Figure 1H. Except for the asWAT, all adipose tissue depots show high SLC35D3 protein expression. Why there is no protein in asWAT?

Response: The lower expression of Slc35d3 in asWAT resulted in the weak band. We have optimized the protein loading amount and antibody concentration and reconducted the experiment. The revised Figure 1H was shown as below:

5. Figure 1 legend. Schematic diagram does not show the generation of ob/ob or DIO mice. This is one example of many others where the text does not describe the illustrations correctly or in detail. The entire text needs to be carefully checked and re-written, I do not see it as my task as a reviewer to edit the entire text.

Response: Sorry for the confusion. We have supplemented the description of the generation of ob/ob or DIO mice in detail. The revised figure and figure legends were shown as below:

Figure 1. Slc35d3 expression was lower in fat tissues of obese mice.

(A) Schematic diagram showed that the generation of ob/ob mice deficient in Lep. Ob/ob mice fed a normocaloric diet (NCD) from fourth week for 10 weeks. (D) Schematic diagram showed that the generation of DIO mice. DIO mice fed a NCD from 4 to 6 weeks of age and fed a high fat diet (HFD) from 6 to 20 weeks of age.

6. Figure 2B. The reductions of SLC35D3 protein in epiWAT, asWAT and BAT are not very convincing. In addition, because protein expression (especially including thermogenic-acting proteins such as UCP1) in adipose tissues often exhibit high variability, it is not appropriate to show protein expression with an n of 1. This is true for almost all Western blots shown, also lacking adequate quantification.

Response: Thank the reviewer for the helpful comment. We have double-checked and conducted QPCR analysis of *Slc35d3* expression in fat pads (IngWAT, EpiWAT, AsWAT and BAT) and other tissues. The revised Figure 2B was shown as below:

In addition, all Western blot data shown represented at least three independent experiments in the study. We have provided more representative immunoblots images and supplemented the summary

of the quantification of three independent experiments for all Western blots shown in the manuscript. For example, Western blot detection of Slc35d3 expression in adipose tissues from SAKO mice and controls, and their quantification were shown in Figure 2C as below:

Furthermore, three independent western blot detection of Ucp1 expression in IngWAT from SAKI mice and controls when fed an HFD and their quantification were shown in Figure 5K as below:

7. Figures showing indirect calorimetry. Compared to their controls, SLC35D3 knockout and knockin mice have a different body weight. The indirect calorimetry data are presented per kg body weight, this is not correct and needs to be changed (see for instance PMID: 34489606). Response: Thanks for the helpful comment. We have revised all indirect calorimetry data in the manuscript according to the methods in PMID: 34489606. The revised results were shown as below:

Figure 3. (D) Energy expenditure of SAKO (n=5) mice and their Flox littermates (n=5) were assessed by ANCOVA (analysis of covariance) over 24 h for fed a NCD

Figure 5. (E) Energy expenditure of SAKI (n=5) mice and their WT littermates (n=5) were assessed by ANCOVA (analysis of covariance) over 24 h for fed a HFD.

Figure S5. (E) Energy expenditure of SAKI (n=5) mice and their WT littermates (n=5) were assessed by ANCOVA (analysis of covariance) over 24 h for fed a NCD.

8. Compared to the normal chow diet, high fat diet feeding should increase ectopic lipid accumulation in the liver. This is not visible in the HE stainings.

Response: We have double-checked and provided more representative HE imaging as below:

9. Presented plasma triglyceride levels are ca 0.05 mMol (4 mg/dl). This is impossible, these values are at least 10 times lower than normal.

Response: Sorry for the mistake. We have double-checked and found that we diluted the serum 10 times with normal saline before measurement. We should multiply the measurement result by 10 to get the correct triglyceride levels. We have supplemented the correct triglyceride levels in the manuscript as below:

Figure 2. (I) Serum lipid profiles (TG, TC and FFA) in SAKO (n=9) mice and their Flox littermates (n=9).

Figure 4. (I) Serum lipid profiles (TG, TC and FFA) in SAKI (n=6) mice and their WT littermates (n=6) on HFD.

Figure S3. (F) Serum lipid profiles (TG, TC and FFA) in SAKI (n=9) mice and their WT littermates (n=9) on NCD.

Figure 7. (G) Serum lipid profiles (TG and TC) in DBZ-treated SAKO (n=6) and Flox mice.

Figure S10. (F) Serum lipid profiles (TG and TC) in DBZ-treated SAKI (n=6) and WT mice.

10. Figure 4K. The glucose values after insulin injection run almost parallel, i.e. different insulin resistance between the genotypes cannot be assumed.

Response: Thank the reviewer for the helpful comment. We have revised the description as “the SAKI mice showed significantly improved glucose tolerance and insulin sensitivity compared to their controls”.

11. Figure 5L, the authors describe that lipolysis-associated genes are higher in knockin mice. Actually, they show the opposite.

Response: Sorry for the error. We have provided the corrected Figure 5N as below:

12. Figure 6B. Knockin mice have 100-fold induction on the gene expression levels. However, on the protein level I cannot see a difference between wild type and knockin mice.

Response: We have provided more representative WB imaging of Figure 6D as below:

13. Figures 6H/J. The IMF pictures are not very convincing. For instance, the ECD of Notch1 is located in one dot close to nucleus, which co-localize with SLC35D3. How does fit to the model? The claims described by the authors in the text are not justified by the presented data.

Response: Sorry for the unclear imaging. Cellular colocalization of SLC35D3 and NOTCH1 ECD was observed. We have provided more representative immunofluorescence imaging of Figure 6M as below:

M

14. The authors used quantitative flow cytometry data of primary adipocytes and HEK cells to show the inhibitory effect of SLC35D3 on Notch1 – ligand interaction. Details on the flow cytometry approach are lacking. Furthermore, Flow cytometry leads to the destruction of sensitive mature adipocytes. Thus, these data must be justified by alternative experimental approaches.

Response: Thank the reviewer for the helpful comment. We have supplemented the flow cytometry approach in detail. The information was shown as below:

Flow ligand assay. The study was performed as previously described³. HEK 293T cells were plated in 10 cm plates and transfected with 9 μ g of SLC35D3 plasmid per well by using transfection reagent. 48 h post-transfection, cells were fixed using 4% PFA for 10 min at room temperature, centrifuged at $115 \times g$ for 10 min and wash the cells twice. Next, cells were incubated for 15 min at 4 °C in FcR block media. Then, soluble ligands were incubated at 4 °C for 1 h with rotation and wash the cells twice. Secondary anti-Fc antibody was incubated for 30 min as above and wash the cells twice. Add 250 μ l of ice-cold ligand binding buffer to each cell pellet, mix and pass through the strainer cap of a 5 ml polystyrene round-bottom tube. This removes clumped cells immediately prior to flow cytometry. The MFI of R-Phycoerythrin (PE)-labeled ligands emitting fluorescence at 660 nm was determined with a FACScalibur (BD Biosciences).

In addition, we have added immunofluorescence assay to determine the binding of NOTCH1 ECD to its ligands Dll4 and Jag1 was inhibited in the presence of SLC35D3 in HEK293T cells. The added Figure 6Q was shown as below:

Q

15. Figure legend 6N. Gene expression are normalized to the housekeeper beta-actin? In the figure, the authors show nucleo-cytoplasmatic ratios of proteins.

Response: Sorry typo. Gene expression in cytosol and nucleus are normalized to the Gapdh and Lamin b1, respectively. In addition, we have added the nucleo-cytoplasmatic ratios of proteins as below:

T**U**
16. Line 418. If SLC35D3, as proposed by the authors, is an inhibitor of Notch1 signaling, the authors have to do the experiments with the pharmacological inhibitor in SLC35D3 knockin mice (and not in the knockout). In the SLC35D3 knockin mice, the effect by DBZ treatment should be blunted.

Response: To confirm the mechanism of SLC35D3 protecting from obesity via Notch1 signaling, we firstly determined the rescue effect of Notch inhibition on the improvement of obesity and glucose metabolism in the SAKO mice. More sensitive response to Notch inhibition on browning of IngWAT and amelioration of obesity was observed in the SAKO mice. We further detected the effect by DBZ treatment on manipulation of obesity in the SLC35D3 knockin mice. The results demonstrated that the effect of Notch inhibition on the improvement of obesity in the SAKI mice was blunted, which were shown in Figure S10 as below:

Reviewer #3 (Remarks to the Author):

The authors put forth intriguing work on the possible novel functions of the protein SLC35D3 in regulation of obesity in mice, possibly through regulation of Notch signaling. While this is a noteworthy result, there are several significantly worrisome issues throughout the report.

Response: Thanks for your helpful suggestions. According to the comments, we have supplemented more data for supporting our statements. Detailed point-to-point responses were shown as below.

The authors put forth intriguing work on the possible novel functions of the protein SLC35D3 in regulation of obesity in mice, possibly through regulation of Notch signaling. While this is a noteworthy result, there are several significantly worrisome issues throughout the report.

1) the authors would benefit from significant editorial oversight; multiple sentences throughout the work are incomplete, confusing, or run-on. Furthermore, the constant shift between figures within the paper along with the supplementary data has led to text and factual inconsistencies (ie in describing expression of lipolysis-related genes and figure 5L, they state "higher expression..." when the figure shows the opposite; similarly, the description of 5K says "smaller" when the data shows larger cell size as plotted.) Connected to this and worrisome when considering data analysis is that MCP1 and CCL2 are suggested/inferred to be different proteins, even to the point of different primers used for expansion, when these are actually the same gene product/protein.

Response: Sorry for the error. Both immunoblotting and immunohistochemical staining also showed higher Ucp1 levels in the SAKI IngWAT (Figures 5K and 5L). Smaller adipocytes were found in the SAKI mice compared to their controls (Figure 5M). We have corrected the Figures 5K-5M and their description as below:

In addition, we have deleted the description of Mcp1 as below:

Figure S1. (C) QPCR analysis of the expression of the inflammatory factors Il-6, Il-1 β , Tnf- α , and Ccl2 in EpiWAT from SAKO (n=9) and Flox (n=9) mice. (D) Representative western blot for Il-6, Tnf- α , Ccl2 and β -Actin protein in EpiWAT of SAKO (n=3) and Flox (n=3) mice fed a NCD. Right panel, summary of the quantification of three independent experiments.

Overall, we have doublechecked the whole manuscript and one native speaker have revised the manuscript.

2) the work is original, but suffers from a several inconsistencies. While the protein itself is definitely expressed in adipose tissue, it is also expressed in liver. Liver Notch signaling alterations override the adipose effect when Notch is inhibited; further exploration of the effects observed in the liver during g-sec inhibition would therefore be a significant benefit. Additional evidence is therefore needed.

Response: We have selected more representative immunoblots images to confirm the adipose tissue specificity of the SLC35D3 KO in Figure 2C. Our Figure 2 showed that the expression of Slc35d3 was significantly reduced in IngWAT, AsWAT, EpiWAT and BAT, while there was no change in other tissues, including liver, spleen, kidney and heart, indicating the adipose tissue specificity of the Slc35d3 KO. The revised Figures 2B-2C were shown as below:

In addition, we added the expression of Notch signaling was similar in SAKO/SAKI livers and their controls. A marked improvement of hepatic steatosis in the Slc35d3 knockin mice, suggested that other factors contributed to the organ-to-organ cross-talk and improvement in liver metabolism. The information was shown as below:

Figure S1. (H) Representative western blot for Slc35d3, Notch1, Hes1 and β -Actin protein in Liver of SAKO (n=3) and Flox (n=3) mice. Right panel, summary of the quantification of three independent experiments.

Figure S4. (H) Representative western blot for Slc35d3, Notch1, Hes1 and β -Actin protein in Liver of SAKI (n=3) and WT (n=3) mice fed a NCD. Right panel, summary of the quantification of three independent experiments. (P) Representative western blot for Slc35d3, Notch1, Hes1 and β -Actin protein in Liver of SAKI (n=3) and WT (n=3) mice fed a HFD. Right panel, summary of the quantification of three independent experiments.

Furthermore, in analyzing metabolic outcomes in the KI and KO mice, food intake and movement should be accounted for. In addition, timing of the variety of tests (GTT/ITT, etc) should be clearly documented; what if the effects on change in GTT and ITT are simply due to the alterations in weight, or is there a direct effect of altering Notch signaling on insulin signaling (ie connections to the Foxo-Notch data)? It would be vital to see if there are changes to the physiology of these mice BEFORE the weight gain.

Response: Thanks for the helpful suggestion. We have provided food intake and movement in the KI and KO mice. The information was shown as below:

Figure 2. (F) Food intake of SAKO (n=8) and Flox (n=7) mice on NCD.

Figure 3. (A) Activity of SAKO (n=5) and Flox (n=5) mice on NCD. (B) Oxygen consumption of SAKO (n=5) mice and their Flox littermates (n=5) fed a NCD.

Figure 5. (A) Food intake of SAKI (n=7) mice and their WT littermates (n=7) on HFD. (B) Activity of SAKI (n=5) mice and their WT littermates (n=5) on HFD.

Figure S5. (A) Food intake of SAKI (n=6) mice and their WT littermates (n=6) on NCD. (B) Activity of SAKI (n=5) mice and their WT littermates (n=5) on NCD.

Furthermore, we supplemented the timing of the variety of tests (GTT/ITT) in Methods. The information was shown as below:

Blood glucose measurements. Five microliters of blood collected from the tail vein was dropped onto a glucose test strip and measured by a glucometer. For glucose tolerance tests, mice were given intraperitoneal injection of 100 mg ml/ml D-glucose (2 g/kg body weight) after overnight fasting 16 h, and tail blood glucose concentrations were monitored, when the mice were 14 weeks old. For insulin tolerance tests, mice were fasted for 4 h from 9:00 am before intraperitoneal administration of human insulin (0.75 U/kg body weight), and tail blood glucose concentrations were monitored, when the mice were 16 weeks old.

In addition, we have detected the GTT/ITT data, which was impaired in SAKO mice, when these mice before the weight gain. The data was shown as below:

Our results also found that more sensitive response to Notch inhibition on glucose tolerance and insulin sensitivity in SAKO mice as below:

The insulin receptor substrate proteins link to the PI3K/AKT pathway, leading to the regulation of multiple metabolic pathways.⁴ One function of insulin action, mediated via AKT, is the phosphorylation of members of the FOXO family of transcription factors,⁵ which are very important downstream targets of the PI3K/AKT pathway,⁶ and play an important roles in insulin action.^{7,8} Our RNA-Seq data showed no significant difference in FOXO signaling of GO enrichment in IngWAT between SAKI mice and their controls. Additionally, we have also measured the expression of Akt and phosphorylated-Akt in IngWAT and found no significant difference between the SAKI mice and their controls.

In the Notch data (figure 6M specifically), further quantification of expression is needed. Only a single data point is shown, and for some reason the WT and flox mice (which should have similar Notch levels) appear to be different.

Response: Thanks for the helpful suggestion. We have selected better representative WB imaging and supplemented the quantification of all WB imaging in the manuscript. In addition, it seems that the Notch levels of WT and flox mice are different, which may be caused by different protein loading. The quantification of the Notch data was shown as below:

R

S

Furthermore, while the protein of interest does appear to bind the extracellular portion of Notch1, could all the data be explained by the down-regulation of Notch1 expression itself?

Response: Our results revealed that Slc35d3 inhibited Notch1 receptor binding to its ligands via binding with Notch1 ECD, inhibiting the activation of notch1 signaling. In turn, the inactivation of Notch1 inhibited NICD translocated to the nucleus. Therefore, not all data could be explained by the down-regulation of Notch1 expression itself.

Finally, throughout the paper, "relative" is used for mRNA, protein quantification, etc. This is confusing, as it is unclear what the number is relative to...all this data would best be explained as fold change from a control

Response: Thanks for the helpful comment. We have used "mRNA expression (relative to β -Actin)" for mRNA, and used "Protein levels (relative to β -Actin)" for protein quantification. we have also changed all figures accordingly.

References

- 1 Bi, P. *et al.* Inhibition of Notch signaling promotes browning of white adipose tissue and ameliorates obesity. *Nat Med* **20**, 911-918, doi:10.1038/nm.3615 (2014).
- 2 Zhang, Z. *et al.* Mutation of SLC35D3 causes metabolic syndrome by impairing dopamine signaling in striatal D1 neurons. *PLoS Genet* **10**, e1004124, doi:10.1371/journal.pgen.1004124 (2014).
- 3 Varshney, S. & Stanley, P. Notch Ligand Binding Assay Using Flow Cytometry. *Bio Protoc* **7**, doi:10.21769/BioProtoc.2637 (2017).
- 4 Boucher, J., Kleinridders, A. & Kahn, C. R. Insulin receptor signaling in normal and insulin-resistant states. *Cold Spring Harb Perspect Biol* **6**, doi:10.1101/cshperspect.a009191 (2014).
- 5 Gross, D. N., Wan, M. & Birnbaum, M. J. The role of FOXO in the regulation of metabolism. *Curr Diab Rep* **9**, 208-214, doi:10.1007/s11892-009-0034-5 (2009).
- 6 Farhan, M. *et al.* FOXO Signaling Pathways as Therapeutic Targets in Cancer. *Int J Biol Sci* **13**, 815-827, doi:10.7150/ijbs.20052 (2017).
- 7 Homan, E. P. *et al.* Differential roles of FOXO transcription factors on insulin action in brown and white adipose tissue. *J Clin Invest* **131**, doi:10.1172/JCI143328 (2021).
- 8 Barthel, A., Schmoll, D. & Unterman, T. G. FoxO proteins in insulin action and metabolism. *Trends Endocrinol Metab* **16**, 183-189, doi:10.1016/j.tem.2005.03.010 (2005).

REVIEWERS' COMMENTS:

Reviewer #1 (Remarks to the Author):

The authors of the new version of “SLC35D3 promotes white adipose tissue browning to improve obesity by Notch signaling” have clearly put in a lot of extra work. As a consequence the first part of the manuscript, especially the characterization of SLC35D3 knockout and overexpression both in vivo as well as ex vivo, has significantly improved. Most importantly, the authors exchanged all western blot panels with low quality against high quality images and added quantifications of at least three independent experiments. In addition, for all experiments with only minor changes, the authors have increased the number of repetitions or cohort sizes to generate clearer data. Lastly, the phenotypic characterization of the SAKO and SAKI mice, especially with respect to their metabolic phenotype, has been improved and important controls have been added. Altogether, those new panels and additional experiments have dramatically improved quality and credibility of the data.

Unfortunately, however, there is still not enough convincing data in order to draw clear conclusions as to how SLC35D3 interacts with Notch to modulate downstream signaling.

1. The adipocytes from IngWAT of the Flox mice in panel F3L look a lot like the adipocytes from SAKI mice in panel F5M. Instead, the adipocytes from WT mice in this panel look really huge and a lot like those of SAKO mice. Of course, cell sizes and appearances can vary between mice and even between different images taken of the same animal, however this difference between the different control mice seems striking.

2. The author's conclusion that binding of SLC35D3 to the ECD of Notch would inhibit the binding of Notch to its ligand is not convincingly supported by the data. A major concern is that the authors show in several experiments (panel F6A-H), that SLC35D3 level inversely correlate with the level of Notch, Hes1 and Hey1. As already suggested by one of the other reviewers, eventually the observed reduction in Notch signaling upon enhanced expression of SLC35D3 steams from a general reduction in Notch. Could it be, that binding of SLC35D3 to Notch would destabilize the receptor rather than interfere directly with the binding to its ligand?

3. The IF performed to probe for a co-localization of SLC35D3 and the ECD of Notch1 lacks quality. The 2 cells shown in panel F6M look rounded up and unhappy. To assess a proper co-localization of both proteins, it would be vital to include markers for the main organelles/membranes, the proteins are described to localize to. Hence, a proper membrane marker, as well as a marker for the ER (and/or endosome) would be required. It would also be great if more than 2 cells were shown per image and if images were taken at higher magnifications. This is in fact one of the most important panels to support the hypothesis that SLC35D3 and the extracellular domain of Notch1 physically interact, especially given that so far, SLC35D3 is believed to localize intracellularly.

4. In line with point 2, the flow cytometric experiments shown in panel F6O and P are insufficient to conclude that binding of SLC35D3 would dampen ligand binding to Notch1. If, as shown in the same figure, Notch itself decreases following overexpression of SLC35D3 it is to be expected, that less of its ligands can bind. To control for this, it would be important to normalize the data over overall notch level between the different conditions. In addition, also the IF shown in panel F6Q doesn't support the authors conclusion. All conditions do still show a co-localization of notch and Jag1 or DLL4. The difference is mostly that protein levels of notch seem overall reduced upon overexpression of SLC35D3.

Minor comments

1. It would have been of great help, if major text changes and new additions would have also been highlighted in the manuscript.
2. It is confusing that in panel F1H, the samples on the western blot have a slightly different order than in the quantification graph. The panel would read much easier if the order was the same.
3. The H&E staining shown in panel SF1F lacks quality and doesn't allow to assess if there is or not a difference between SAKO and control mice.
4. In panel F6O and P it is hard to distinguish between HEK293T WT and SLC35D3 and WT and SAKI, respectively, because one cannot see the shape of the dots well.
5. The paragraph "SLC35D3 inhibited the Notch1 signaling by interacting with NOTCH1 extracellular domains" still contains a lot of language mistakes and would benefit from some editing.

6. To show significance, the authors sometimes show stars and sometimes actual p values. It would be easier if this was kept consistent.

Reviewer #2 (Remarks to the Author):

In the revised version, the quality of the manuscript could be increased. However, there are still open questions and some points have not been addressed in an adequate way.

1- In Figure 1, the authors show that SLC35D3 is significantly reduced in obesity. This observation raises the question Why is SLC35D3 downregulation not associated with increased browning?

2- Figure S1F. The magnification in the two images is not the same. The bars cannot be correct.

3- Figure S1G. The content of hepatic FFA is almost as high as that of hepatic triglycerides. This is most likely an artefact of the method used.

4- Figures 3B and 3C. Oxygen consumption and carbon dioxide production as still calculated by kg body weight. This is not correct. The interpretation of the data is also difficult (impossible) as the animals are disturbed in such a way that the natural day-night rhythm of energy consumption is abrogated.

5- Figure 3/S2. The authors claim that browning of WAT is responsible for the phenotype. However, the appearance of BAT tissues from wild type and knockout mice looks quite different. The expression of thermogenic genes / proteins other than UCP1 is not shown.

6- Figures 5C/D / Figures 7 L/M. See point 4.

7- The manuscript is still not well written. The text contains several grammatical errors, sentences are not ending correctly, and in the description of the results details are lacking and causality / connection between experiments are difficult to follow. An extensive correction by a scientific editor would be fundamental prior to publication.

8- Figure 7Q. The IMF pictures are not convincing, what is the meaning of fold change in this context?

9- The model that ligand interaction with the extracellular domain of Notch1 is inhibited by SLC35D3 at the plasma membrane is not supported by the data presented. IMF stainings show vesicular co-localization of SLC35D3 and the ECD.

Reviewer #3 (Remarks to the Author):

The authors have re-submitted a paper outlining novel findings surrounding the function of SLC35D3 and its role in obesity in mice. While the findings are interesting, the edits still have not completely solved the initial paper's problems.

1) while the authors have had some improvement in the overall writing, there are still multiple sentence structure problems, typos, etc, scattered throughout the paper, that leads to significant confusion when reading and trying to follow the scientific argument. Ongoing significant editing is needed.

2) i still have concerns that no effect was seen in via Notch inhibition; while i agree that there may be an interaction between SLC35D3 and the ECD of Notch, whole-body treatment of mice with DBZ would have a direct effect on the liver as well as on adipose tissue; the liver-mediated improvements via Notch inhibition in whole body metabolism may outweigh the effects on adipose tissue. Adipose-specific targeting of Notch will be required to make the claims that all the benefits of Notch blockade by overexpression of SLC35D3 is at the level of adipose tissue changes. The improvements or damage by over-expression or KO of the protein might simply be tied to the weight effects

If the purpose of the paper is to show that the effects of this protein are all mediated via the Notch cascade, the notch findings should not be relegated to supplemental data

3) the data still is consistent with much of the metabolic effects being tied specifically to the weight gain; with the new data of GTT of aged/weight matched pre-weight gain, the effects are blunted and rather minimal; only the 0 and 90 min time points are affected. I would think then that a majority of the outcomes observed are tied more to weight gain, esp. in light of the note that SLC35D3 is a UDP glucose transporter expressed in neurons, and that mutations lead to lower motor activity etc., and that it is classically on vesicular, not cell membranes

4) the methodology for glucose uptake is flawed; it should be a clamp study or the like with a nonmetabolizable analogue

Responses to Reviewer #1

Remarks to the Author

Unfortunately, however, there is still not enough convincing data in order to draw clear conclusions as to how SLC35D3 interacts with Notch to modulate downstream signaling.

Response: Thank you for your helpful comments. We apologize for the confusion regarding how SLC35D3 interacted with Notch to modulate downstream signaling. Our hypothesis was that the interaction between SLC35D3 and NOTCH1 in the ER retained NOTCH1 in the cytoplasm, especially in the ER, thereby inhibiting the transportation of NOTCH1 to the cytomembrane as well as the activation of NOTCH1 signaling. We have supplemented more data to make this clear.

To investigate the potential mechanism by which Slc35d3 regulates IngWAT browning, we isolated IngWAT tissues from SAKI mice for transcriptome sequencing analysis. Both Kyoto Encyclopedia of Genes and Genomes (KEGG) pathway enrichment analysis and Gene-set Enrichment Analysis (GSEA) revealed that Notch signaling was markedly inhibited in IngWAT from SAKI mice (Fig. 6A-B). The qPCR results also suggested that the expression levels of Notch1 signaling-related genes were significantly downregulated in the IngWAT of SAKI mice but markedly upregulated in the IngWAT of SAKO mice (Fig. 6C, E), which was further confirmed by immunoblotting assay (Fig. 6D, F).

Inhibition of Notch1 signaling may be caused by decreased receptor levels resulting from decreased de novo expression, increased degradation, and impaired transport leading to decreased insertion into the plasma membrane¹. We speculated that the downregulation of Notch1 signaling might be due to the inhibition of Notch1 transcription or translation levels by Slc35d3 in mice. We then determined the NOTCH1 expression levels at different times in HEK293T cells overexpressing SLC35D3. Intriguingly, no difference in *NOTCH1* mRNA levels was found after 36 h of SLC35D3 overexpression, but a significant decrease was observed after 48 h (Fig. 6G). The results of the protein level assay were also interesting, showing that the protein

levels of NOTCH1 did not change after 48 h but decreased significantly after 60 h (Fig. 6H). These results showed that reduced NOTCH1 signaling was not caused by the inhibition of SLC35D3 on NOTCH1 transcription or translation levels. Next, we determined whether SLC35D3 altered the half-life of the NOTCH1 protein in HEK293T cells treated with cycloheximide (CHX). The results showed that SLC35D3 did not change the protein degradation of NOTCH1 (Fig. 6I). These results suggested that the inhibition of NOTCH1 signaling by SLC35D3 was not caused by inhibition of NOTCH1 transcription or translation levels or increased protein degradation.

In signal-receiving cells, NOTCH1 receptors are first generated in the ER and then transported to the Golgi apparatus where NOTCH1 receptors are cleaved into heterodimers (S1 cleavage) and transported to the cell membrane. Transit out of the endoplasmic reticulum (ER) has been shown to play a critical role in the expression of Notch1 at the cell surface¹⁻³. We speculated that SLC35D3 regulated the trafficking of NOTCH1 from the ER to the cytomembrane and detected the subcellular localization and levels of NOTCH1 in the cytoplasm and on the cytomembrane. Interestingly, the levels of NOTCH1 on the cytomembrane were decreased in HEK293T cells overexpressing SLC35D3, while most NOTCH1 proteins were retained in the cytoplasm, especially in the ER (Fig. 7A-B). Consistently, western blot and flow cytometric assays also showed decreased levels of NOTCH1 on the cytomembrane (Fig. 7C-D).

To assess how SLC35D3 regulates the trafficking of NOTCH1, we detected the interaction between SLC35D3 and NOTCH1 in HEK293T cells via a coimmunoprecipitation (CoIP) assay. As shown in Figure 7E, NOTCH1 was coimmunoprecipitated with SLC35D3 in HEK293T cells. NOTCH1 proteins consist of a large extracellular domain (ECD), a transmembrane region, and a large NICD^{2,4}. Furthermore, we determined the specific domain to which SLC35D3 binds. First, the binding between SLC35D3 and NICD was determined in HEK293T cells transfected with both the SLC35D3 plasmid and NICD plasmid via immunofluorescence and CoIP assays. As shown in Figures 7F and S7A, neither coprecipitation of NICD protein with SLC35D3 nor cellular colocalization of SLC35D3 with NICD was observed. We then

determined the binding between SLC35D3 and ECD. Myc-tagged NOTCH1 ECD was coimmunoprecipitated with SLC35D3 in HEK293T cells and cellular colocalization of SLC35D3 with NOTCH1 ECD was also observed (Fig. 7G, I). The direct physical interaction between SLC35D3 and the NOTCH1 ECD was further confirmed by an in vitro pull-down assay (Fig. 7H).

Taken together, these results revealed that SLC35D3 interacted with the NOTCH1 ECD in the cytoplasm and then retained NOTCH1 in the cytoplasm, especially in the ER, thereby inhibiting the translocation of NOTCH1 to the cell membrane and thus inhibiting NOTCH1 signaling.

All of the results supported our hypothesis. These are described in Figure 6, Figure 7 and Figure S7A of the revised manuscript, which are also shown below.

Figure 6. SLC35D3 inhibited the activation of NOTCH1 signaling.

(A) KEGG pathway enrichment analysis results of IngWAT from WT and SAKI mice. (B) GSEA results of Notch signaling pathway gene sets ([NES] -1.92, [ES] -0.38; FDR q 1.51E-06). (C, E) QPCR analysis of *Slc35d3* and Notch1-related genes in IngWAT from SAKI (n=6) mice and their WT (n=6) littermates (C), as well as in IngWAT from SAKI (n=6) mice and their WT (n=6) littermates (E). (D, F) Western blot showing the expression of Notch1-related proteins in IngWAT from SAKI (n=3) mice and their controls (n=3) (D), as well as in IngWAT from SAKO (n=3) mice and their controls (n=3) (F).

(n=3) (F). (G) qPCR detection of *NOTCH1* expression in HEK293T cells overexpressing *SLC35D3*. (H) Western blot analysis of NOTCH1 protein in HEK293T cells overexpressing *SLC35D3*. Right panel, summary of the quantification of three independent experiments. (I) Western blot analysis of protein in HEK293T cells treated with CHX. Values are the fold induction of gene expression normalized to the housekeeping gene β -Actin, and data are expressed as the mean \pm SEM. Statistical significance was assessed by two-sided Student's t test (C-I). Statistical significance was determined as $P < 0.05$.

Figure 7. SLC35D3 increased the ER accumulation of NOTCH1 by interacting with the NOTCH1 extracellular domain.

(A-B) Immunofluorescence assays revealed the increased intracellular accumulation of NOTCH1 in the ER in SLC35D3-overexpressing cells (A), as well as the reduced NOTCH1 in plasma (B). (C) NOTCH1 levels were downregulated in the cell plasma when SLC35D3 was overexpressed. Right panel, summary of the quantification of three independent experiments. (D) Cell-based NOTCH1 levels in SLC35D3-overexpressing HEK293T cells and control cells were detected by flow cytometric analysis. Binding curves were generated by determining the NOTCH1 MFI. Right panel, summary of the quantification of three independent experiments. (E-G) CoIP was performed to assess the interaction between SLC35D3 and NOTCH1 (E), SLC35D3 and NICD (F), and SLC35D3 and the NOTCH1 ECD (G). (H) Direct interaction between SLC35D3 and NOTCH1-ECD. The bait protein, purified GST-SLC35D3, was immobilized on glutathione affinity resin and incubated with MYC-ECD protein lysate at 4 °C for at least 1 hour with gentle rocking motion on a rotating platform. The prey protein, MYC-ECD, was captured by the bait protein, and then the bait-prey protein was eluted. ECD and SLC35D3 were subsequently analyzed by western blotting. (I) Colocalization of SLC35D3 and NOTCH1 ECD. Data are represented as the mean \pm SEM. Statistical significance was assessed by two-sided Student's t test (C, D) and was determined as $P < 0.05$.

Figure S7.

A

(A) SLC35D3 and NICD did not colocalize.

1. The adipocytes from IngWAT of the Flox mice in panel F3L look a lot like the adipocytes from SAKI mice in panel F5M. Instead, the adipocytes from WT mice in this panel look really huge and a lot like those of SAKO mice. Of course, cell sizes and appearances can vary between mice and even between different images taken of the same animal, however this difference between the different control mice seems striking.

Response: Thank you for your helpful suggestion. We apologize for the confusion. Figure 3L shows H&E staining of IngWAT from the normal chow diet-fed (NCD-fed) SAKO mice and their Flox littermates, while Figure 5M shows H&E staining of IngWAT from HFD-fed SAKI mice and their WT littermates. HFD could induce adipocyte hypertrophy.^{5,6} Thus, the adipocytes from IngWAT of the WT mice in F5M were relatively large. We have double-checked our results and provided more representative HE images and cell size measurements. These are presented in Figure 3L and Figure 5M of the revised manuscript, which are also shown below.

Figure 3.

L

(L) Hematoxylin and eosin (H&E) staining and adipocyte cell size measurements of IngWAT from NCD-fed SAKO mice and their Flox littermates; scale bar, 500 μm ; 20 μm .

Figure 5.

M

(M) H&E staining and adipocyte cell size measurements of the IngWAT from HFD-fed SAKI mice and their WT littermates. Scale bar, 500 μm ; 20 μm .

2. The author's conclusion that binding of SLC35D3 to the ECD of Notch would inhibit the binding of Notch to its ligand is not convincingly supported by the data. A major concern is that the authors show in several experiments (panel F6A-H), that SLC35D3 level inversely correlate with the level of Notch, Hes1 and Hey1. As already suggested by one of the other reviewers, eventually the observed reduction in Notch signaling upon enhanced expression of SLC35D3 steams from a general reduction in Notch. Could it be, that binding of SLC35D3 to Notch would destabilize the receptor rather than interfere directly with the binding to its ligand?

Response: Thank you for your helpful comments. We apologize for the confusion regarding how SLC35D3 interacted with Notch to modulate downstream signaling. Our hypothesis was that the interaction between SLC35D3 and NOTCH1 in the ER retained NOTCH1 in the cytoplasm, especially in the ER, thereby inhibiting the transportation of NOTCH1 to the cytomembrane as well as the activation of NOTCH1 signaling. We have supplemented more data to make this clear.

To investigate the potential mechanism by which Slc35d3 regulates IngWAT browning, we isolated IngWAT tissues from SAKI mice for transcriptome sequencing analysis. Both Kyoto Encyclopedia of Genes and Genomes (KEGG) pathway enrichment analysis and Gene-set Enrichment Analysis (GSEA) revealed that Notch signaling was markedly inhibited in IngWAT from SAKI mice (Fig. 6A-B). The qPCR results also suggested that the expression levels of Notch1 signaling-related genes were significantly downregulated in the IngWAT of SAKI mice but markedly upregulated in the IngWAT of SAKO mice (Fig. 6C, E), which was further confirmed by immunoblotting assay (Fig. 6D, F).

Inhibition of Notch1 signaling may be caused by decreased receptor levels resulting from decreased de novo expression, increased degradation, and impaired transport leading to decreased insertion into the plasma membrane ¹. We speculated that the downregulation of Notch1 signaling might be due to the inhibition of Notch1 transcription or translation levels by Slc35d3 in mice. We then determined the NOTCH1 expression levels at different times in HEK293T cells overexpressing

SLC35D3. Intriguingly, no difference in *NOTCH1* mRNA levels was found after 36 h of SLC35D3 overexpression, but a significant decrease was observed after 48 h (Fig. 6G). The results of the protein level assay were also interesting, showing that the protein levels of NOTCH1 did not change after 48 h but decreased significantly after 60 h (Fig. 6H). These results showed that reduced NOTCH1 signaling was not caused by the inhibition of SLC35D3 on NOTCH1 transcription or translation levels. Next, we determined whether SLC35D3 altered the half-life of the NOTCH1 protein in HEK293T cells treated with cycloheximide (CHX). The results showed that SLC35D3 did not change the protein degradation of NOTCH1 (Fig. 6I). These results suggested that the inhibition of NOTCH1 signaling by SLC35D3 was not caused by inhibition of NOTCH1 transcription or translation levels or increased protein degradation.

In signal-receiving cells, NOTCH1 receptors are first generated in the ER and then transported to the Golgi apparatus where NOTCH1 receptors are cleaved into heterodimers (S1 cleavage) and transported to the cell membrane. Transit out of the endoplasmic reticulum (ER) has been shown to play a critical role in the expression of Notch1 at the cell surface¹⁻³. We speculated that SLC35D3 regulated the trafficking of NOTCH1 from the ER to the cytomembrane and detected the subcellular localization and levels of NOTCH1 in the cytoplasm and on the cytomembrane. Interestingly, the levels of NOTCH1 on the cytomembrane were decreased in HEK293T cells overexpressing SLC35D3, while most NOTCH1 proteins were retained in the cytoplasm, especially in the ER (Fig. 7A-B). Consistently, western blot and flow cytometric assays also showed decreased levels of NOTCH1 on the cytomembrane (Fig. 7C-D).

To assess how SLC35D3 regulates the trafficking of NOTCH1, we detected the interaction between SLC35D3 and NOTCH1 in HEK293T cells via a coimmunoprecipitation (CoIP) assay. As shown in Figure 7E, NOTCH1 was coimmunoprecipitated with SLC35D3 in HEK293T cells. NOTCH1 proteins consist of a large extracellular domain (ECD), a transmembrane region, and a large NICD^{2,4}. Furthermore, we determined the specific domain to which SLC35D3 binds. First, the

binding between SLC35D3 and NICD was determined in HEK293T cells transfected with both the SLC35D3 plasmid and NICD plasmid via immunofluorescence and CoIP assays. As shown in Figures 7F and S7A, neither coprecipitation of NICD protein with SLC35D3 nor cellular colocalization of SLC35D3 with NICD was observed. We then determined the binding between SLC35D3 and ECD. Myc-tagged NOTCH1 ECD was coimmunoprecipitated with SLC35D3 in HEK293T cells and cellular colocalization of SLC35D3 with NOTCH1 ECD was also observed (Fig. 7G, I). The direct physical interaction between SLC35D3 and the NOTCH1 ECD was further confirmed by an in vitro pull-down assay (Fig. 7H).

Taken together, these results revealed that SLC35D3 interacted with the NOTCH1 ECD in the cytoplasm and then retained NOTCH1 in the cytoplasm, especially in the ER, thereby inhibiting the translocation of NOTCH1 to the cell membrane and thus inhibiting NOTCH1 signaling.

All of the results supported our hypothesis. These are described in Figure 6, Figure 7 and Figure S7A of the revised manuscript, which are also shown below.

Figure 6. SLC35D3 inhibited the activation of NOTCH1 signaling.

(A) KEGG pathway enrichment analysis results of IngWAT from WT and SAKI mice. (B) GSEA results of Notch signaling pathway gene sets ([NES] -1.92, [ES] -0.38; FDR q 1.51E-06). (C, E) QPCR analysis of *Slc35d3* and Notch1-related genes in IngWAT from SAKI (n=6) mice and their WT (n=6) littermates (C), as well as in IngWAT from SAKI (n=6) mice and their WT (n=6) littermates (E). (D, F) Western blot showing the expression of Notch1-related proteins in IngWAT from SAKI (n=3) mice and their controls (n=3) (D), as well as in IngWAT from SAKO (n=3) mice and their controls (n=3) (F).

(n=3) (F). (G) qPCR detection of *NOTCH1* expression in HEK293T cells overexpressing *SLC35D3*. (H) Western blot analysis of NOTCH1 protein in HEK293T cells overexpressing *SLC35D3*. Right panel, summary of the quantification of three independent experiments. (I) Western blot analysis of protein in HEK293T cells treated with CHX. Values are the fold induction of gene expression normalized to the housekeeping gene β -Actin, and data are expressed as the mean \pm SEM. Statistical significance was assessed by two-sided Student's t test (C-I). Statistical significance was determined as $P < 0.05$.

Figure 7. SLC35D3 increased the ER accumulation of NOTCH1 by interacting with the NOTCH1 extracellular domain.

(A-B) Immunofluorescence assays revealed the increased intracellular accumulation of NOTCH1 in the ER in SLC35D3-overexpressing cells (A), as well as the reduced NOTCH1 in plasma (B). (C) NOTCH1 levels were downregulated in the cell plasma when SLC35D3 was overexpressed. Right panel, summary of the quantification of three independent experiments. (D) Cell-based NOTCH1 levels in SLC35D3-overexpressing HEK293T cells and control cells were detected by flow cytometric analysis. Binding curves were generated by determining the NOTCH1 MFI. Right panel, summary of the quantification of three independent experiments. (E-G) CoIP was performed to assess the interaction between SLC35D3 and NOTCH1 (E), SLC35D3 and NICD (F), and SLC35D3 and the NOTCH1 ECD (G). (H) Direct interaction between SLC35D3 and NOTCH1-ECD. The bait protein, purified GST-SLC35D3, was immobilized on glutathione affinity resin and incubated with MYC-ECD protein lysate at 4 °C for at least 1 hour with gentle rocking motion on a rotating platform. The prey protein, MYC-ECD, was captured by the bait protein, and then the bait-prey protein was eluted. ECD and SLC35D3 were subsequently analyzed by western blotting. (I) Colocalization of SLC35D3 and NOTCH1 ECD. Data are represented as the mean \pm SEM. Statistical significance was assessed by two-sided Student's t test (C, D) and was determined as $P < 0.05$.

Figure S7.

A

(A) SLC35D3 and NICD did not colocalize.

3. The IF performed to probe for a co-localization of SLC35D3 and the ECD of Notch1 lacks quality. The 2 cells shown in panel F6M look rounded up and unhappy. To assess a proper co-localization of both proteins, it would be vital to include markers for the main organelles/membranes, the proteins are described to localize to. Hence, a proper membrane marker, as well as a marker for the ER (and/or endosome) would be required. It would also be great if more than 2 cells were shown per image and if images were taken at higher magnifications. This is in fact one of the most important panels to support the hypothesis that SLC35D3 and the extracellular domain of Notch1 physically interact, especially given that so far, SLC35D3 is believed to localize intracellularly.

Response: Thank you for your helpful suggestion. We have repeated the experiment and supplemented more representative immunofluorescence images of the colocalization of SLC35D3 and the ECD/NICD of NOTCH1. Meanwhile, we have replaced those with images including more than 2 cells and supplemented ER marker (CALNEXIN). These are presented in Figure 7A-B, Figure 7I and Figure S7A of the revised manuscript, which are also shown below.

Figure 7.

(A-B) Immunofluorescence assays revealed the increased intracellular accumulation of NOTCH1 in the ER in SLC35D3-overexpressing cells (A), as well as the reduced NOTCH1 in plasma (B). (I) Colocalization of SLC35D3 and NOTCH1 ECD.

Figure S7.

(A) SLC35D3 and NICD did not colocalize.

4. In line with point 2, the flow cytometric experiments shown in panel F6O and P are insufficient to conclude that binding of SLC35D3 would dampen ligand binding to Notch1. If, as shown in the same figure, Notch itself decreases following overexpression of SLC35D3 it is to be expected, that less of its ligands can bind. To control for this, it would be important to normalize the data over overall notch level between the different conditions. In addition, also the IF shown in panel F6Q doesn't support the authors conclusion. All conditions do still show a co-localization of notch and Jag1 or DLL4. The difference is mostly that protein levels of notch seem overall reduced upon overexpression of SLC35D3.

Response: Thank you for your helpful suggestion. We apologize for the confusion regarding how SLC35D3 regulates Notch signaling. Our hypothesis was that the interaction between SLC35D3 and NOTCH1 in the ER retained NOTCH1 in the cytoplasm, especially in the ER, inhibiting the transportation of NOTCH1 to the cytomembrane as well as the activation of NOTCH1 signaling. We have supplemented more data to demonstrate that NOTCH1 was retained in the cytoplasm and its cytomembrane level was decreased in the presence of SLC35D3 via immunofluorescence, western blotting and flow cytometry. We have removed the receptor–ligand binding experiment to avoid confusion.

Minor comments

1. It would have been of great help, if major text changes and new additions would have also been highlighted in the manuscript.

Response: Thanks for your helpful comment. We have highlighted the major text changes and new additions in the revised manuscript to ensure clarity and ease of review.

2. It is confusing that in panel F1H, the samples on the western blot have a slightly different order than in the quantification graph. The panel would read much

easier if the order was the same.

Response: Thanks for your helpful comment. We apologize for the confusion. We have updated the panel to reflect the same order as the quantification graph in the revised manuscript to ensure consistency and ease of reading. These are presented in Figure 1A-B of the revised manuscript, which are also shown below.

Figure 1.

(A) RT-qPCR analysis of *Slc35d3* levels in EpiWAT, BAT, IngWAT, AsWAT, heart, spleen, liver, kidney and brain from WT (n=6) mice. (B) Representative western blot of *Slc35d3* and β -Actin protein in EpiWAT, BAT, IngWAT, AsWAT, heart, spleen, liver, kidney and brain from WT (n=3) mice. The summary of the quantifications of three independent experiments is shown in the right panel.

3. The H&E staining shown in panel SF1F lacks quality and doesn't allow to assess if there is or not a difference between SAKO and control mice.

Response: We apologize for the lack of quality in the H&E staining in Supplemental Figure S1F. In the revised version, we have provided more representative H&E staining images to demonstrate the differences between the SAKO and control mice. These are presented in Figure S1F of the revised manuscript, which are also shown below.

Figure S1.

(F) Representative images of H&E staining results of liver tissues from SAKO and Flox mice. Scale bar, 20 μ m.

4. In panel F6O and P it is hard to distinguish between HEK293T WT and SLC35D3 and WT and SAKI, respectively, because one cannot see the shape of the dots well.

Response: Thank you for your helpful suggestion. We have supplemented more data to demonstrate that NOTCH1 was retained in the cytoplasm and its cytomembrane level was decreased in the presence of SLC35D3 via immunofluorescence, western blotting and flow cytometry. We have removed the receptor-ligand binding experiment to avoid confusion. We have provided flow cytometric images and quantification analysis to demonstrate that cytomembrane NOTCH1 levels are reduced in the presence of SLC35D3. These are presented in Figure 7D of the revised manuscript, which are also shown below.

Figure 7

(D) Cell-based NOTCH1 levels in SLC35D3-overexpressing HEK293T cells and control cells were detected by flow cytometric analysis. Binding curves were generated by determining the NOTCH1 MFI. Right panel, summary of the quantification of three independent experiments.

5. The paragraph “SLC35D3 inhibited the Notch1 signaling by interacting with NOTCH1 extracellular domains” still contains a lot of language mistakes and would benefit from some editing.

Response: Thanks for your helpful suggestion. We apologize for the language mistakes. We rewrote this paragraph based on additional experiments and carefully checked and revised the entire manuscript.

6. To show significance, the authors sometimes show stars and sometimes actual p values. It would be easier if this was kept consistent.

Response: Thanks for your helpful suggestion. We have supplemented the actual p values in the revised manuscript. In some panels like the body weight curve, p values were provided in the figure legends due to lots of p values.

Responses to Reviewer #2

1- In Figure 1, the authors show that SLC35D3 is significantly reduced in obesity. This observation raises the question Why is SLC35D3 downregulation not associated with increased browning?

Response: Thanks for your helpful suggestion. Obesity mainly occurs due to an imbalance in energy homeostasis, including high energy intake and low energy expenditure⁷. both IngWAT browning and BAT activation can increase thermogenesis to consume energy, in which beige cells play an important role in metabolism and thermogenesis^{7,8}. Reduced energy expenditure may suggest suppression of WAT browning and/or BAT activation in SAKO mice. Our results showed that the mRNA expression levels of thermogenesis genes, including *Ucp1*, *Prdm16* and *Pgc1- α* , were decreased in the IngWAT of SAKO mice, as well as beige cell markers (*Tbx1* and *Tmem26*) and mitochondrial genes (*Cpt1a*, *Cpt1b*, and *Cpt2*) (Fig. 3G-I). Similarly, immunoblot analysis also revealed significantly lower expression of *Ucp1* and *Pgc1- α* in SAKO IngWAT (Fig. 3J). Less positive UCP1 staining in SAKO IngWAT further suggested decreased browning of IngWAT (Fig. 3K). In addition, adipocytes were larger in SAKO mice (Fig. 3L). Given the correlation between lipolysis and metabolic homeostasis, we then examined the expression levels of lipolysis-associated genes (*Atgl*, also called *Pnpla2*, and *Hsl*, also called *Lipe*) by qPCR, which were decreased in the IngWAT of SAKO mice (Fig. 3M). We also observed similar phenotypes in the EpiWAT of SAKO mice (Fig. S2A-D). However, the color of BAT from SAKO mice was consistent with that from Flox mice, indicating a similar lipid content. H&E staining further confirmed that the multilocular lipid droplets of BAT from the SAKO mice were similar to those of the controls (Fig. S2E). In addition, no differences in the mRNA or protein levels of *Ucp1* were observed in classical BAT (Fig. S2F-H).

Together, these results suggested that adipocyte-specific deletion of *Slc35d3* impaired the browning of WAT. These are presented in Figure 3G-M and Figure S2 of the revised manuscript, which are also shown below.

Figure 3.

(G-I) qPCR detection results of thermogenesis gene expression (G), beige adipocyte marker (H) and mitochondria-related gene expression (I) in IngWAT from SAKO (n=9) mice and their Flox littermates (n=9). (J) Western blot analysis of thermogenesis protein expression in IngWAT from SAKO (n=3) mice and their Flox littermates (n=3). Right panel, summary of the quantification of three independent experiments. (K) Immunostaining for Ucp1 in the IngWAT of SAKO and Flox mice. Scale bar, 50 μm . (L) Hematoxylin and eosin (H&E) staining and adipocyte cell size measurements of IngWAT. Scale bar, 500 μm ; 20 μm . (M) qPCR analysis of the expression of lipolysis genes in IngWAT from SAKO mice (n=6) and their Flox littermates (n=6).

Figure S2.

(A, E) Representative images of H&E staining results of the EpiWAT (A) and BAT (E) from SAKO and Flox mice. Scale bar, 50 μm . (B, F) qPCR analysis of *Ucp1* expression in the EpiWAT (B) and BAT (F) from SAKO (n=6) mice and their Flox littermates (n=6). (C, G) Immunostaining for Ucp1 in the EpiWAT (C) and BAT (G) from SAKO and Flox mice. Scale bar, 50 μm . (D, H) Western blot analysis of Ucp1 expression in the EpiWAT (D) and BAT (H) from SAKO (n=3) mice and their Flox littermates (n=3). Right panel, summary of the quantification of three independent experiments.

Statistical significance was assessed by two-sided Student's t test (A-B, D, F and H). Values are expressed as the mean \pm SEM. Statistical significance was determined as $P < 0.05$.

2- Figure S1F. The magnification in the two images is not the same. The bars cannot be correct.

Response: We apologize for the confusion. The different magnifications of the two images in Supplemental Figure S1F resulted in incorrect bar sizes. We have revised SF1F with the same bar chart size and provided more representative H&E staining images to demonstrate the differences between the SAKO and control mice. These are presented in Figure S1F of the revised manuscript, which are also shown below.

Figure S1.

(F) Representative images of H&E staining results of liver tissues from SAKO and Flox mice. Scale bar, 20 μ m.

3- Figure S1G. The content of hepatic FFA is almost as high as that of hepatic triglycerides. This is most likely an artefact of the method used.

Response: We apologize for the mistake. We have double-checked and found that we made a normalization calculation error by liver weight. We have corrected this in the revised Figure S1G, which is also shown below.

Figure S1

(G) Hepatic TG, TC and FFA levels in SAKO (n=6) and Flox (n=6) mice.

4- Figures 3B and 3C. Oxygen consumption and carbon dioxide production as still calculated by kg body weight. This is not correct. The interpretation of the data is also difficult (impossible) as the animals are disturbed in such a way that the natural day-night rhythm of energy consumption is abrogated.

Response: Thank you for the helpful comment. We have recalculated the oxygen consumption and carbon dioxide production data of the SAKO mice by ANCOVA (analysis of covariance) in the revised Figure 3B-C, which are also shown below.

Figure 3.

(B) Oxygen consumption of NCD-fed SAKO (n=5) mice and their Flox littermates (n=5) was assessed over 24 h by ANCOVA (analysis of covariance). (C) Carbon dioxide

production of SAKO (n=5) mice and their Flox littermates (n=5) was assessed by ANCOVA over 24 h.

5- Figure 3/S2. The authors claim that browning of WAT is responsible for the phenotype. However, the appearance of BAT tissues from wild type and knockout mice looks quite different. The expression of thermogenic genes / proteins other than UCP1 is not shown.

Response: We apologize for the confusion. We have checked and found that the stacking of individual adipocytes in IngWAT of SAKO mice increased the staining observed in the previous slides. To rectify this issue, we have provided IHC images with higher quality in the revised manuscript. In addition, we have supplemented the qPCR results of more thermogenic genes, and the protein levels of two significantly different thermogenic genes. These are presented in Figure 3G-M and Figure S2 of the revised manuscript, which are also shown below.

Figure 3.

(G-I) qPCR detection results of thermogenesis gene expression (G), beige adipocyte marker (H) and mitochondria-related gene expression (I) in IngWAT from SAKO (n=9) mice and their Flox littermates (n=9). (J) Western blot analysis of thermogenesis protein expression in IngWAT from SAKO (n=3) mice and their Flox littermates (n=3). Right panel, summary of the quantification of three independent experiments. (K) Immunostaining for Ucp1 in the IngWAT of SAKO and Flox mice. Scale bar, 50 μm . (L) Hematoxylin and eosin (H&E) staining and adipocyte cell size measurements of IngWAT. Scale bar, 500 μm ; 20 μm . (M) qPCR analysis of the expression of lipolysis genes in IngWAT from SAKO mice (n=6) and their Flox littermates (n=6).

Figure S2.

(A, E) Representative images of H&E staining results of the EpiWAT (A) and BAT (E) from SAKO and Flox mice. Scale bar, 50 μm . (B, F) qPCR analysis of *Ucp1* expression in the EpiWAT (B) and BAT (F) from SAKO (n=6) mice and their Flox littermates (n=6). (C, G) Immunostaining for Ucp1 in the EpiWAT (C) and BAT (G) from SAKO and Flox mice. Scale bar, 50 μm . (D, H) Western blot analysis of Ucp1 expression in the EpiWAT (D) and BAT (H) from SAKO (n=3) mice and their Flox littermates (n=3). Right panel, summary of the quantification of three independent experiments.

Statistical significance was assessed by two-sided Student's t test (A-B, D, F and H). Values are expressed as the mean \pm SEM. Statistical significance was determined as $P < 0.05$.

6- Figures 5C/D / Figures 7 L/M. See point 4.

Response: Thanks for your helpful suggestion. We have recalculated the oxygen consumption, carbon dioxide production and energy expenditure data in the revised manuscript. These are presented in Figure 5C-E and Figure S5C-E of the revised manuscript, which are also shown below.

Figure 5.

(C-E) Oxygen consumption (C), carbon dioxide production (D) and energy expenditure (E) of HFD-fed SAKI ($n=5$) mice and their WT littermates ($n=5$) were assessed by ANCOVA over 24 h.

Figure S5.

(C-E) Oxygen consumption (C), carbon dioxide production (D) and energy expenditure

(E) of NCD-fed SAKI (n=5) mice and their WT littermates (n=5).

7- The manuscript is still not well written. The text contains several grammatical errors, sentences are not ending correctly, and in the description of the results details are lacking and causality / connection between experiments are difficult to follow. An extensive correction by a scientific editor would be fundamental prior to publication.

Response: Thanks for your helpful suggestion. We apologize for the language mistakes. We have re-examined and carefully revised the entire manuscript.

8- Figure 7Q. The IMF pictures are not convincing, what is the meaning of fold change in this context?

Response: We apologize for the confusion. We have supplemented more data to demonstrate that NOTCH1 was retained in the cytoplasm and its cytomembrane level was decreased in the presence of SLC35D3 via immunofluorescence, western blotting and flow cytometry in the revised manuscript. We have removed the receptor-ligand binding experiments to avoid confusion.

9- The model that ligand interaction with the extracellular domain of Notch1 is inhibited by SLC35D3 at the plasma membrane is not supported by the data presented. IMF stainings show vesicular co-localization of SLC35D3 and the ECD.

Response: Thank you for your helpful comments. We apologize for the confusion regarding how SLC35D3 interacted with Notch to modulate downstream signaling. Our hypothesis was that the interaction between SLC35D3 and NOTCH1 in the ER retained NOTCH1 in the cytoplasm, especially in the ER, thereby inhibiting the transportation of NOTCH1 to the cytomembrane as well as the activation of NOTCH1 signaling. We have supplemented more data to make this clear.

To investigate the potential mechanism by which Slc35d3 regulates IngWAT

browning, we isolated IngWAT tissues from SAKI mice for transcriptome sequencing analysis. Both Kyoto Encyclopedia of Genes and Genomes (KEGG) pathway enrichment analysis and Gene-set Enrichment Analysis (GSEA) revealed that Notch signaling was markedly inhibited in IngWAT from SAKI mice (Fig. 6A-B). The qPCR results also suggested that the expression levels of Notch1 signaling-related genes were significantly downregulated in the IngWAT of SAKI mice but markedly upregulated in the IngWAT of SAKO mice (Fig. 6C, E), which was further confirmed by immunoblotting assay (Fig. 6D, F).

Inhibition of Notch1 signaling may be caused by decreased receptor levels resulting from decreased de novo expression, increased degradation, and impaired transport leading to decreased insertion into the plasma membrane ¹. We speculated that the downregulation of Notch1 signaling might be due to the inhibition of Notch1 transcription or translation levels by *Slc35d3* in mice. We then determined the NOTCH1 expression levels at different times in HEK293T cells overexpressing SLC35D3. Intriguingly, no difference in *NOTCH1* mRNA levels was found after 36 h of SLC35D3 overexpression, but a significant decrease was observed after 48 h (Fig. 6G). The results of the protein level assay were also interesting, showing that the protein levels of NOTCH1 did not change after 48 h but decreased significantly after 60 h (Fig. 6H). These results showed that reduced NOTCH1 signaling was not caused by the inhibition of SLC35D3 on NOTCH1 transcription or translation levels. Next, we determined whether SLC35D3 altered the half-life of the NOTCH1 protein in HEK293T cells treated with cycloheximide (CHX). The results showed that SLC35D3 did not change the protein degradation of NOTCH1 (Fig. 6I). These results suggested that the inhibition of NOTCH1 signaling by SLC35D3 was not caused by inhibition of NOTCH1 transcription or translation levels or increased protein degradation.

In signal-receiving cells, NOTCH1 receptors are first generated in the ER and then transported to the Golgi apparatus where NOTCH1 receptors are cleaved into heterodimers (S1 cleavage) and transported to the cell membrane. Transit out of the endoplasmic reticulum (ER) has been shown to play a critical role in the expression of

Notch1 at the cell surface¹⁻³. We speculated that SLC35D3 regulated the trafficking of NOTCH1 from the ER to the cytomembrane and detected the subcellular localization and levels of NOTCH1 in the cytoplasm and on the cytomembrane. Interestingly, the levels of NOTCH1 on the cytomembrane were decreased in HEK293T cells overexpressing SLC35D3, while most NOTCH1 proteins were retained in the cytoplasm, especially in the ER (Fig. 7A-B). Consistently, western blot and flow cytometric assays also showed decreased levels of NOTCH1 on the cytomembrane (Fig. 7C-D).

To assess how SLC35D3 regulates the trafficking of NOTCH1, we detected the interaction between SLC35D3 and NOTCH1 in HEK293T cells via a coimmunoprecipitation (CoIP) assay. As shown in Figure 7E, NOTCH1 was coimmunoprecipitated with SLC35D3 in HEK293T cells. NOTCH1 proteins consist of a large extracellular domain (ECD), a transmembrane region, and a large NICD^{2,4}. Furthermore, we determined the specific domain to which SLC35D3 binds. First, the binding between SLC35D3 and NICD was determined in HEK293T cells transfected with both the SLC35D3 plasmid and NICD plasmid via immunofluorescence and CoIP assays. As shown in Figures 7F and S7A, neither coprecipitation of NICD protein with SLC35D3 nor cellular colocalization of SLC35D3 with NICD was observed. We then determined the binding between SLC35D3 and ECD. Myc-tagged NOTCH1 ECD was coimmunoprecipitated with SLC35D3 in HEK293T cells and cellular colocalization of SLC35D3 with NOTCH1 ECD was also observed (Fig. 7G, I). The direct physical interaction between SLC35D3 and the NOTCH1 ECD was further confirmed by an *in vitro* pull-down assay (Fig. 7H).

Taken together, these results revealed that SLC35D3 interacted with the NOTCH1 ECD in the cytoplasm and then retained NOTCH1 in the cytoplasm, especially in the ER, thereby inhibiting the translocation of NOTCH1 to the cell membrane and thus inhibiting NOTCH1 signaling.

All of the results supported our hypothesis. These are described in Figure 6, Figure 7 and Figure S7A of the revised manuscript, which are also shown below.

Figure 6. SLC35D3 inhibited the activation of NOTCH1 signaling.

(A) KEGG pathway enrichment analysis results of IngWAT from WT and SAKI mice. (B) GSEA results of Notch signaling pathway gene sets ([NES] -1.92, [ES] -0.38; FDR q 1.51E-06). (C, E) QPCR analysis of *Slc35d3* and Notch1-related genes in IngWAT from SAKI (n=6) mice and their WT (n=6) littermates (C), as well as in IngWAT from SAKI (n=6) mice and their WT (n=6) littermates (E). (D, F) Western blot showing the expression of Notch1-related proteins in IngWAT from SAKI (n=3) mice and their controls (n=3) (D), as well as in IngWAT from SAKO (n=3) mice and their controls

(n=3) (F). (G) qPCR detection of *NOTCH1* expression in HEK293T cells overexpressing *SLC35D3*. (H) Western blot analysis of NOTCH1 protein in HEK293T cells overexpressing *SLC35D3*. Right panel, summary of the quantification of three independent experiments. (I) Western blot analysis of protein in HEK293T cells treated with CHX. Values are the fold induction of gene expression normalized to the housekeeping gene β -Actin, and data are expressed as the mean \pm SEM. Statistical significance was assessed by two-sided Student's t test (C-I). Statistical significance was determined as $P < 0.05$.

Figure 7. SLC35D3 increased the ER accumulation of NOTCH1 by interacting with the NOTCH1 extracellular domain.

(A-B) Immunofluorescence assays revealed the increased intracellular accumulation of NOTCH1 in the ER in SLC35D3-overexpressing cells (A), as well as the reduced NOTCH1 in plasma (B). (C) NOTCH1 levels were downregulated in the cell plasma when SLC35D3 was overexpressed. Right panel, summary of the quantification of three independent experiments. (D) Cell-based NOTCH1 levels in SLC35D3-overexpressing HEK293T cells and control cells were detected by flow cytometric analysis. Binding curves were generated by determining the NOTCH1 MFI. Right panel, summary of the quantification of three independent experiments. (E-G) CoIP was performed to assess the interaction between SLC35D3 and NOTCH1 (E), SLC35D3 and NICD (F), and SLC35D3 and the NOTCH1 ECD (G). (H) Direct interaction between SLC35D3 and NOTCH1-ECD. The bait protein, purified GST-SLC35D3, was immobilized on glutathione affinity resin and incubated with MYC-ECD protein lysate at 4 °C for at least 1 hour with gentle rocking motion on a rotating platform. The prey protein, MYC-ECD, was captured by the bait protein, and then the bait-prey protein was eluted. ECD and SLC35D3 were subsequently analyzed by western blotting. (I) Colocalization of SLC35D3 and NOTCH1 ECD. Data are represented as the mean \pm SEM. Statistical significance was assessed by two-sided Student's t test (C, D) and was determined as $P<0.05$.

Figure S7.

A

(A) SLC35D3 and NICD did not colocalize.

Responses to Reviewer #3

1) while the authors have had some improvement in the overall writing, there are still multiple sentence structure problems, typos, etc, scattered throughout the paper, that leads to significant confusion when reading and trying to follow the scientific argument. Ongoing significant editing is needed.

Response: Thanks for your helpful suggestion. We apologize for the language mistakes, and we have re-examined and carefully revised the entire manuscript.

2) i still have concerns that no effect was seen in via Notch inhibition; while i agree that there may be an interaction between SLC35D3 and the ECD of Notch, whole-body treatment of mice with DBZ would have a direct effect on the liver as well as on adipose tissue; the liver-mediated improvements via Notch inhibition in whole body metabolism may outweigh the effects on adipose tissue. Adipose-specific targeting of Notch will be required to make the claims that all the benefits of Notch blockade by overexpression of SLC35D3 is at the level of adipose tissue changes. The improvements or damage by over-expression or KO of the protein might simply be tied to the weight effects.

If the purpose of the paper is to show that the effects of this protein are all mediated via the Notch cascade, the notch findings should not be relegated to supplemental data.

Response: Thanks for raising this point. We have supplemented experiments to study the impact of Notch1 knockdown in the IngWAT of SAKO and SAKI mice by in situ injection of AAV carrying shRNA targeting Notch1 (AAV8-adiponectin-shNotch1) or control shRNA. The results showed that knockdown of Notch1 in the IngWAT of SAKO mice exhibited a significant rescue effect, as shown by resistance to obesity and improved metabolism. In contrast, Notch1 knockdown showed only a slight improvement in adiposity and metabolism in SAKI mice. In SAKI mice, Notch1 signaling was suppressed by overexpressed Slc35d3, which was correlated with the

protective efficacy against obesity. This might explain why knockdown of Notch1 significantly improved metabolism in SAKO mice, but the effect was not evident in SAKI mice. These are presented in Figure 8 of the revised manuscript, which are also shown below.

In addition, to test whether Slc35d3 leads to Notch1 signaling alterations in the liver, we examined hepatic Notch1-related gene expression levels, which were similar between SAKO and control mice (Fig. S1H). Moreover, compared with control mice, the expression levels of Notch1-related genes were not significantly altered in either NCD-fed SAKI mice or HFD-fed SAKI mice (Fig. S4H, P). These data suggested that alterations of Slc35d3 in adipose tissue did not affect Notch1 signaling in the liver. These are presented in Figure S1H, Figure S4H and Figure S4P of the revised manuscript, which are also shown below.

Figure 8.

(A, K) Illustration of *Notch1* knockdown in the IngWAT of SAKO (A, n=4 per group) and SAKI (K, n=4 per group) mice. (B, L) *Notch1* was specifically knocked down in the IngWAT of SAKO (B) and SAKI (L) mice. (C, M) Body weights of NCD-fed SAKO mice (C) and HFD-fed SAKI mice (M). (D-E, N-O) GTT (D) and ITT (E) results of NCD-fed SAKO mice (n=4 per group), as well as GTT (N) and ITT (O) results of HFD-fed SAKI mice (n=4 per group). (F-G, P-Q) Serum TG and TC levels in NCD-fed

SAKO mice (F-G) and HFD-fed SAKI mice (P-Q). (H, R) Lipolytic gene expression levels of IngWAT in NCD-fed SAKO mice (H) and HFD-fed SAKI mice (R). (I, S) Thermogenic gene expression levels of IngWAT in NCD-fed SAKO mice (I) and HFD-fed SAKI mice (S). (J, T) HE staining results and adipocyte size quantification of IngWAT in NCD-fed SAKO mice (J) and HFD-fed SAKI mice (T). Values are the fold induction of gene expression normalized to the housekeeping gene β -Actin, and data are expressed as the mean \pm SEM. Statistical significance between two groups was assessed by two-sided Student's t test and was determined as $P < 0.05$.

Figure S1.

(H) Representative western blot for Slc35d3, Notch1, Hes1 and β -Actin protein in livers of SAKO (n=3) and Flox (n=3) mice. Right panel, summary of the quantification of three independent experiments.

Figure S4.

(H, P) Representative western blot for Slc35d3, Notch1, Hes1 and β -Actin protein in the liver of NCD-fed SAKI and WT mice (H), as well as HFD-fed SAKI and WT mice (P); n=3 per group. Right panel, summary of the quantification of three independent experiments.

3) the data still is consistent with much of the metabolic effects being tied specifically to the weight gain; with the new data of GTT of aged/weight matched pre-weight gain, the effects are blunted and rather minimal; only the 0 and 90 min time points are affected. I would think then that a majority of the outcomes observed are tied more to weight gain, esp. in light of the note that SLC35D3 is a UDP glucose transporter expressed in neurons, and that mutations lead to lower motor activity etc., and that it is classically on vesicular, not cell membranes

Response: Thank you for your helpful suggestion. To verify that weight gain does not affect the GTT/ITT in mice in our study, we included 5-week-old SAKO and control mice (8 vs. 8, male) for glucose tolerance and insulin sensitivity tests. Body weights were similar between SAKO and control mice at this time period. The results showed that glucose tolerance and insulin sensitivity were significantly impaired in 5-week-old SAKO mice. These results indicated that GTT and ITT phenotype differences in SAKO mice predated weight gain. The GTT and ITT results are shown below.

In addition, we have supplemented more data to support our statement on how Slc35d3 interacts with Notch1 to modulate downstream signaling. As we described in the revised manuscript, SLC35D3 interacted with the NOTCH1 ECD in the cytoplasm and then retained NOTCH1 in the cytoplasm, especially in the ER, thereby inhibiting the translocation of NOTCH1 to the cell membrane and thus inhibiting NOTCH1 signaling. These are described in Figure 6, Figure 7 and Figure S7A of the revised manuscript, which are also shown below.

(A) Blood glucose concentrations during IP-GTT in 5-week-old SAKO mice and their controls. The area under the curve (AUC) values of GTT are shown in the right panel. (B) Blood glucose concentrations during IP-ITT in 5-week-old SAKO mice and their controls. The AUC values of ITT are shown in the right panel.

Figure 6. SLC35D3 inhibited the activation of NOTCH1 signaling.

(A) KEGG pathway enrichment analysis results of IngWAT from WT and SAKI mice. (B) GSEA results of Notch signaling pathway gene sets ([NES] -1.92, [ES] -0.38; FDR q 1.51E-06). (C, E) QPCR analysis of *Slc35d3* and Notch1-related genes in IngWAT from SAKI (n=6) mice and their WT (n=6) littermates (C), as well as in IngWAT from SAKI (n=6) mice and their WT (n=6) littermates (E). (D, F) Western blot showing the expression of Notch1-related proteins in IngWAT from SAKI (n=3) mice and their controls (n=3) (D), as well as in IngWAT from SAKO (n=3) mice and their controls (n=3) (F).

(n=3) (F). (G) qPCR detection of *NOTCH1* expression in HEK293T cells overexpressing *SLC35D3*. (H) Western blot analysis of NOTCH1 protein in HEK293T cells overexpressing *SLC35D3*. Right panel, summary of the quantification of three independent experiments. (I) Western blot analysis of protein in HEK293T cells treated with CHX. Values are the fold induction of gene expression normalized to the housekeeping gene β -Actin, and data are expressed as the mean \pm SEM. Statistical significance was assessed by two-sided Student's t test (C-I). Statistical significance was determined as $P < 0.05$.

Figure 7. SLC35D3 increased the ER accumulation of NOTCH1 by interacting with the NOTCH1 extracellular domain.

(A-B) Immunofluorescence assays revealed the increased intracellular accumulation of NOTCH1 in the ER in SLC35D3-overexpressing cells (A), as well as the reduced NOTCH1 in plasma (B). (C) NOTCH1 levels were downregulated in the cell plasma when SLC35D3 was overexpressed. Right panel, summary of the quantification of three independent experiments. (D) Cell-based NOTCH1 levels in SLC35D3-overexpressing HEK293T cells and control cells were detected by flow cytometric analysis. Binding curves were generated by determining the NOTCH1 MFI. Right panel, summary of the quantification of three independent experiments. (E-G) CoIP was performed to assess the interaction between SLC35D3 and NOTCH1 (E), SLC35D3 and NICD (F), and SLC35D3 and the NOTCH1 ECD (G). (H) Direct interaction between SLC35D3 and NOTCH1-ECD. The bait protein, purified GST-SLC35D3, was immobilized on glutathione affinity resin and incubated with MYC-ECD protein lysate at 4 °C for at least 1 hour with gentle rocking motion on a rotating platform. The prey protein, MYC-ECD, was captured by the bait protein, and then the bait-prey protein was eluted. ECD and SLC35D3 were subsequently analyzed by western blotting. (I) Colocalization of SLC35D3 and NOTCH1 ECD. Data are represented as the mean ± SEM. Statistical significance was assessed by two-sided Student's t test (C, D) and was determined as $P < 0.05$.

Figure S7.

(A) SLC35D3 and NICD did not colocalize.

4) the methodology for glucose uptake is flawed; it should be a clamp study or the like with a nonmetabolizable analogue

Response: Thank you for your helpful suggestion. We supplemented hyperinsulinemic euglycemic clamps to further characterize insulin sensitivity in male SAKO mice and SAKI mice. We also examined 2-deoxyglucose uptake in the IngWAT. The results showed that the glucose infusion rate required to maintain euglycemia was significantly lower in SAKO mice (Fig. 2K-L). Whole-body insulin-stimulated glucose turnover was significantly downregulated (Fig. 2M), while 2-deoxyglucose uptake in the IngWAT of SAKO mice was also markedly reduced (Fig. 2N). However, the opposite was observed in SAKI mice (Fig. 4K-N). These are presented in Figure 2K-N and Figure 4K-N of the revised manuscript, which are also shown below.

Figure 2.

(K-N) Glucose infusion rates (K-L), whole-body insulin-stimulated glucose turnover (M) and 2-deoxyglucose uptake in IngWAT (N) during hyperinsulinemic euglycemic clamps of NCD-fed SAKO (n=5) and Flox (n=5) mice.

Figure 4.

(K-N) Glucose infusion rates (K-L), whole-body insulin-stimulated glucose turnover (M) and 2-deoxyglucose uptake in IngWAT (N) during hyperinsulinemic euglycemic clamps of HFD-fed SAKI (n=5) and WT (n=5) mice.

References

- 1 Zhou, B. *et al.* Notch signaling pathway: architecture, disease, and therapeutics. *Signal transduction and targeted therapy* **7**, 95, doi:10.1038/s41392-022-00934-y (2022).
- 2 Kopan, R. & Ilagan, M. X. The canonical Notch signaling pathway: unfolding the activation mechanism. *Cell* **137**, 216-233, doi:10.1016/j.cell.2009.03.045 (2009).
- 3 Siebel, C. & Lendahl, U. Notch Signaling in Development, Tissue Homeostasis, and Disease. *Physiol Rev* **97**, 1235-1294, doi:10.1152/physrev.00005.2017 (2017).
- 4 Rana, N. A. *et al.* O-glucose trisaccharide is present at high but variable stoichiometry at multiple sites on mouse Notch1. *J Biol Chem* **286**, 31623-31637, doi:10.1074/jbc.M111.268243 (2011).
- 5 Pelgrim, C. E. *et al.* Butyrate Reduces HFD-Induced Adipocyte Hypertrophy and Metabolic Risk Factors in Obese LDLr^{-/-}.Leiden Mice. *Nutrients* **9**, doi:10.3390/nu9070714 (2017).
- 6 Suarez-Zamorano, N. *et al.* Microbiota depletion promotes browning of white adipose tissue and reduces obesity. *Nat Med* **21**, 1497-1501, doi:10.1038/nm.3994 (2015).
- 7 Wu, Z. *et al.* CD146 is a Novel ANGPTL2 Receptor that Promotes Obesity by Manipulating Lipid Metabolism and Energy Expenditure. *Adv Sci (Weinh)* **8**, 2004032, doi:10.1002/advs.202004032 (2021).
- 8 Bi, P. *et al.* Inhibition of Notch signaling promotes browning of white adipose tissue and ameliorates obesity. *Nat Med* **20**, 911-918, doi:10.1038/nm.3615 (2014).

REVIEWER COMMENTS

Reviewer #1 (Remarks to the Author):

The authors have significantly revised the text of the previous version of the manuscript, added new pieces of data and tried to address all the reviewer's concerns. However, one set of experiments that, despite extensive work and revision, unfortunately remains inconclusive, are the IMF experiments.

Major comments

1 The immunofluorescence experiments shown in Figure 7A and B are still difficult to evaluate. First, although the western blot results depicted in panel B do indeed show an increase in cytoplasmic NOTCH1, no difference in signal intensity between control and SLC35D3 overexpressing cells is visible in the IF. Second, while in panel B there is a clearly visible decrease in membrane localized NOTCH1, it is unclear, why the staining for NOTCH1 looks different between panel A and B. In case the authors used a different staining protocol (e.g., no permeabilization) this should be indicated in the text or figure legend, as well as in the material and methods section. Finally, the figure legend of Figure 7 is poorly written and requires editing (e.g., "at the plasma membrane" instead of "in plasma").

2 While Figure 7I does show a co-localization between calnexin and the ECD, the staining pattern for SLC35D3 looks different and the areas with the strongest intensities, don't overlap for the SLC35D3 and the ECD staining as they however did, in the previous version of the manuscript. How would the authors explain this?

3. Western Blots in Figure 6D and 6F are (still) of poor quality and the from it resulting quantification (presumably by image analysis based on pixel intensity) is therefore not convincing to make all the claims.

Minor comments

1 The material and methods section is not up to date. Sections as for instance the description of the immunofluorescence or the cellular fractionation are missing.

2 In panel SF7 B, there is a discrepancy between IP and input. While in the input, it is indicated that the truncated SLC35D3 version was used (AA -30, 40 kDA), in the IP its states that the full version (50 kDA) is shown.

Reviewer #2 (Remarks to the Author):

The authors have made substantial changes to the manuscript and the model is now more in line with the data presented. Most of my concerns have been addressed by the authors. However, I am still not convinced by the data showing the relevance of SLC35D3 for NOTCH1 trafficking (Figure 7). Therefore, I have one major point that I think is very critical and needs to be answered by convincing experiments.

Major point: The claim that SLC35D3 increases ER accumulation of NOTCH is based on the data presented in Figure 7A and 7B. However, the IMF images and Western blots presented, including the statistical analysis, are not convincing, so the mechanism proposed by the authors is still not supported by the data presented.

Minor points: In many places in the text, the description of the data is still imprecise or even incorrect. For example, what is a cytomembrane (e.g. page 16)? Is NOTCH1 really reduced in plasma (legend to Figure 7)?

Reviewer #3 (Remarks to the Author):

The authors put forth an intriguing examination of the role of SLC35D3 in regulation of adipose tissue via the Notch cascade. They have revised this work several times, performed multiple new experiments to meet reviewers' comments/suggestions, and have significantly strengthened the paper. They found that SLC35d3 knockdown had decreased EE, worsened adipose expression of thermogenic genes, and worsened glucose tolerance; the opposite phenotype was noted in the overexpressors. Furthermore, they show relatively well that this protein seems to bind the Notch1 receptor ECL and hold in intracellularly, inhibiting Notch signaling and altering the browning phenotype. The findings are novel, the methodology is sound and much improved from the past. The abstract needs minor editing

(ie line 35 is unclear for "rescue of obesity" but other than that the work is noteworthy and worthy of publication.

Responses to Reviewer #1

The authors have significantly revised the text of the previous version of the manuscript, added new pieces of data and tried to address all the reviewer's concerns. However, one set of experiments that, despite extensive work and revision, unfortunately remains inconclusive, are the IMF experiments.

Major comments

1. The immunofluorescence experiments shown in Figure 7A and B are still difficult to evaluate. First, although the western blot results depicted in panel B do indeed show an increase in cytoplasmic NOTCH1, no difference in signal intensity between control and SLC35D3 overexpressing cells is visible in the IF. Second, while in panel B there is a clearly visible decrease in membrane localized NOTCH1, it is unclear, why the staining for NOTCH1 looks different between panel A and B. In case the authors used a different staining protocol (e.g., no permeabilization) this should be indicated in the text or figure legend, as well as in the material and methods section. Finally, the figure legend of Figure 7 is poorly written and requires editing (e.g., “at the plasma membrane” instead of “in plasma”).

Response: Thank you for your helpful comments. We apologize for the poor quality of the fluorescence results shown in the figure and the poorly written legends. For Figure 7A in the previous manuscript (Figure S7E in the revised manuscript), cells were permeabilized with 0.3% Triton X-100 for 20 min at room temperature in order to label the ER marker CALNEXIN in the cytoplasm. To adequately detect NOTCH1 proteins in the cytoplasm, we used a high antibody dilution (CST, 4380S, 1:80) for NOTCH1 proteins staining, which might be somewhat too high resulting in a strong fluorescence background, and eventually NOTCH1 fluorescence did not show a difference between control and SLC35D3 overexpressing cells. For Figure 7B in the previous manuscript (Figure S7F in the revised manuscript), cells were not permeabilized in order to detect cell membrane surface proteins, which might make the fluorescence results of panel A and panel B in the previous manuscript look different. Relevant descriptions are described in the Materials and Methods section, as well as in the figure legends (Figure

S7) of the revised manuscript.

In addition, to further confirm the regulatory effect of SLC35D3 on NOTCH1 and to minimize the differences in fluorescence staining results between different batches of experiments, in supplementary experiments, we uniformly used the Opal™ 7 color kit (Akoya Biosciences, NEL811001KT) to perform multiple fluorescent staining of cells according to the recommended procedures. New data and representative images are provided in the revised manuscript. Combined with the results of previous experiments, these further confirmed our hypothesis that SLC35D3 promoted the ER accumulation of NOTCH1 by interacting with the NOTCH1 extracellular domain (ECD), which in turn inhibited NOTCH1 signaling. Furthermore, we have examined and modified the relevant legends in detail.

All of the results support our claim that SLC35D3 promotes ER accumulation of NOTCH1. These are described in detail in the Results section of the revised manuscript and data are shown in Figure 7, Figure S7 and Figure S8, which are also shown below.

Results

7. SLC35D3 increased the ER accumulation of NOTCH1 via interaction with the NOTCH1 extracellular domain.

The above results further aroused our curiosity about the exact pathway through which SLC35D3 inhibited NOTCH1 signaling. Next, we examined the localization of SLC35D3 and NOTCH1 by immunofluorescence in HEK293T cells overexpressing SLC35D3. The results were very intriguing, and we found that SLC35D3 and NOTCH1 colocalized intracellularly (Fig. S7A), suggesting a possible interaction between SLC35D3 and NOTCH1. Subsequently, we detected the interaction between SLC35D3 and NOTCH1 in HEK293T cells via a coimmunoprecipitation (CoIP) assay. As shown in Figure S7B, NOTCH1 was coimmunoprecipitated with SLC35D3 in HEK293T cells (Fig. S7B). NOTCH1 proteins consist of a large extracellular domain (ECD), a transmembrane region, and a large NICD^{1,2}. Furthermore, we determined the specific domain to which SLC35D3 binds. First, the binding between SLC35D3 and NICD was determined in HEK293T cells transfected with both the SLC35D3 plasmid and NICD plasmid via CoIP assays. As shown in Figure S7C, no coprecipitation of NICD with SLC35D3 was observed (Fig. S7C). We then determined the binding between SLC35D3 and ECD, and myc-tagged NOTCH1 ECD was coimmunoprecipitated with SLC35D3 in HEK293T cells (Fig. 7A). The direct physical interaction between SLC35D3 and the NOTCH1 ECD was further confirmed by an in vitro pull-down assay (Fig. 7B). Interestingly, SLC35D3 lacking the last 30 amino acids (SLC35D3-30aa) remained immunoprecipitated with the NOTCH1 ECD (Fig. S7D). Taken together,

these results revealed that SLC35D3 interacted with NOTCH1 through ECD, the extracellular domain of NOTCH1.

We next sought to investigate how the interaction of SLC35D3 with NOTCH1 affected NOTCH1 signaling. In signal-receiving cells, NOTCH1 receptors are first generated in the ER and then transported to the Golgi apparatus where NOTCH1 receptors are cleaved into heterodimers (S1 cleavage) and transported to the cell membrane. Transport of NOTCH1 in the endoplasmic reticulum (ER) has been shown to play a critical role in the regulation of NOTCH1 levels at the cell membrane²⁻⁴. Considering that SLC35D3 also localizes to the ER intracellularly⁵, we speculated that SLC35D3 regulated the trafficking of NOTCH1 from ER to the plasma membrane and detected the subcellular localization of NOTCH1 in the ER and on the plasma membrane. Interestingly, the levels of NOTCH1 on the plasma membrane were decreased in HEK293T cells overexpressing SLC35D3, while most NOTCH1 showed colocalization with CALNEXIN, an marker of ER (Fig. S7E-F). Consistently, western blot and flow cytometric assays also showed decreased levels of NOTCH1 on the plasma membrane (Fig. S7G-J).

These results suggested that SLC35D3 might regulate NOTCH1 trafficking. To further investigate the modulatory effect of SLC35D3 on NOTCH1, HEK293T cells were transfected with both SLC35D3 plasmid and GFP-tagged NOTCH1 plasmid, and cells transfected with both empty plasmid and GFP-tagged NOTCH1 plasmid were used as controls. 48 hours after transfection, we labeled the ER in living cells using ER Tracker Red Dyes and detected NOTCH1 localization in living cells by a High Content Imaging System (Opera Phenix, PerkinElmer). Intriguingly, NOTCH1 did not show extensive colocalization with ER in control cells; instead, NOTCH1 was more distributed at the periphery of ER, surrounding the cell. However, in cells overexpressing SLC35D3, there was extensive colocalization of NOTCH1 with ER, indicating that overexpression of SLC35D3 promoted NOTCH1 colocalization with ER (Fig. 7C, Fig. S8A).

We then performed multiple fluorescent staining in fixed cells using an Opal multiplex staining protocol with Opal™ 7 color kit. HEK293T cells were transfected with SLC35D3 plasmid and NOTCH1 plasmid, and we then examined the subcellular localization of SLC35D3 and NOTCH1 in cells 24h and 48h post-transfection. The results showed that little NOTCH1 and ER colocalization was observed in both control and SLC35D3 overexpressing cells 24 h post-transfection; however, compared with controls, the overexpression of SLC35D3 promoted NOTCH1 and ER colocalization 48 h after transfection (Fig. S8B). We further transfected HEK293T cells with SLC35D3 plasmid to mediate overexpression of SLC35D3 and siRNA targeting SLC35D3 to mediate knockdown of SLC35D3. Next, we examined the intracellular localization of NOTCH1 48 h post-transfection. Immunofluorescence results showed that overexpression of SLC35D3 promoted NOTCH1 retention in ER, as shown by increased NOTCH1 colocalization with ER in SLC35D3 overexpressing cells (Fig. 7D). SLC35D3 knockdown led to the opposite results, as revealed by decreased colocalization of NOTCH1 with ER (Fig. 7E). We have shown in the results above that SLC35D3 interacts with the ECD of NOTCH1, but not with NICD. We subsequently

transfected HEK293T cells with both SLC35D3 plasmid and ECD plasmid, as well as SLC35D3 plasmid and NICD plasmid. Multiple fluorescence staining results showed that SLC35D3 and ECD colocalized and both colocalized with ER; in contrast, NICD did not colocalize with SLC35D3 (Fig. S8C). This further confirmed the interaction between SLC35D3 and ECD, which was mainly in ER.

To further confirm that SLC35D3 promoted NOTCH1 retention in the ER, we isolated cell membrane proteins and endoplasmic reticulum proteins using Plasma Membrane Protein Isolation and Cell Fractionation Kit and Endoplasmic Reticulum Isolation Kit, and detected NOTCH1 levels by western blotting assays. HEK293T cells were transfected with SLC35D3 plasmid and siRNA targeting SLC35D3, and harvested at different time points to detect NOTCH1 protein levels. The results were intriguing, showing that overexpression of SLC35D3 did not alter total NOTCH1 protein levels (Fig. 7F-G). However, SLC35D3 overexpression significantly promoted the retention of NOTCH1 in the ER, accompanied by a decrease in NOTCH1 protein levels on the cell membrane, as shown by elevated NOTCH1 levels in the ER and decreased NOTCH1 levels on the cell membrane at 40h and 48h (Fig. 7F-I). In contrast, knockdown of SLC35D3 promoted trafficking of NOTCH1 protein from ER to cell membrane, as shown by decreased NOTCH1 levels in the ER and elevated NOTCH1 levels on the cell membrane at 48 h (Fig. S8D-G).

Summarizing all the results above, SLC35D3 increased ER accumulation of NOTCH1 and thereby inhibited NOTCH1 signaling, which was achieved by interacting with NOTCH1 extracellular domain.

Figure 7

(A) CoIP assay results showed the interaction between SLC35D3 and ECD. (B) Direct interaction between SLC35D3 and ECD was further confirmed by pull down assay. (C) The live cell imaging results obtained using a High Content Imaging System. Representative images are shown, and the inset in the right panel shows a magnified view of the indicated area. Scale bar = 20 μ m. (D-E) SLC35D3 overexpression promoted colocalization of NOTCH1 and ER (D), while knockdown of SLC35D3 reduced colocalization of NOTCH1 and ER (E). Representative confocal images are shown, and the inset in the right panel shows a magnified view of the indicated area. Scale bar = 25 μ m. CALNEXIN, an ER marker protein. (F-I) Western blotting results (F) of total NOTCH1, membrane NOTCH1, and ER NOTCH1 proteins at different time points in control and SLC35D3 overexpressing cells. The quantitative analysis results of three independent experiments (G-I) are shown. Total NOTCH1 expression was normalized to β -ACTIN expression, membrane NOTCH1 expression was normalized to Na-K ATPase, and ER NOTCH1 expression was normalized to CALNEXIN.

Figure S7

(A) In SLC35D3 overexpressing cells, SLC35D3 and NOTCH1 colocalized. Representative confocal images are shown, and the inset in the bottom panel shows a magnified view of the indicated area. Scale bar = 25 μ m. (B, D) CoIP assay results showed the interaction between SLC35D3 and NOTCH1 (B), as well as the interaction between SLC35D3-30aa and ECD (D). (C) CoIP assay results showed no interaction between SLC35D3 and NICD. (E-F) Immunofluorescence assays showed the subcellular localization of NOTCH1 in control and SLC35D3 overexpressing cells (cells in the panel E were permeabilized, whereas cells in the panel F were not). Overexpression of SLC35D3 reduced the colocalization of NOTCH1 and plasma

membrane (F). CALNEXIN, an ER marker protein; Na-K ATPase, a plasma membrane marker protein. Scale bar = 25 μ m. (G-H) NOTCH1 levels were downregulated in the plasma membrane when SLC35D3 was overexpressed. The quantitative analysis results of three independent experiments (H) are shown. (I-J) Membrane NOTCH1 levels in SLC35D3 overexpressing HEK293T cells and control cells were detected by flow cytometric analysis. Binding curves were generated by determining the NOTCH1 mean fluorescence density (MFI). The quantitative analysis results of three independent experiments (J) are shown.

Figure S8
(A) SLC35D3 overexpression promoted NOTCH1 colocalization with ER. Representative images are shown. Scale bar = 20 μm . (B) The multiplex fluorescence staining results of cells at 24h and 48h post-transfection. SLC35D3 overexpression promoted colocalization of NOTCH1 and ER at 48h. Representative confocal images are shown. Scale bar = 50 μm . CALNEXIN, an ER marker protein. (C) The multiplex fluorescence staining results of cells transfected with SLC35D3 plasmid and ECD plasmid, as well as cells transfected with SLC35D3 plasmid and NICD plasmid. SLC35D3 and ECD colocalized and both colocalized with ER, while SLC35D3 and NICD did not colocalize. Representative confocal images are shown, and the inset in the bottom panel shows a magnified view of the indicated area. Scale bar = 50 μm . (D-G) Western blotting results (F) of total NOTCH1, membrane NOTCH1, and ER NOTCH1 proteins at different time points in control and SLC35D3 knockdown cells. The quantitative analysis results of three independent experiments (E-G) are shown.

2. While Figure 7I does show a co-localization between calnexin and the ECD, the staining pattern for SLC35D3 looks different and the areas with the strongest intensities, don't overlap for the SLC35D3 and the ECD staining as they however did, in the previous version of the manuscript. How would the authors explain this?

Response: Thank you for your helpful suggestion, and we apologize for the confusion. In the previous version of the manuscript, cells were transfected with his-tagged SLC35D3 plasmid and myc-tagged ECD plasmid, and were subsequently stained with rabbit anti-SLC35D3 antibody (Thermo, PA5-72721) and rat anti-myc antibody (Abcam, ab206486) following standard staining steps including permeabilization. However, in the last version of the manuscript, cells were transfected with his-tagged SLC35D3 plasmid and myc-tagged ECD plasmid, and the associated fluorescent staining was performed by another investigator blinded to the grouping information who performed the staining shown in panel B and panel I in the same batch of experiments. Similar to staining in panel B, cells were not permeabilized. In addition, cells were stained with rabbit anti-his antibody (CST, 12698S), mouse anti-Calnexin antibody (Thermo, MA3-027) and rat anti-myc antibody (Abcam, ab206486). We apologize again for the differences in the relevant fluorescence results caused by different experimental steps and for not giving a detailed description.

In the revised manuscript, cells were transfected with myc-tagged ECD plasmid and his-tagged SLC35D3 plasmid, as well as myc-tagged NICD plasmid and his-tagged SLC35D3 plasmid. To minimize the differences in fluorescence staining results between different batches of experiments, in supplementary experiments, the Opal™ 7 color kit (Akoya Biosciences, NEL811001KT) was used to perform multiple fluorescent staining of cells according to the recommended procedures by an investigator who was blinded to the grouping information. Three antibodies including rabbit anti-SLC35D3 antibody (Thermo, PA5-72721), mouse anti-Calnexin antibody (Thermo, MA3-027) and mouse anti-myc antibody (CST, 2276S) were used. The staining results showed that SLC35D3 and ECD colocalized and both colocalized with CALNEXIN, an ER marker. However, there was no colocalization between NICD and

SLC35D3.

These are presented in Figure S8C of the revised manuscript, which are also shown below.

Figure S8

(C) The multiplex fluorescence staining results of cells transfected with SLC35D3 plasmid and ECD plasmid, as well as cells transfected with SLC35D3 plasmid and NICD plasmid. SLC35D3 and ECD colocalized and both colocalized with ER, while SLC35D3 and NICD did not colocalize. Representative confocal images are shown, and the inset in the bottom panel shows a magnified view of the indicated area. Scale bar = 50 μ m.

3. Western Blots in Figure 6D and 6F are (still) of poor quality and the from it resulting quantification (presumably by image analysis based on pixel intensity) is therefore not convincing to make all the claims.

Response: Thank you for your helpful comments. We apologize for the poor quality of western blots in Figure 6D and 6F. We have repeated western blotting assays and more representative experimental results are shown. An investigator blinded to group information redid the quantitative analysis of the results of three independent experiments. These new data are presented in Figure 6D and Figure 6F of the revised manuscript. In addition, for other western blotting results with poor quality (Figure 6H, Figure 7E and Figure S7B in the previous manuscript), we also repeated western blotting assays and new representative results are shown.

These are presented in Figure 6D, Figure 6F, Figure 6H, Figure S7B and Figure S7D of the revised manuscript, which are also shown below.

Figure 6

(D, F) Western blot results showing the expression of Notch1-related proteins in IngWAT from SAKI (n=3) mice and their controls (n=3) (D), as well as in IngWAT from SAKO (n=3) mice and their controls (n=3) (F). (H) Western blot analysis of NOTCH1 protein in HEK293T cells overexpressing *SLC35D3*. The quantitative analysis results of three independent experiments are shown in the right panel.

Figure S7

(B, D) CoIP assay results showed the interaction between *SLC35D3* and NOTCH1 (B), as well as the interaction between *SLC35D3*-30aa and ECD (D).

Minor comments

1. The material and methods section is not up to date. Sections as for instance the description of the immunofluorescence or the cellular fractionation are missing.

Response: Thank you for your helpful suggestion. We have updated the Materials and

Methods section in the revised manuscript and supplemented relevant descriptions regarding experimental methods. These are presented in the Materials and Methods section of the revised manuscript, which are also shown below.

Materials and Methods

Cell culture

HEK293T cells were cultured in DMEM (Gibco, Rockford) with 10% fetal bovine serum (FBS), 100 U/ml penicillin and 100 µg/ml streptomycin (P/S). Cells were used between passages 8 and 10. The cells were incubated at 37 °C (with 5% CO₂) until confluent and then transferred to 6-, 12- or 24-well plates. Knockdown of SLC35D3 was mediated by siRNA transfection using RNAiMAX (Invitrogen, Rockford, IL) transfection reagent (the sequences of siRNA: sense 5'-CCA UGU ACG UGG UCU UCA ATT -3'; antisense 5'-UUG AAG ACC ACG UAC AUG GTT -3'). Overexpression of SLC35D3 was achieved by his-tagged SLC35D3 plasmid transfection using Lipofectamine 3000 (Invitrogen, Rockford, IL) transfection reagent, and overexpression of different domains of NOTCH1, ECD and NICD, were achieved by myc-tagged ECD plasmid and myc-tagged NICD plasmid transfection. Transfection was performed when the degree of cell fusion reached 70-80%. After incubation for the appropriate time, the cells were harvested for further experiments.

Live-cell imaging

HEK293T cells were cultured in black 96-well plates (PerkinElmer) and transfected with SLC35D3 plasmid or empty vector plasmid, along with NOTCH1-GFP plasmid expressing NOTCH1-GFP fusion proteins. 48 hours after transfection, cells were treated with ER Tracker Red Dyes (Invitrogen, E34250) to label ER in living cells according to the manufacturer's instructions, and Hoechst 33342 (Thermo Scientific, 62249) was then used to label cell nuclei. Following labeling, a High Content Imaging System (Opera Phenix, PerkinElmer) was used to perform live-cell imaging.

Immunofluorescence staining

For the fluorescence results shown in Figure S7E and Figure S7F, immunofluorescence staining was performed following standard staining procedures. Briefly, cells were fixed with 4% paraformaldehyde for 20-30 min at 37 °C. For ER marker CALNEXIN staining, cells were permeabilized with 0.3% Triton X-100 for 20 min at room temperature; for cell membrane marker Na-K ATPase staining, cells were not permeabilized. Following blocking with goat serum for 1 h, the cells were incubated with appropriate primary antibodies overnight at 4 °C. Alexa Fluor 488-conjugated and Alexa Fluor 594-conjugated secondary antibodies were then used. DAPI was used to label cell nuclei. Confocal imaging was performed using a Leica SP8 confocal microscope. For all other immunofluorescent staining, the Opal™ 7 color kit (Akoya Biosciences, NEL811001KT) was used to perform multiple fluorescent staining of cells according to the recommended procedures. Both ECD and NICD were stained with antibodies to the fusion-expressed myc proteins, and the other proteins were fluorescently stained with their own antibodies. The antibodies used are shown in Table S1.

Plasma membrane protein and endoplasmic reticulum protein extraction

For plasma membrane protein and ER protein extraction, cells were harvested at various time points after transfection and washed with ice-cold PBS. Then, isolation and extraction of plasma membrane proteins were performed by a spin column method

using the Plasma Membrane Protein Isolation and Cell Fractionation Kit (Invent Biotechnologies, Inc, SM-005) according to the manufacturer's instructions, while the Endoplasmic Reticulum Isolation Kit (Sigma-Aldrich, ER0100) was used for isolation and extraction of ER proteins following the recommended procedures.

2. In panel SF7 B, there is a discrepancy between IP and input. While in the input, it is indicated that the truncated SLC35D3 version was used (AA -30, 40 kDa), in the IP its states that the full version (50 kDa) is shown.

Response: Thank you for your helpful suggestion. We apologize for the mislabeled information. We have checked the relevant experimental results and found that it was indeed a labeling error. We have corrected the associated mislabeled information. Furthermore, we have repeated the relevant western blotting assays and western blotting results with better quality are shown. These are presented in Figure S7D of the revised manuscript, which are also shown below.

Figure S7D

D

(D) CoIP assay results showed the interaction between SLC35D3-30aa and ECD.

We greatly appreciate your helpful and valuable comments, and we would like to thank you again for your precious time and efforts in reviewing the revised version of our manuscript.

Responses to Reviewer #2

The authors have made substantial changes to the manuscript and the model is now more in line with the data presented. Most of my concerns have been addressed by the authors. However, I am still not convinced by the data showing the relevance of SLC35D3 for NOTCH1 trafficking (Figure 7). Therefore, I have one major point that I think is very critical and needs to be answered by convincing experiments.

Major point: The claim that SLC35D3 increases ER accumulation of NOTCH is based on the data presented in Figure 7A and 7B. However, the IMF images and Western blots presented, including the statistical analysis, are not convincing, so the mechanism proposed by the authors is still not supported by the data presented.

Response: Thank you for your helpful suggestion. Recognizing the shortcomings of our previous studies, we supplemented new experiments and data to support our hypothesis that SLC35D3 increased ER accumulation of NOTCH1, including new live-cell imaging results, new immunofluorescence results, and ER protein extraction and analysis results. New data are presented in the revised manuscript.

All of the results support our claim that SLC35D3 promotes ER accumulation of NOTCH1. These are described in detail in the Results section of the revised manuscript and data are shown in Figure 7, Figure S7 and Figure S8, which are also shown below.

Results

7. SLC35D3 increased the ER accumulation of NOTCH1 via interaction with the NOTCH1 extracellular domain.

The above results further aroused our curiosity about the exact pathway through which SLC35D3 inhibited NOTCH1 signaling. Next, we examined the localization of SLC35D3 and NOTCH1 by immunofluorescence in HEK293T cells overexpressing SLC35D3. The results were very intriguing, and we found that SLC35D3 and NOTCH1 colocalized intracellularly (Fig. S7A), suggesting a possible interaction between SLC35D3 and NOTCH1. Subsequently, we detected the interaction between SLC35D3 and NOTCH1 in HEK293T cells via a coimmunoprecipitation (CoIP) assay. As shown in Figure S7B, NOTCH1 was coimmunoprecipitated with SLC35D3 in HEK293T cells (Fig. S7B). NOTCH1 proteins consist of a large extracellular domain (ECD), a transmembrane region, and a large NICD^{1,2}. Furthermore, we determined the specific domain to which SLC35D3 binds. First, the binding between SLC35D3 and NICD was

determined in HEK293T cells transfected with both the SLC35D3 plasmid and NICD plasmid via CoIP assays. As shown in Figure S7C, no coprecipitation of NICD with SLC35D3 was observed (Fig. S7C). We then determined the binding between SLC35D3 and ECD, and myc-tagged NOTCH1 ECD was coimmunoprecipitated with SLC35D3 in HEK293T cells (Fig. 7A). The direct physical interaction between SLC35D3 and the NOTCH1 ECD was further confirmed by an in vitro pull-down assay (Fig. 7B). Interestingly, SLC35D3 lacking the last 30 amino acids (SLC35D3-30aa) remained immunoprecipitated with the NOTCH1 ECD (Fig. S7D). Taken together, these results revealed that SLC35D3 interacted with NOTCH1 through ECD, the extracellular domain of NOTCH1.

We next sought to investigate how the interaction of SLC35D3 with NOTCH1 affected NOTCH1 signaling. In signal-receiving cells, NOTCH1 receptors are first generated in the ER and then transported to the Golgi apparatus where NOTCH1 receptors are cleaved into heterodimers (S1 cleavage) and transported to the cell membrane. Transport of NOTCH1 in the endoplasmic reticulum (ER) has been shown to play a critical role in the regulation of NOTCH1 levels at the cell membrane²⁻⁴. Considering that SLC35D3 also localizes to the ER intracellularly⁵, we speculated that SLC35D3 regulated the trafficking of NOTCH1 from ER to the plasma membrane and detected the subcellular localization of NOTCH1 in the ER and on the plasma membrane. Interestingly, the levels of NOTCH1 on the plasma membrane were decreased in HEK293T cells overexpressing SLC35D3, while most NOTCH1 showed colocalization with CALNEXIN, an marker of ER (Fig. S7E-F). Consistently, western blot and flow cytometric assays also showed decreased levels of NOTCH1 on the plasma membrane (Fig. S7G-J).

These results suggested that SLC35D3 might regulate NOTCH1 trafficking. To further investigate the modulatory effect of SLC35D3 on NOTCH1, HEK293T cells were transfected with both SLC35D3 plasmid and GFP-tagged NOTCH1 plasmid, and cells transfected with both empty plasmid and GFP-tagged NOTCH1 plasmid were used as controls. 48 hours after transfection, we labeled the ER in living cells using ER Tracker Red Dyes and detected NOTCH1 localization in living cells by a High Content Imaging System (Opera Phenix, PerkinElmer). Intriguingly, NOTCH1 did not show extensive colocalization with ER in control cells; instead, NOTCH1 was more distributed at the periphery of ER, surrounding the cell. However, in cells overexpressing SLC35D3, there was extensive colocalization of NOTCH1 with ER, indicating that overexpression of SLC35D3 promoted NOTCH1 colocalization with ER (Fig. 7C, Fig. S8A).

We then performed multiple fluorescent staining in fixed cells using an Opal multiplex staining protocol with Opal™ 7 color kit. HEK293T cells were transfected with SLC35D3 plasmid and NOTCH1 plasmid, and we then examined the subcellular localization of SLC35D3 and NOTCH1 in cells 24h and 48h post-transfection. The results showed that little NOTCH1 and ER colocalization was observed in both control and SLC35D3 overexpressing cells 24 h post-transfection; however, compared with controls, the overexpression of SLC35D3 promoted NOTCH1 and ER colocalization 48 h after transfection (Fig. S8B). We further transfected HEK293T cells with

SLC35D3 plasmid to mediate overexpression of SLC35D3 and siRNA targeting SLC35D3 to mediate knockdown of SLC35D3. Next, we examined the intracellular localization of NOTCH1 48 h post-transfection. Immunofluorescence results showed that overexpression of SLC35D3 promoted NOTCH1 retention in ER, as shown by increased NOTCH1 colocalization with ER in SLC35D3 overexpressing cells (Fig. 7D). SLC35D3 knockdown led to the opposite results, as revealed by decreased colocalization of NOTCH1 with ER (Fig. 7E). We have shown in the results above that SLC35D3 interacts with the ECD of NOTCH1, but not with NICD. We subsequently transfected HEK293T cells with both SLC35D3 plasmid and ECD plasmid, as well as SLC35D3 plasmid and NICD plasmid. Multiple fluorescence staining results showed that SLC35D3 and ECD colocalized and both colocalized with ER; in contrast, NICD did not colocalize with SLC35D3 (Fig. S8C). This further confirmed the interaction between SLC35D3 and ECD, which was mainly in ER.

To further confirm that SLC35D3 promoted NOTCH1 retention in the ER, we isolated cell membrane proteins and endoplasmic reticulum proteins using Plasma Membrane Protein Isolation and Cell Fractionation Kit and Endoplasmic Reticulum Isolation Kit, and detected NOTCH1 levels by western blotting assays. HEK293T cells were transfected with SLC35D3 plasmid and siRNA targeting SLC35D3, and harvested at different time points to detect NOTCH1 protein levels. The results were intriguing, showing that overexpression of SLC35D3 did not alter total NOTCH1 protein levels (Fig. 7F-G). However, SLC35D3 overexpression significantly promoted the retention of NOTCH1 in the ER, accompanied by a decrease in NOTCH1 protein levels on the cell membrane, as shown by elevated NOTCH1 levels in the ER and decreased NOTCH1 levels on the cell membrane at 40h and 48h (Fig. 7F-I). In contrast, knockdown of SLC35D3 promoted trafficking of NOTCH1 protein from ER to cell membrane, as shown by decreased NOTCH1 levels in the ER and elevated NOTCH1 levels on the cell membrane at 48 h (Fig. S8D-G).

Summarizing all the results above, SLC35D3 increased ER accumulation of NOTCH1 and thereby inhibited NOTCH1 signaling, which was achieved by interacting with NOTCH1 extracellular domain.

Figure 7
(A) CoIP assay results showed the interaction between SLC35D3 and ECD. (B) Direct interaction between SLC35D3 and ECD was further confirmed by pull down assay. (C) The live cell imaging results obtained using a High Content Imaging System. Representative images are shown, and the inset in the right panel shows a magnified view of the indicated area. Scale bar = 20 μ m. (D-E) SLC35D3 overexpression promoted colocalization of NOTCH1 and ER (D), while knockdown of SLC35D3 reduced colocalization of NOTCH1 and ER (E). Representative confocal images are shown, and the inset in the right panel shows a magnified view of the indicated area. Scale bar = 25 μ m. CALNEXIN, an ER marker protein. (F-I) Western blotting results (F) of total NOTCH1, membrane NOTCH1, and ER NOTCH1 proteins at different time points in control and SLC35D3 overexpressing cells. The quantitative analysis results of three independent experiments (G-I) are shown. Total NOTCH1 expression was normalized to β -ACTIN expression, membrane NOTCH1 expression was normalized to Na-K ATPase, and ER NOTCH1 expression was normalized to CALNEXIN.

Figure S7

(A) In SLC35D3 overexpressing cells, SLC35D3 and NOTCH1 colocalized. Representative confocal images are shown, and the inset in the bottom panel shows a magnified view of the indicated area. Scale bar = 25 μ m. (B, D) CoIP assay results showed the interaction between SLC35D3 and NOTCH1 (B), as well as the interaction between SLC35D3-30aa and ECD (D). (C) CoIP assay results showed no interaction between SLC35D3 and NICD. (E-F) Immunofluorescence assays showed the subcellular localization of NOTCH1 in control and SLC35D3 overexpressing cells (cells in the panel E were permeabilized, whereas cells in the panel F were not). Overexpression of SLC35D3 reduced the colocalization of NOTCH1 and plasma

membrane (F). CALNEXIN, an ER marker protein; Na-K ATPase, a plasma membrane marker protein. Scale bar = 25 μ m. (G-H) NOTCH1 levels were downregulated in the plasma membrane when SLC35D3 was overexpressed. The quantitative analysis results of three independent experiments (H) are shown. (I-J) Membrane NOTCH1 levels in SLC35D3-overexpressing HEK293T cells and control cells were detected by flow cytometric analysis. Binding curves were generated by determining the NOTCH1 mean fluorescence density (MFI). The quantitative analysis results of three independent experiments (J) are shown.

Figure S8

(A) SLC35D3 overexpression promoted NOTCH1 colocalization with ER. Representative images are shown. Scale bar = 20 μ m. (B) The multiplex fluorescence staining results of cells at 24h and 48h post-transfection. SLC35D3 overexpression promoted colocalization of NOTCH1 and ER at 48h. Representative confocal images are shown. Scale bar = 50 μ m. CALNEXIN, an ER marker protein. (C) The multiplex fluorescence staining results of cells transfected with SLC35D3 plasmid and ECD plasmid, as well as cells transfected with SLC35D3 plasmid and NICD plasmid. SLC35D3 and ECD colocalized and both colocalized with ER, while SLC35D3 and NICD did not colocalize. Representative confocal images are shown, and the inset in the bottom panel shows a magnified view of the indicated area. Scale bar = 50 μ m. (D-G) Western blotting results (F) of total NOTCH1, membrane NOTCH1, and ER NOTCH1 proteins at different time points in control and SLC35D3 knockdown cells. The quantitative analysis results of three independent experiments (E-G) are shown.

Minor points: In many places in the text, the description of the data is still imprecise or even incorrect. For example, what is a cytomembrane (e.g. page 16)? Is NOTCH1 really reduced in plasma (legend to Figure 7)?

Response: Thank you for your helpful suggestion, and we apologize for the imprecise or even incorrect description that remains. We have re-examined the entire manuscript in detail and carefully revised the imprecise or even incorrect descriptions. We apologize again for these poorly written words.

We greatly appreciate your helpful and valuable comments, and we would like to thank you again for your precious time and efforts in reviewing the revised version of our manuscript.

Responses to Reviewer #3

The authors put forth an intriguing examination of the role of SLC35D3 in regulation of adipose tissue via the Notch cascade. They have revised this work several times, performed multiple new experiments to meet reviewers' comments/suggestions, and have significantly strengthened the paper. They found that SLC35d3 knockdown had decreased EE, worsened adipose expression of thermogenic genes, and worsened glucose tolerance; the opposite phenotype was noted in the overexpressors. Furthermore, they show relatively well that this protein seems to bind the Notch1 receptor ECL and hold in intracellularly, inhibiting Notch signaling and altering the browning phenotype. The findings are novel, the methodology is sound and much improved from the past. The abstract needs minor editing (ie line 35 is unclear for "rescue of obesity" but other than that the work is noteworthy and worthy of publication.

Response: Thank you for your helpful suggestion. We apologize for the imprecise descriptions that still exist. We have re-examined the entire manuscript in detail and carefully revised the imprecise descriptions.

We greatly appreciate your kind consideration, and we would like to thank you again for your precious time and efforts in reviewing our manuscript.

References

- 1 Rana, N. A. *et al.* O-glucose trisaccharide is present at high but variable stoichiometry at multiple sites on mouse Notch1. *J Biol Chem* **286**, 31623-31637, doi:10.1074/jbc.M111.268243 (2011).
- 2 Kopan, R. & Ilagan, M. X. The canonical Notch signaling pathway: unfolding the activation mechanism. *Cell* **137**, 216-233, doi:10.1016/j.cell.2009.03.045 (2009).
- 3 Zhou, B. *et al.* Notch signaling pathway: architecture, disease, and therapeutics. *Signal transduction and targeted therapy* **7**, 95, doi:10.1038/s41392-022-00934-y (2022).
- 4 Siebel, C. & Lendahl, U. Notch Signaling in Development, Tissue Homeostasis, and Disease. *Physiol Rev* **97**, 1235-1294, doi:10.1152/physrev.00005.2017 (2017).
- 5 Zhang, Z. *et al.* Mutation of SLC35D3 causes metabolic syndrome by impairing dopamine signaling in striatal D1 neurons. *PLoS genetics* **10**, e1004124, doi:10.1371/journal.pgen.1004124 (2014).

REVIEWER COMMENTS

Reviewer #1 (Remarks to the Author):

This manuscript has been reviewed and adapted extensively multiple times and has improved. The authors have put additional effort in demonstrating the physical interaction between the ECD of NOTCH that results in the protein's retention in the ER. The quality of IF, as well as pull down and western blot experiments has significantly improved compared to the last version. In addition, important control experiments, such as knockdown of SLC35D3 have been added. In conclusion, the new data do relatively well show a retention of NOTCH in the ER upon overexpression of SLC35D3 and support the mechanism proposed by the authors in which SLC35D3 influences obesity via inhibiting NOTCH. As a suggest the authors remains to do some final formal editing, as for example unifying scalebars in figure 8 (they all have different styles and thickness).

Reviewer #2 (Remarks to the Author):

In the second revision of the manuscript, the authors performed a series of new immunofluorescence and cell fractionation experiments. The authors claim that these experiments are consistent with the proposed mechanism that SLC35D3 causes retention of NOTCH1 in the ER, preventing its trafficking to the plasma membrane. The quality of the new immunofluorescence data is still very poor, for example the images are pixelated, the images often appear overexposed and only representative (?) images are shown without any quantitative analysis. In addition, expression of GFP-tagged NOTCH1 constructs resulted in a vesicular pattern, although it is unclear which endosomal structures are involved. An accumulation of GFP-NOTCH1 at the plasma membrane is not visible. The claim of the authors that SLC35D3 overexpression or SLC35D3 inhibition alter the pattern of co-localization with the ER marker calnexin is not justified by the images shown in Figures 7D/E. The Western Blot data showing that SLC35D3 overexpression leads to an accumulation of NOTCH in the ER are convincing (Figure 7F). On the other hand, an enrichment of NOTCH1 at the plasma membrane in SLC35D3 knockdown cells is not visible (Figure S8D). To convincingly show that SLC35D3 regulate cellular NOTCH1 trafficking and signaling, the authors should repeat the cellular fractionation in primary adipocytes and/or adipose

tissues isolated from wild type versus Slc35d3 KI (and/or Slc35d3 KO) mice. In addition, cell surface biotinylation approaches in primary adipocytes from wild type versus Slc35d3 KI (and/or Slc35d3 KO) are needed to convincingly show quantitative differences in NOTCH1 abundance at the plasma membrane.

Responses to Reviewer #1

This manuscript has been reviewed and adapted extensively multiple times and has improved. The authors have put additional effort in demonstrating the physical interaction between the ECD of NOTCH that results in the protein's retention in the ER. The quality of IF, as well as pull down and western blot experiments has significantly improved compared to the last version. In addition, important control experiments, such as knockdown of SLC35D3 have been added. In conclusion, the new data do relatively well show a retention of NOTCH in the ER upon overexpression of SLC35D3 and support the mechanism proposed by the authors in which SLC35D3 influences obesity via inhibiting NOTCH. As a suggest the authors remains to do some final formal editing, as for example unifying scalebars in figure 8 (they all have different styles and thickness).

Response: Thank you for your helpful comments. We have standardized the styles and thickness of the scale bars in the figures, which are defined in the figure legends. In addition, we have made further revisions and refinements in the manuscript and figures.

We greatly appreciate your kind consideration, and we would like to thank you again for your valuable time and effort in reviewing our manuscript.

Responses to Reviewer #2

In the second revision of the manuscript, the authors performed a series of new immunofluorescence and cell fractionation experiments. The authors claim that these experiments are consistent with the proposed mechanism that SLC35D3 causes retention of NOTCH1 in the ER, preventing its trafficking to the plasma membrane. The quality of the new immunofluorescence data is still very poor, for example the images are pixelated, the images often appear overexposed and only representative (?) images are shown without any quantitative analysis. In addition, expression of GFP-tagged NOTCH1 constructs resulted in a vesicular pattern, although it is unclear which endosomal structures are involved. An accumulation of GFP-NOTCH1 at the plasma membrane is not visible. The claim of the authors that SLC35D3 overexpression or SLC35D3 inhibition alter the pattern of co-localization with the ER marker calnexin is not justified by the images shown in Figures 7D/E. The Western Blot data showing that SLC35D3 overexpression leads to an accumulation of NOTCH in the ER are convincing (Figure 7F). On the other hand, an enrichment of NOTCH1 at the plasma membrane in SLC35D3 knockdown cells is not visible (Figure S8D). To convincingly show that SLC35D3 regulate cellular NOTCH1 trafficking and signaling, the authors should repeat the cellular fractionation in primary adipocytes and/or adipose tissues isolated from wild type versus Slc35d3 KI (and/or Slc35d3 KO) mice. In addition, cell surface biotinylation approaches in primary adipocytes from wild type versus Slc35d3 KI (and/or Slc35d3 KO) are needed to convincingly show quantitative differences in NOTCH1 abundance at the plasma membrane.

Response: Thank you for your helpful comments. For fluorescent labeling of living cells, the molecular motion within cells was dynamic due to the active state of cells, which might result in a vesicular pattern. In addition, since our aim was to preliminarily determine the subcellular localization of NOTCH1 and ER, we used an ER living cell dye (Red 594 fluorescence) for labeling, and we did not fluorescently label the cell membrane and other organelles of the live cells because other live cell fluorescent dyes

currently available would overlap with 594 fluorescence and GFP fluorescence. The live cell imaging results only provided us with a preliminary judgment of the subcellular localization of NOTCH1 and ER.

To eliminate differences in fluorescence staining resulting from different batches of experiments, the multiplex immunofluorescence experiments shown in the figures were performed uniformly with the Opal™ 7 color kit (Akoya Biosciences, NEL811001KT), and we chose to take the fluorescence images at a similar scale to the fluorescence images presented in other articles¹⁻⁴. Representative images are shown in the figures. Although the multiplex immunofluorescence staining results we obtained were not very perfect, this did not affect our judgment of the results. To more clearly demonstrate and analyze the proportion of NOTCH1 colocalized with ER, we performed quantitative analysis of NOTCH1 (red fluorescence) and CALNEXIN (green fluorescence), and the relevant results are shown in Figure 7d-e and Figure S8a-b in the revised manuscript. In addition, to confirm the effect of SLC35D3 overexpression and knockdown on NOTCH1 levels at the cell membrane, we performed immunofluorescence staining on tissue sections of IngWAT from Slc35d3 KO mice and Slc35d3 KI mice. Tissue sections were not permeabilized to better label proteins at the cell membrane surface and reduce the effect of protein staining in the cytoplasm. The fluorescence results showed that NOTCH1 levels at the cell membrane were significantly increased in adipose tissues from SAKO mice compared to control mice (Fig. 7f-g). Additionally, the adipocyte sizes in SAKO mice were visibly larger than those in control mice, which was consistent with our previous results (Fig.3l and Fig.7f). However, NOTCH1 levels at the cell membrane were significantly lower in adipose tissues from SAKI mice than in those from control mice (Fig. S8c-d).

For cellular fractionation of SLC35D3 knockdown 293T cells, NOTCH1 levels at the cell membrane did not change much within 24–40 h after transfection, but a reduction in NOTCH1 in the ER was observed 32 h after transfection (Fig. S9e-h). However, with the extension of time (48-72 h), knockdown of SLC35D3 promoted the enrichment of NOTCH1 at the cell membrane, especially at 72 h posttransfection. Furthermore, although total NOTCH1 levels did not change at 48 h, knockdown of

SLC35D3 resulted in higher total NOTCH1 levels at 72 h (Fig. S9i-j). We also performed cellular fractionation in adipose tissues from SAKI mice and SAKO mice. The detection results further confirmed the alteration of NOTCH1 levels in the ER and the cell membrane by SLC35D3.

To further confirm the inhibition of NOTCH1 levels at the cell membrane by SLC35D3, we also employed cell-surface biotinylation assays to detect NOTCH1 on the cell membrane, including SLC35D3-overexpressing 293T cells, SLC35D3-knockdown 293T cells, and primary adipocytes isolated from SAKI mice and SAKO mice (Fig. 7k-m and Fig. S9l-n). These results confirmed the effect of SLC35D3 on NOTCH1 levels at the cell membrane.

These results support our hypothesis that SLC35D3 promotes the accumulation of NOTCH1 in the ER while decreasing the levels at the cell membrane. These are described in detail in the Results section of the revised manuscript, and the data are shown in Figure 7, Figure S7, Figure S8 and Figure S9, which are also shown below.

We greatly appreciate your helpful and valuable comments, and we would like to thank you again for your valuable time and effort in reviewing the revised version of our manuscript.

Results

SLC35D3 increased the ER accumulation of NOTCH1 via interaction with the NOTCH1 extracellular domain.

The above results further aroused our curiosity about the exact pathway through which SLC35D3 inhibited NOTCH1 signaling. Next, we examined the localization of SLC35D3 and NOTCH1 by immunofluorescence in HEK293T cells overexpressing SLC35D3. The results were very intriguing, and we found that SLC35D3 and NOTCH1 colocalized intracellularly (Supplementary Fig. 7a), suggesting a possible interaction between SLC35D3 and NOTCH1. Subsequently, we detected the interaction between SLC35D3 and NOTCH1 in HEK293T cells via a coimmunoprecipitation (CoIP) assay. As shown in Figure 7a, NOTCH1 was coimmunoprecipitated with SLC35D3 in HEK293T cells (Fig. 7a). NOTCH1 proteins consist of a large extracellular domain (ECD), a transmembrane region, and a large NICD^{5,6}. Furthermore, we determined the specific domain to which SLC35D3 binds. First, the binding between SLC35D3 and NICD was determined in HEK293T cells transfected with both the SLC35D3 plasmid and NICD plasmid via CoIP assays. As shown in Supplementary Figure 7b, no

coprecipitation of NICD with SLC35D3 was observed (Supplementary Fig. 7b). We then determined the binding between SLC35D3 and ECD, and myc-tagged ECD was coimmunoprecipitated with SLC35D3 in HEK293T cells (Fig. 7b). The direct physical interaction between SLC35D3 and the NOTCH1 ECD was further confirmed by an *in vitro* pull-down assay (Fig. 7c). Interestingly, SLC35D3 lacking the last 30 amino acids (SLC35D3-30aa) remained immunoprecipitated with the NOTCH1 ECD (Supplementary Fig. 7c). We subsequently transfected HEK293T cells with both the SLC35D3 plasmid and ECD plasmid, as well as the SLC35D3 plasmid and NICD plasmid. Multiple fluorescence staining results showed that SLC35D3 and ECD colocalized (Supplementary Fig. 7d); in contrast, NICD did not colocalize with SLC35D3 (Supplementary Fig. 7e). Taken together, these results revealed that SLC35D3 interacted with NOTCH1 through the ECD, the extracellular domain of NOTCH1.

We next sought to investigate how the interaction of SLC35D3 with NOTCH1 affected NOTCH1 signaling. In signal-receiving cells, NOTCH1 receptors are first generated in the ER and then transported to the Golgi apparatus where NOTCH1 receptors are cleaved into heterodimers (S1 cleavage) and transported to the cell membrane. Transport of NOTCH1 in the endoplasmic reticulum (ER) has been shown to play a critical role in the regulation of NOTCH1 levels at the cell membrane⁶⁻⁸. Considering that SLC35D3 also localizes to the ER intracellularly⁹, we speculated that SLC35D3 regulated the trafficking of NOTCH1 from the ER to the plasma membrane. Then, we isolated membrane and cytoplasmic proteins from 293T cells overexpressing SLC35D3, and western blotting results showed that SLC35D3 overexpression significantly decreased NOTCH1 levels at the plasma membrane but increased NOTCH1 levels in the cytosol (Supplementary Fig. 7f-g). The flow cytometry assay results further confirmed that SLC35D3 overexpression significantly reduced NOTCH1 levels at the plasma membrane (Supplementary Fig. 7h-i). These results suggested that SLC35D3 might regulate NOTCH1 trafficking. To further investigate the modulatory effect of SLC35D3 on NOTCH1, HEK293T cells were transfected with both SLC35D3 plasmid and GFP-tagged NOTCH1 plasmid, and cells transfected with both empty plasmid and GFP-tagged NOTCH1 plasmid were used as controls. Forty-eight hours after transfection, we labeled the ER in living cells using ER Tracker Red Dyes and detected NOTCH1 localization in living cells by a High Content Imaging System (Opera Phenix, PerkinElmer). Intriguingly, NOTCH1 did not show extensive colocalization with the ER in control cells; instead, NOTCH1 was more distributed at the periphery of the ER, surrounding the cell. However, in cells overexpressing SLC35D3, there was extensive colocalization of NOTCH1 with the ER, indicating that overexpression of SLC35D3 promoted NOTCH1 colocalization with the ER (Supplementary Fig. 7j).

The above results suggested that SLC35D3 might inhibit NOTCH1 signaling by promoting the ER accumulation of NOTCH1. To confirm whether SLC35D3 actually regulated NOTCH1 trafficking, we performed multiple fluorescence staining in fixed cells using an Opal multiplex staining protocol with an Opal™ 7 color kit. HEK293T cells were transfected with SLC35D3 plasmid and NOTCH1 plasmid, and we then

examined the subcellular localization of SLC35D3 and NOTCH1 in cells 48 h posttransfection. The results showed that less NOTCH1 and ER colocalization was observed in control cells, while the overexpression of SLC35D3 promoted NOTCH1 and ER colocalization (Fig. 7d-e). We further transfected HEK293T cells with siRNA targeting SLC35D3 to mediate the knockdown of SLC35D3, and then examined the intracellular localization of NOTCH1. Immunofluorescence results showed that SLC35D3 knockdown reduced NOTCH1 retention in the ER, as shown by decreased NOTCH1 colocalization with the ER in SLC35D3-knockdown cells (Supplementary Fig. 8a-b). We have demonstrated that SLC35D3 interacts with the ECD of NOTCH1, but not with the NICD, in the above results. Interestingly, the fluorescence staining results showed that SLC35D3 and ECD colocalized and both colocalized with ER (Supplementary Fig. 7d). This further confirmed the interaction between SLC35D3 and the ECD, which was mainly in the ER.

To further confirm that SLC35D3 promoted NOTCH1 accumulation in the ER and decreased NOTCH1 levels at the cell membrane, we performed immunofluorescence staining for IngWAT from SAKO mice and SAKI mice. Tissue sections were not permeabilized to better label proteins at the cell membrane surface and reduce the effect of protein staining in the cytoplasm. Fluorescence results showed that NOTCH1 levels at the cell membrane were significantly increased in adipose tissues from SAKO mice compared to control mice (Fig. 7f-g). Additionally, the adipocyte sizes in SAKO mice were visibly larger than those in control mice, which was consistent with our previous results (Fig. 3l and Fig. 7f). However, NOTCH1 levels at the cell membrane were significantly lower in adipose tissues from SAKI mice than in those from control mice (Supplementary Fig. 8c-d). We also isolated cell membrane proteins and endoplasmic reticulum proteins using a Plasma Membrane Protein Isolation and Cell Fractionation Kit and an Endoplasmic Reticulum Isolation Kit, and detected NOTCH1 levels by western blotting assays. HEK293T cells were transfected with SLC35D3 plasmid and siRNA targeting SLC35D3, and harvested at different time points to detect NOTCH1 protein levels. The results were intriguing, showing that overexpression of SLC35D3 did not alter total NOTCH1 protein levels at 48 h (Supplementary Fig. 9a-b). However, SLC35D3 overexpression significantly promoted the retention of NOTCH1 in the ER, accompanied by a decrease in NOTCH1 protein levels on the cell membrane at 40 h and 48 h (Supplementary Fig. 9a-d). In contrast, knockdown of SLC35D3 promoted trafficking of the NOTCH1 protein from the ER to the cell membrane, as shown by decreased NOTCH1 levels in the ER (Supplementary Fig. 9e-h). Interestingly, although NOTCH1 levels on the cell membrane did not change much at 40 h, knockdown of SLC35D3 significantly increased NOTCH1 levels on the cell membrane over time, and total NOTCH1 levels also increased at 72 h (Supplementary Fig. 9i-j). This finding was also consistent with previous results showing that total NOTCH1 levels were altered after prolonged exposure (Fig. 6h). Subsequently, we also isolated the cellular fractions of IngWAT from SAKI mice and SAKO mice for detection, and the results showed that NOTCH1 levels on the cell membrane were significantly higher in adipose tissues of SAKO mice than in control mice, accompanied by decreased NOTCH1 levels in the ER (Fig. 9h-i). The opposite results were observed in SAKI mice, as shown by

decreased NOTCH1 levels on the cell membrane (Fig. 9h, j). Summarizing the above results, SLC35D3 regulated NOTCH1 trafficking from the ER to the cell membrane.

To further confirm that SLC35D3 affected NOTCH1 levels on the cell membrane, we employed cell-surface biotinylation assays to detect NOTCH1 protein on the cell membrane. We first incubated 293T cells and mouse primary adipocytes with PBS and biotin (Sulfo-NHS-SS-biotin), respectively, and detected NOTCH1 proteins by western blotting. The results showed that NOTCH1 proteins on the cell membrane could indeed be labeled by biotin (Supplementary Fig. 9k). Next, we performed cell-surface biotinylation assays with 293T cells at 48 h and 72 h after transfection and the results showed that SLC35D3 overexpression significantly decreased NOTCH1 levels on the cell membrane, whereas SLC35D3 knockdown caused an increase in NOTCH1 levels on the cell membrane (Supplementary Fig. 9l-n). Primary adipocytes were tested similarly. Notch1 levels on the cell membranes in primary adipocytes from SAKO mice were significantly higher than those in primary adipocytes from control mice, and the opposite results were observed in adipocytes from SAKI mice (Fig. 7k-m).

Summarizing all the results above, SLC35D3 increased ER accumulation of NOTCH1 and thereby inhibited NOTCH1 signaling, which was achieved by interacting with the NOTCH1 extracellular domain.

Figure 7. SLC35D3 increased the ER accumulation of NOTCH1 by interacting with the NOTCH1 extracellular domain.

a-b CoIP assay results showed the interaction between SLC35D3 and NOTCH1 (a), as well as the interaction between SLC35D3 and ECD (b). Representative images from three independent experiments are shown. **c** Direct interaction between SLC35D3 and ECD was further confirmed by pull-down assay. Representative images from three

independent experiments are shown. **d-e** SLC35D3 overexpression promoted the colocalization of NOTCH1 and ER. Representative confocal images of cells are shown (d), as well as the quantification results of four independent parallel experiments (e). The inset in the right panel shows a magnified view of the indicated area. Scale bar, 20 μm . CALNEXIN, an ER marker protein. **f-g** SAKO mice showed higher levels of Notch1 at the cell membrane in IngWAT than control mice. Representative confocal images are shown (f), as well as the quantification results of six independent parallel experiments (g). Scale bar, 50 μm . Na-K ATPase, a plasma membrane marker protein. **h-j** Membrane proteins and endoplasmic reticulum proteins of IngWAT were isolated from SAKO and SAKI mice (n=3 per group). The western blot results are shown (h), as well as the quantitative analysis results (i-j). Total NOTCH1 expression was normalized to β -ACTIN expression, membrane NOTCH1 expression was normalized to Na-K ATPase, and ER NOTCH1 expression was normalized to CALNEXIN. **k-m** Primary adipocytes were isolated from the IngWAT of SAKO mice and SAKI mice (n=3 per group). Cell surface biotinylation assay results are shown (k), as well as the quantitative analysis results (l-m). Data are expressed as the mean \pm SEM. Statistical significance was assessed by unpaired two-sided Student's t test (e, g, i-j, l-m). The exact *P* values are shown in the figure. Source data are provided as a Source Data file.

Figure S7. The interaction between SLC35D3 and NOTCH1.

a In SLC35D3-overexpressing cells, SLC35D3 and NOTCH1 colocalized. Representative confocal images of three independent experiments are shown, and the inset in the bottom panel shows a magnified view of the indicated area. Scale bar, 20 μm .

b CoIP assay results showed no interaction between SLC35D3 and NICD.

Representative images from three independent experiments are shown. **c** CoIP assay results showed the interaction between SLC35D3-30aa and ECD. Representative images from three independent experiments are shown. **d-e** Multiplex fluorescence staining results of cells transfected with SLC35D3 plasmid and ECD plasmid, as well as cells transfected with SLC35D3 plasmid and NICD plasmid. SLC35D3 and ECD colocalized, and both colocalized with ER (d), while SLC35D3 and NICD did not colocalize (e). Representative confocal images of four independent experiments are shown, and the inset in the bottom panel shows a magnified view of the indicated area. Scale bar, 50 μ m. **f-g** NOTCH1 levels were downregulated in the plasma membrane when SLC35D3 was overexpressed (f). The quantitative analysis results of three independent experiments (g) are shown. **h-i** Membrane NOTCH1 levels in SLC35D3-overexpressing HEK293T cells and control cells were detected by flow cytometric analysis. Binding curves were generated by determining the NOTCH1 mean fluorescence density (MFI). The quantitative analysis results of three independent experiments (i) are shown. **j** Live cell imaging results obtained using a high-content imaging system. Representative images of four independent experiments are shown, and the inset in the right panel shows a magnified view of the indicated area. Scale bar, 20 μ m. Data are expressed as the mean \pm SEM. Statistical significance was assessed by unpaired two-sided Student's t test (g, i). The exact P values are shown in the figure. Source data are provided as a Source Data file.

Figure S8. SLC35D3 regulated NOTCH1 export from the ER.

a-b SLC35D3 knockdown reduced the colocalization of NOTCH1 and ER. Representative confocal images of cells are shown (a), as well as the quantification results of four independent parallel experiments (b). The inset in the right panel shows a magnified view of the indicated area. Scale bar, 20 μ m. CALNEXIN, an ER marker protein. **c-d** SAKI mice showed lower levels of Notch1 at the cell membrane in IngWAT than control mice. Representative confocal images are shown (c), as well as the quantification results of six independent parallel experiments (d). Scale bar, 50 μ m. Na-K ATPase, a plasma membrane marker protein. Data are expressed as the mean \pm SEM. Statistical significance was assessed by unpaired two-sided Student's *t* test (b, d). The exact *P* values are shown in the figure. Source data are provided as a Source Data file.

Figure S9. SLC35D3 regulated NOTCH1 levels in the ER and cell membrane.

a-d Representative western blotting results (a) of total NOTCH1, membrane NOTCH1, and ER NOTCH1 proteins at different time points in control and SLC35D3-overexpressing cells. The quantitative analysis results of three independent experiments (b-d) are shown. Total NOTCH1 expression was normalized to β-ACTIN expression, membrane NOTCH1 expression was normalized to Na-K ATPase, and ER NOTCH1 expression was normalized to CALNEXIN. **e-h** Representative western blotting results (e) of total NOTCH1, membrane NOTCH1, and ER NOTCH1 proteins at different time points in control and SLC35D3 knockdown cells. The quantitative analysis results of three independent experiments (f-h) are shown. **i-j** Representative western blotting results (i) of total NOTCH1, membrane NOTCH1, and ER NOTCH1 proteins 48 h after transfection (i). The quantitative analysis results of NOTCH1 levels at 72 h in three

independent experiments (j) are shown. **k** Western blotting results of cell surface biotinylation assays in 293T cells and primary adipocytes; PBS incubation served as a negative control. Representative images from three independent experiments are shown. **l-n** 293T cells were transfected with SLC35D3 plasmid and siRNA, and then cells were harvested 48 h and 72 h after transfection for cell surface biotinylation assays. Representative blots (l) from three independent experiments are shown, as well as the quantitative analysis results of three experiments (m-n). Data are expressed as the mean \pm SEM. Statistical significance was assessed by unpaired two-sided Student's t test (b-d, f-h, j, m-n). The exact P values are shown in the figure. Source data are provided as a Source Data file.

References

- 1 Ding, L. *et al.* Canagliflozin primes antitumor immunity by triggering PD-L1 degradation in endocytic recycling. *The Journal of clinical investigation* **133**, doi:10.1172/jci154754 (2023).
- 2 Song, R. *et al.* A novel polypeptide encoded by the circular RNA ZKSCAN1 suppresses HCC via degradation of mTOR. *Molecular cancer* **22**, 16, doi:10.1186/s12943-023-01719-9 (2023).
- 3 Chen, B. *et al.* Rab8 GTPase regulates Klotho-mediated inhibition of cell growth and progression by directly modulating its surface expression in human non-small cell lung cancer. *EBioMedicine* **49**, 118-132, doi:10.1016/j.ebiom.2019.10.040 (2019).
- 4 Körner, A. *et al.* Sympathetic nervous system controls resolution of inflammation via regulation of repulsive guidance molecule A. *Nat Commun* **10**, 633, doi:10.1038/s41467-019-08328-5 (2019).
- 5 Rana, N. A. *et al.* O-glucose trisaccharide is present at high but variable stoichiometry at multiple sites on mouse Notch1. *J Biol Chem* **286**, 31623-31637, doi:10.1074/jbc.M111.268243 (2011).
- 6 Kopan, R. & Ilagan, M. X. The canonical Notch signaling pathway: unfolding the activation mechanism. *Cell* **137**, 216-233, doi:10.1016/j.cell.2009.03.045 (2009).
- 7 Zhou, B. *et al.* Notch signaling pathway: architecture, disease, and therapeutics. *Signal transduction and targeted therapy* **7**, 95, doi:10.1038/s41392-022-00934-y (2022).
- 8 Siebel, C. & Lendahl, U. Notch Signaling in Development, Tissue Homeostasis, and Disease. *Physiol Rev* **97**, 1235-1294, doi:10.1152/physrev.00005.2017 (2017).
- 9 Zhang, Z. *et al.* Mutation of SLC35D3 causes metabolic syndrome by impairing dopamine signaling in striatal D1 neurons. *PLoS genetics* **10**, e1004124, doi:10.1371/journal.pgen.1004124 (2014).

REVIEWERS' COMMENTS

Reviewer #2 (Remarks to the Author):

In the second revised version, the additional data are convincing and show the relevance of SLC35D3 for NOTCH-dependent signalling pathways in adipocytes. Overall, the authors were able to adequately answer the remaining points of criticism.

REVIEWERS' COMMENTS

Reviewer #2 (Remarks to the Author):

In the second revised version, the additional data are convincing and show the relevance of SLC35D3 for NOTCH-dependent signalling pathways in adipocytes. Overall, the authors were able to adequately answer the remaining points of criticism.

Response to Reviewer 2

Thank you for your kind consideration about our work. We greatly appreciate your precious time and efforts in reviewing our manuscript.